# Accelerated Quasi-Newton Proximal Extragradient: Faster Rate for Smooth Convex Optimization

**Ruichen Jiang**
ECE Department
The University of Texas at Austin
rjiang@utexas.edu

**Aryan Mokhtari**
ECE Department
The University of Texas at Austin
mokhtari@austin.utexas.edu

## Abstract

In this paper, we present an accelerated quasi-Newton proximal extragradient method for solving unconstrained smooth convex optimization problems. With access only to the gradients of the objective function, we prove that our method can achieve a convergence rate of $\mathcal{O}\left(\min\left\{\frac{1}{k^2}, \frac{\sqrt{d\log k}}{k^{2.5}}\right\}\right)$, where $d$ is the problem dimension and $k$ is the number of iterations. In particular, in the regime where $k = \mathcal{O}(d)$, our method matches the *optimal rate* of $\mathcal{O}(\frac{1}{k^2})$ by Nesterov's accelerated gradient (NAG). Moreover, in the the regime where $k = \Omega(d\log d)$, it outperforms NAG and converges at a *faster rate* of $\mathcal{O}\left(\frac{\sqrt{d\log k}}{k^{2.5}}\right)$. To the best of our knowledge, this result is the first to demonstrate a provable gain for a quasi-Newton-type method over NAG in the convex setting. To achieve such results, we build our method on a recent variant of the Monteiro-Svaiter acceleration framework and adopt an online learning perspective to update the Hessian approximation matrices, in which we relate the convergence rate of our method to the dynamic regret of a specific online convex optimization problem in the space of matrices.

## 1 Introduction

In this paper, we consider the following unconstrained convex minimization problem

$$\min_{\mathbf{x}\in\mathbb{R}^d} \quad f(\mathbf{x}), \tag{1}$$

where the objective function $f : \mathbb{R}^d \to \mathbb{R}$ is convex and differentiable. We are particularly interested in quasi-Newton methods, which are among the most popular iterative methods for solving the problem in (1) [1–8]. Like gradient descent and other first-order methods, quasi-Newton methods require only the objective's gradients to update the iterates. On the other hand, they can better exploit the local curvature of $f$ by constructing a Hessian approximation matrix and using it as a preconditioner, leading to superior convergence performance. In particular, when the objective function in (1) is strictly convex or strongly convex, it has long been proved that quasi-Newton methods achieve an asymptotic superlinear convergence rate [7–15], which significantly improves the linear convergence rate obtained by first-order methods. More recently, there has been progress on establishing a local non-asymptotic superlinear rate of the form $\mathcal{O}((1/\sqrt{k})^k)$ for classical quasi-Newton methods and their variants [16–22].

However, all of the results above only apply under the restrictive assumption that the objective function $f$ is strictly or strongly convex. In the more general setting where $f$ is merely convex, to the best of our knowledge, there is no result that demonstrates any form of convergence improvement by quasi-Newton methods over first-order methods. More precisely, it is well known that Nesterov's accelerated gradient (NAG) [23] can achieve a convergence rate of $\mathcal{O}(1/k^2)$ if $f$ is convex and has Lipschitz gradients. On the other hand, under the same setting, asymptotic convergence of

37th Conference on Neural Information Processing Systems (NeurIPS 2023).

classical quasi-Newton methods has been shown in [13, 24] but no explicit rate is given. With certain conditions on the Hessian approximation matrices, the works in [25–27] presented quasi-Newton-type methods with convergence rates of $\mathcal{O}(1/k)$ and $\mathcal{O}(1/k^2)$, respectively, which are no better than the rate of NAG and, in fact, can be even worse in terms of constants. This gap raises the following fundamental question:

> *Can we design a quasi-Newton-type method that achieves a convergence rate faster than $\mathcal{O}(1/k^2)$ for the smooth convex minimization problem in* (1)*?*

At first glance, this may seem impossible, as for any first-order method that has access only to a gradient oracle, one can construct a "worst-case" instance and establish a lower bound of $\Omega(1/k^2)$ on the optimality gap [28, 29]. It is worth noting that while such a lower bound is typically shown under a "linear span" assumption, i.e., the methods only query points in the span of the gradients they observe, this assumption is in fact not necessary and can be removed by the technique of resisting oracle (see, [30, Section 3.3]). In particular, this $\Omega(1/k^2)$ lower bound applies for any iterative method that only queries gradients of the objective, including quasi-Newton methods. On the other hand, this lower bound is subject to a crucial assumption: it only works in the high-dimensional regime where the problem dimension $d$ exceeds the number of iterations $k$. As such, it does not rule out the possibility of a faster rate than $\mathcal{O}(1/k^2)$ when the number of iterations $k$ is larger than $d$. Hence, ideally, we are looking for a method that attains the optimal rate of $O(1/k^2)$ in the regime that $k = \mathcal{O}(d)$ and surpasses this rate in the regime that $k = \Omega(d)$.

**Contributions.** In this paper, we achieve the above goal by presenting an accelerated quasi-Newton proximal extragradient (A-QNPE) method. Specifically, under the assumptions that $f$ in (1) is convex and its gradient and Hessian are Lipschitz, we prove the following guarantees:

- From any initialization, A-QNPE can attain a global convergence rate of $\mathcal{O}\big(\min\{\frac{1}{k^2}, \frac{\sqrt{d \log k}}{k^{2.5}}\}\big)$. In particular, this implies that our method matches the optimal rate of $\mathcal{O}(\frac{1}{k^2})$ when $k = \mathcal{O}(d)$, while it converges at a faster rate of $\mathcal{O}(\frac{\sqrt{d \log k}}{k^{2.5}})$ when $k = \Omega(d \log d)$. Alternatively, we can bound the number of iterations required to achieve an $\epsilon$-accurate solution by $N_\epsilon = \mathcal{O}\big(\min\{\frac{1}{\epsilon^{0.5}}, \frac{d^{0.2}}{\epsilon^{0.4}}(\log \frac{d}{\epsilon^2})^{0.2}\}\big)$.

- In terms of computational cost, we show that the total number of gradient queries after $N$ iterations can be bounded by $3N$, i.e., on average no more than 3 per iteration. Moreover, the number of matrix-vector products to achieve an $\epsilon$-accurate solution can be bounded by $\tilde{\mathcal{O}}\big(\min\{\frac{d^{0.25}}{\epsilon^{0.5}}, \frac{1}{\epsilon^{0.625}}\}\big)$.

Combining the two results above, we conclude that A-QNPE requires $\mathcal{O}\big(\min\{\frac{1}{\epsilon^{0.5}}, \frac{d^{0.2}}{\epsilon^{0.4}}(\log \frac{d}{\epsilon^2})^{0.2}\}\big)$ gradient queries to reach an $\epsilon$-accurate solution, which is at least as good as NAG and is further superior when $\epsilon = \mathcal{O}\big(\frac{1}{d^2 \log^2(d)}\big)$. To the best of our knowledge, this is the first result that demonstrates a provable advantage of a quasi-Newton-type method over NAG in terms of gradient oracle complexity in the smooth convex setting.

To obtain these results, we significantly deviate from the classical quasi-Newton methods such as BFGS and DFP. Specifically, instead of mimicking Newton's method as in the classical updates, our A-QNPE method is built upon the celebrated Monteiro-Svaiter (MS) acceleration framework [31, 32], which can be regarded as an inexact version of the accelerated proximal point method [33, 34]. Another major difference lies in the update rule of the Hessian approximation matrix. Classical quasi-Newton methods typically perform a low-rank update of the Hessian approximation matrix while enforcing the secant condition. On the contrary, our update rule is purely driven by our convergence analysis of the MS acceleration framework. In particular, inspired by [35], we assign certain loss functions to the Hessian approximation matrices and formulate the Hessian approximation matrix update as an online convex optimization problem in the space of matrices. Therefore, we propose to update the Hessian approximation matrices via an online learning algorithm.

**Related work.** The authors in [32] proposed a refined MS acceleration framework, which simplifies the line search subroutine in the original MS method [31]. By instantiating it with an adaptive *second-order oracle*, they presented an accelerated second-order method that achieves the optimal rate of $\mathcal{O}(\frac{1}{k^{3.5}})$. The framework in [32] serves as a basis for our method, but we focus on the setting where we have access only to a *gradient oracle* and we consider a quasi-Newton-type update. Another closely related work is [35], where the authors proposed a quasi-Newton proximal extragradient method with a global non-asymptotic superlinear rate. In particular, our Hessian approximation update is inspired by the online learning framework in [35]. On the other hand, the major difference is that the authors in [35] focused on the case where $f$ is strongly convex and presented a global

superlinear rate, while we consider the more general convex setting where $f$ is only convex (may not be strongly convex). Moreover, we further incorporate the acceleration mechanism into our method, which greatly complicates the convergence analysis; see Remark 2 for more discussions.

Another class of optimization algorithms with better gradient oracle complexities than NAG are cutting plane methods [36–44], which are distinct from quasi-Newton methods we study in this paper. In particular, in the regime where $\epsilon = \tilde{\mathcal{O}}(\frac{1}{d^2})$, they can achieve the optimal gradient oracle complexity of $\mathcal{O}(d \log \frac{1}{\epsilon})$ [38]. On the other hand, in the regime where $\epsilon = \Omega(\frac{1}{d^2})$, the complexity of the cutting plane methods is worse than NAG, while our proposed method matches the complexity of NAG.

## 2  Preliminaries

Next, we formally state the required assumptions for our main results.

**Assumption 1.** *The function $f$ is twice differentiable, convex, and $L_1$-smooth. As a result, we have $0 \preceq \nabla^2 f(\mathbf{x}) \preceq L_1 \mathbf{I}$ for any $\mathbf{x} \in \mathbb{R}^d$, where $\mathbf{I} \in \mathbb{R}^{d \times d}$ is the identity matrix.*

**Assumption 2.** *The Hessian of $f$ is $L_2$-Lipschitz, i.e., we have $\|\nabla^2 f(\mathbf{x}) - \nabla^2 f(\mathbf{y})\|_{\mathrm{op}} \le L_2 \|\mathbf{x} - \mathbf{y}\|_2$ for any $\mathbf{x}, \mathbf{y} \in \mathbb{R}^d$, where $\|\mathbf{A}\|_{\mathrm{op}} \triangleq \sup_{\mathbf{x}: \|\mathbf{x}\|_2 = 1} \|\mathbf{A}\mathbf{x}\|_2$.*

We note that both assumptions are standard in the optimization literature and are satisfied by various loss functions such as the logistic loss and the log-sum-exp function (see, e.g., [16]).

*Remark* 1. We note that the additional assumption of Lipschitz Hessian does not alter the lower bound of $\Omega(1/k^2)$ that we discussed in the introduction. Indeed, this lower bound is established by a worst-case quadratic function, whose Hessian is constant (Lipschitz continuous with $L_2 = 0$). Therefore, Assumption 2 does not eliminate this worst-case construction from the considered problem class, and thus the lower bound also applies to our setting.

**Monteiro-Svaiter acceleration.** As our proposed method uses ideas from the celebrated Monteiro-Svaiter (MS) acceleration algorithm [31], we first briefly recap this method. MS acceleration, also known as accelerated hybrid proximal extragradient (A-HPE), consists of intertwining sequences of iterates $\{\mathbf{x}_k\}, \{\mathbf{y}_k\}, \{\mathbf{z}_k\}$, scalar variables $\{a_k\}$ and $\{A_k\}$ as well as step sizes $\{\eta_k\}$. The algorithm has three main steps. In the first step, we compute the auxiliary iterate $\mathbf{y}_k$ according to

$$\mathbf{y}_k = \frac{A_k}{A_k + a_k} \mathbf{x}_k + \frac{a_k}{A_k + a_k} \mathbf{z}_k, \qquad \text{where} \quad a_k = \frac{\eta_k + \sqrt{\eta_k^2 + 4\eta_k A_k}}{2}. \tag{2}$$

In the second step, an inexact proximal point step $\mathbf{x}_{k+1} \approx \mathbf{y}_k - \eta_k \nabla f(\mathbf{x}_{k+1})$ is performed. To be precise, given a parameter $\sigma \in [0, 1)$, we find $\mathbf{x}_{k+1}$ that satisfies

$$\|\mathbf{x}_{k+1} - \mathbf{y}_k + \eta_k \nabla f(\mathbf{x}_{k+1})\| \le \sigma \|\mathbf{x}_{k+1} - \mathbf{y}_k\|. \tag{3}$$

Then in the third step, the iterate $\mathbf{z}$ is updated by following the update

$$\mathbf{z}_{k+1} = \mathbf{z}_k - a_k \nabla f(\mathbf{x}_{k+1}).$$

Finally, we update the scalar $A_{k+1}$ by $A_{k+1} = A_k + a_k$. The above method has two implementation issues. First, to perform the update in (3) directly, one needs to solve the nonlinear system of equations $\mathbf{x} - \mathbf{y}_k + \eta_k \nabla f(\mathbf{x}) = 0$ to a certain accuracy, which could be costly in general. To address this issue, a principled approach is to replace the gradient operator $\nabla f(\mathbf{x})$ with a simpler approximation function $P(\mathbf{x}; \mathbf{y}_k)$ and select $\mathbf{x}_{k+1}$ as the (approximate) solution of the equation:

$$\mathbf{x}_{k+1} - \mathbf{y}_k + \eta_k P(\mathbf{x}_{k+1}; \mathbf{y}_k) = 0. \tag{4}$$

For instance, we can use $P(\mathbf{x}; \mathbf{y}_k) = \nabla f(\mathbf{y}_k)$ and accordingly (4) is equivalent to $\mathbf{x}_{k+1} = \mathbf{y}_k - \eta_k \nabla f(\mathbf{y}_k)$, leading to the accelerated first-order method in [31]. If we further have access to the Hessian oracle, we can use $P(\mathbf{x}; \mathbf{y}_k) = \nabla f(\mathbf{y}_k) + \nabla^2 f(\mathbf{y}_k)(\mathbf{x} - \mathbf{y}_k)$ and (4) becomes $\mathbf{x}_{k+1} = \mathbf{y}_k - \eta_k (\mathbf{I} + \eta_k \nabla^2 f(\mathbf{y}_k))^{-1} \nabla f(\mathbf{y}_k)$, leading to the second-order method in [31].

However, approximating $\nabla f(\mathbf{x})$ by $P(\mathbf{x}; \mathbf{y}_k)$ leads to a second issue related to finding a proper step size $\eta_k$. More precisely, one needs to first select $\eta_k$, compute $\mathbf{y}_k$ from (2), and then solve the system in (4) exactly or approximately to obtain $\mathbf{x}_{k+1}$. However, these three variables, i.e., $\mathbf{x}_{k+1}, \mathbf{y}_k$ and $\eta_k$ may not satisfy the condition in (3) due to the gap between $\nabla f(\mathbf{x})$ and $P(\mathbf{x}; \mathbf{y}_k)$. If that happens, we need to re-select $\eta_k$ and recalculate both $\mathbf{y}_k$ and $\mathbf{x}_{k+1}$ until the condition in (3) is satisfied. To

address this issue, several bisection subroutines have been proposed in the literature [31, 45–47] and they all incur a computational cost of $\log(1/\epsilon)$ per iteration.

**Optimal Monteiro-Svaiter acceleration.** A recent paper [32] refines the MS acceleration algorithm by separating the update of $\mathbf{y}_k$ from the line search subroutine. In particular, in the first stage, we use $\eta_k$ to compute $a_k$ and then $\mathbf{y}_k$ from (2), which will stay fixed throughout the line search scheme. In the second stage, we aim to find a pair $\hat{\mathbf{x}}_{k+1}$ and $\hat{\eta}_k$ such that they satisfy

$$\|\hat{\mathbf{x}}_{k+1} - \mathbf{y}_k + \hat{\eta}_k \nabla f(\hat{\mathbf{x}}_{k+1})\| \leq \sigma \|\hat{\mathbf{x}}_{k+1} - \mathbf{y}_k\|. \tag{5}$$

To find that pair, we follow a similar line search scheme as above, with the key difference that $\mathbf{y}_k$ is fixed and $\hat{\eta}_k$ can be different from $\eta_k$ that is used to compute $\mathbf{y}_k$. More precisely, for a given $\hat{\eta}_k$, we find the solution of (4) denoted by $\hat{\mathbf{x}}_{k+1}$ and check whether it satisfies (5) or not. If it does not, then we adapt the step size and redo the process until (5) is satisfied. Then given the values of these two parameters $\eta_k$ and $\hat{\eta}_k$, the updates for $\mathbf{x}$ and $\mathbf{z}$ would change as we describe next:

- If $\hat{\eta}_k \geq \eta_k$, we update $\mathbf{x}_{k+1} = \hat{\mathbf{x}}_{k+1}$, $A_{k+1} = A_k + a_k$ and $\mathbf{z}_{k+1} = \mathbf{z}_k - a_k \nabla f(\hat{\mathbf{x}}_{k+1})$. Moreover, we increase the next tentative step size by choosing $\eta_{k+1} = \eta_k/\beta$ for some $\beta \in (0, 1)$.

- Otherwise, if $\hat{\eta}_k < \eta_k$, the authors in [32] introduced a *momentum damping mechanism*. Define $\gamma_k = \hat{\eta}_k/\eta_k < 1$. We then choose $\mathbf{x}_{k+1} = \frac{(1-\gamma_k)A_k}{A_k+\gamma_k a_k}\mathbf{x}_k + \frac{\gamma_k(A_k+a_k)}{A_k+\gamma_k a_k}\hat{\mathbf{x}}_{k+1}$, which is a convex combination of $\mathbf{x}_k$ and $\hat{\mathbf{x}}_{k+1}$. Moreover, we update $A_{k+1} = A_k + \gamma_k a_k$ and $\mathbf{z}_{k+1} = \mathbf{z}_k - \gamma_k a_k \nabla f(\hat{\mathbf{x}}_{k+1})$. Finally, we decrease the next tentative step size by choosing $\eta_{k+1} = \beta \eta_k$.

This approach not only simplifies the procedure by separating the update of $\{\mathbf{y}_k\}$ from the line search scheme, but it also shaves a factor of $\log(1/\epsilon)$ from the computational cost of the algorithm, leading to optimal first and second-order variants of the MS acceleration method. Therefore, as we will discuss in the next section, we build our method upon this more refined MS acceleration framework.

## 3 Accelerated Quasi-Newton Proximal Extragradient

In this section, we present our accelerated quasi-Newton proximal extragradient (A-QNPE) method. An informal description of our method is provided in Algorithm 1. On a high level, our method can be viewed as the quasi-Newton counterpart of the adaptive Monteiro-Svaiter-Newton method proposed in [32]. In particular, we only query a gradient oracle and choose the approximation function in (4) as $P(\mathbf{x}; \mathbf{y}_k) = \nabla f(\mathbf{y}_k) + \mathbf{B}_k(\mathbf{x} - \mathbf{y}_k)$, where $\mathbf{B}_k$ is a Hessian approximation matrix obtained only using gradient information. Moreover, another central piece of our method is the update scheme of $\mathbf{B}_k$. Instead of following the classical quasi-Newton updates such as BFGS or DFP, we use an online learning framework, where we choose a sequence of matrices $\mathbf{B}_k$ to achieve a small dynamic regret for an online learning problem defined by our analysis; more details will be provided later in Section 3.2. We initialize our method by choosing $\mathbf{x}_0, \mathbf{z}_0 \in \mathbb{R}^d$ and setting $A_0 = 0$ and $\eta_0 = \sigma_0$, where $\sigma_0$ is a user-specified parameter. Our method can be divided into the following four stages:

- In the **first stage**, we compute the scalar $a_k$ and the auxiliary iterate $\mathbf{y}_k$ according to (2) using the step size $\eta_k$. Note that $\mathbf{y}_k$ is then fixed throughout the $k$-th iteration.

- In the **second stage**, given the Hessian approximation matrix $\mathbf{B}_k$ and the iterate $\mathbf{y}_k$, we use a line search scheme to find the step size $\hat{\eta}_k$ and the iterate $\hat{\mathbf{x}}_{k+1}$ such that

$$\|\hat{\mathbf{x}}_{k+1} - \mathbf{y}_k + \hat{\eta}_k(\nabla f(\mathbf{y}_k) + \mathbf{B}_k(\hat{\mathbf{x}}_{k+1} - \mathbf{y}_k))\| \leq \alpha_1 \|\hat{\mathbf{x}}_{k+1} - \mathbf{y}_k\|, \tag{6}$$

$$\|\hat{\mathbf{x}}_{k+1} - \mathbf{y}_k + \hat{\eta}_k \nabla f(\hat{\mathbf{x}}_{k+1})\| \leq (\alpha_1 + \alpha_2)\|\hat{\mathbf{x}}_{k+1} - \mathbf{y}_k\|, \tag{7}$$

where $\alpha_1 \in [0, 1)$ and $\alpha_2 \in (0, 1)$ are user-specified parameters with $\alpha_1 + \alpha_2 < 1$. The first condition in (6) requires that $\hat{\mathbf{x}}_{k+1}$ inexactly solves the linear system of equations $(\mathbf{I} + \hat{\eta}_k \mathbf{B}_k)(\mathbf{x} - \mathbf{y}_k) + \hat{\eta}_k \nabla f(\mathbf{y}_k) = 0$, where $\alpha_1 \in [0, 1)$ controls the accuracy. As a special case, we have $\hat{\mathbf{x}}_{k+1} = \mathbf{y}_k - (\mathbf{I} + \hat{\eta}_k \mathbf{B}_k)^{-1}\nabla f(\mathbf{y}_k)$ when $\alpha_1 = 0$. The second condition in (7) directly comes from (5) in the optimal MS acceleration framework, which ensures that we approximately follow the proximal point step $\hat{\mathbf{x}}_{k+1} = \mathbf{y}_k - \hat{\eta}_k \nabla f(\hat{\mathbf{x}}_{k+1})$. To find the pair $(\hat{\eta}_k, \hat{\mathbf{x}}_{k+1})$ satisfying both (6) and (7), we implement a backtracking line search scheme. Specifically, for some $\beta \in (0, 1)$, we iteratively try $\hat{\eta}_k = \eta_k \beta^i$ for $i \geq 0$ and solve $\hat{\mathbf{x}}_{k+1}$ from (6) until the condition in (7) is satisfied. The line search scheme will be discussed in more detail in Section 3.1.

---

**Algorithm 1** Accelerated Quasi-Newton Proximal Extragradient Method

---

1: **Input:** initial points $\mathbf{x}_0, \mathbf{z}_0 \in \mathbb{R}^d$, initial step size $\sigma_0 > 0$, $\alpha_1, \alpha_2 \in (0,1)$ with $\alpha_1 + \alpha_2 < 1$, $\beta \in (0,1)$
2: **Initialize:** set $A_0 \leftarrow 0$ and $\eta_0 \leftarrow \sigma_0$
3: **for** iteration $k = 0, \ldots, N-1$ **do**
4:    Compute $a_k \leftarrow \frac{\eta_k + \sqrt{\eta_k^2 + 4\eta_k A_k}}{2}$ and $\mathbf{y}_k \leftarrow \frac{A_k}{A_k + a_k}\mathbf{x}_k + \frac{a_k}{A_k + a_k}\mathbf{z}_k$
5:    Let $\hat{\eta}_k$ be the largest possible step size in $\{\eta_k \beta^i : i \geq 0\}$ such that

$$\hat{\mathbf{x}}_{k+1} \approx_{\alpha_1} \mathbf{y}_k - \hat{\eta}_k(\mathbf{I} + \hat{\eta}_k \mathbf{B}_k)^{-1}\nabla f(\mathbf{y}_k) \ \text{ and } \ \|\hat{\mathbf{x}}_{k+1} - \mathbf{y}_k - \hat{\eta}_k \nabla f(\hat{\mathbf{x}}_{k+1})\| \leq (\alpha_1 + \alpha_2)\|\hat{\mathbf{x}}_{k+1} - \mathbf{y}_k\|$$

6:    **if** $\hat{\eta}_k = \eta_k$ **then**
7:       Set $\mathbf{x}_{k+1} \leftarrow \hat{\mathbf{x}}_{k+1}$, $\mathbf{z}_{k+1} \leftarrow \mathbf{z}_k - a_k\nabla f(\hat{\mathbf{x}}_{k+1})$, $A_{k+1} \leftarrow A_k + a_k$
8:       Set $\eta_{k+1} \leftarrow \hat{\eta}_k/\beta$
9:       Set $\mathbf{B}_{k+1} \leftarrow \mathbf{B}_k$
10:   **else**
11:      Let $\gamma_k \leftarrow \hat{\eta}_k/\eta_k < 1$
12:      Set $\mathbf{x}_{k+1} \leftarrow \frac{(1-\gamma_k)A_k}{A_k + \gamma_k a_k}\mathbf{x}_k + \frac{\gamma_k(A_k + a_k)}{A_k + \gamma_k a_k}\hat{\mathbf{x}}_{k+1}$, $\mathbf{z}_{k+1} \leftarrow \mathbf{z}_k - \gamma_k a_k\nabla f(\hat{\mathbf{x}}_{k+1})$, $A_{k+1} \leftarrow A_k + \gamma_k a_k$
13:      Set $\eta_{k+1} \leftarrow \hat{\eta}_k$
14:      Set $\mathbf{w}_k \leftarrow \nabla f(\tilde{\mathbf{x}}_k) - \nabla f(\mathbf{y}_k)$ and $\mathbf{s}_k \leftarrow \tilde{\mathbf{x}}_k - \mathbf{y}_k$, where $\tilde{\mathbf{x}}_k$ is the last rejected iterate in LS
15:      Feed $\ell_k(\mathbf{B}) \triangleq \frac{\|\mathbf{w}_k - \mathbf{B}\mathbf{s}_k\|^2}{\|\mathbf{s}_k\|^2}$ to an online learning algorithm and obtain $\mathbf{B}_{k+1}$
16:   **end if**
17: **end for**

---

- In the **third stage**, we update the variables $\mathbf{x}_{k+1}$, $\mathbf{z}_{k+1}$, $A_{k+1}$ and set the step size $\eta_{k+1}$ in the next iteration. Specifically, the update rule we follow depends on the outcome of the line search scheme. In the first case where $\hat{\eta}_k = \eta_k$, i.e., the line search scheme accepts the initial trial step size, we let

$$\mathbf{x}_{k+1} = \hat{\mathbf{x}}_{k+1}, \quad \mathbf{z}_{k+1} = \mathbf{z}_k - a_k\nabla f(\hat{\mathbf{x}}_{k+1}), \quad A_{k+1} = A_k + a_k, \tag{8}$$

  as in the original MS acceleration framework. Moreover, this also suggests our choice of the step size $\eta_k$ may be too conservative. Therefore, we increase the step size in the next iteration by $\eta_{k+1} = \hat{\eta}_k/\beta$. In the second case where $\hat{\eta}_k < \eta_k$, i.e., the line search scheme backtracks, we adopt the momentum damping mechanism in [32]:

$$\mathbf{x}_{k+1} = \frac{(1-\gamma_k)A_k}{A_k + \gamma_k a_k}\mathbf{x}_k + \frac{\gamma_k(A_k + a_k)}{A_k + \gamma_k a_k}\hat{\mathbf{x}}_{k+1}, \ \mathbf{z}_{k+1} = \mathbf{z}_k - \gamma_k a_k\nabla f(\hat{\mathbf{x}}_{k+1}), \ A_{k+1} = A_k + \gamma_k a_k, \tag{9}$$

  where $\gamma_k = \hat{\eta}_k/\eta_k < 1$. Accordingly, we decrease the step size in the next iteration by letting $\eta_{k+1} = \hat{\eta}_k$ (note that $\hat{\eta}_k < \eta_k$).

- In the **fourth stage**, we update the Hessian approximation matrix $\mathbf{B}_{k+1}$. Inspired by [35], we depart from the classical quasi-Newton methods and instead let the convergence analysis guide our update scheme. As we will show in Section 3.2, the convergence rate of our method is closely related to the cumulative loss $\sum_{k \in \mathcal{B}} \ell_k(\mathbf{B}_k)$ incurred by our choices of $\{\mathbf{B}_k\}$, where $\mathcal{B} = \{k : \hat{\eta}_k < \eta_k\}$ denotes the indices where the line search scheme backtracks. Moreover, the loss function has the form $\ell_k(\mathbf{B}_k) \triangleq \frac{\|\mathbf{w}_k - \mathbf{B}_k\mathbf{s}_k\|^2}{\|\mathbf{s}_k\|^2}$, where $\mathbf{w}_k \triangleq \nabla f(\tilde{\mathbf{x}}_k) - \nabla f(\mathbf{x}_k)$, $\mathbf{s}_k \triangleq \tilde{\mathbf{x}}_k - \mathbf{x}_k$ and $\tilde{\mathbf{x}}_k$ is an auxiliary iterate returned by our line search scheme. Thus, this motivates us to employ an online learning algorithm to minimize the cumulative loss. Specifically, in the first case where $\hat{\eta}_k = \eta_k$ (i.e., $k \notin \mathcal{B}$), the current Hessian approximation matrix $\mathbf{B}_k$ does not contribute to the cumulative loss and thus we keep it unchanged (cf. Line 9). Otherwise, we follow an online learning algorithm in the space of matrices. The details will be discussed in Section 3.2.

Finally, we provide a convergence result in the following Proposition for Algorithm 1, which serves as the basis for our convergence analysis. We note that the following results do not require additional conditions on $\mathbf{B}_k$ other than the ones in (6) and (7). The proof is available in Appendix A.1.

**Proposition 1.** *Let $\{\mathbf{x}_k\}_{k=0}^N$ be the iterates generated by Algorithm 1. If $f$ is convex, we have*

$$f(\mathbf{x}_N) - f(\mathbf{x}^*) \leq \frac{\|\mathbf{z}_0 - \mathbf{x}^*\|^2}{2A_N} \quad \text{and} \quad A_N \geq \frac{(1-\sqrt{\beta})^2}{4(2-\sqrt{\beta})^2}\left(\sum_{k=0}^{N-1}\sqrt{\hat{\eta}_k}\right)^2.$$

Proposition 1 characterizes the convergence rate of Algorithm 1 by the quantity $A_N$, which can be further lower bounded in terms of the step sizes $\{\hat{\eta}_k\}$. Moreover, we can observe that larger step sizes will lead to a faster convergence rate. On the other hand, the step size $\hat{\eta}_k$ is constrained by the condition in (7), which, in turn, depends on our choice of the Hessian approximation matrix $\mathbf{B}_k$. Thus, the central goal of our line search scheme and the Hessian approximation update is to make the step size $\hat{\eta}_k$ as large as possible, which we will describe next.

### 3.1 Line Search Subroutine

In this section, we specify our line search subroutine to select the step size $\hat{\eta}_k$ and the iterate $\hat{\mathbf{x}}_{k+1}$ in the second stage of A-QNPE. For simplicity, denote $\nabla f(\mathbf{y}_k)$ by $\mathbf{g}$ and drop the subscript $k$ in $\mathbf{y}_k$ and $\mathbf{B}_k$. In light of (6) and (7), our goal in the second stage is to find a pair $(\hat{\eta}, \hat{\mathbf{x}}_+)$ such that

$$\|\hat{\mathbf{x}}_+ - \mathbf{y} + \hat{\eta}(\mathbf{g} + \mathbf{B}(\hat{\mathbf{x}}_+ - \mathbf{y}))\| \leq \alpha_1 \|\hat{\mathbf{x}}_+ - \mathbf{y}\|, \tag{10}$$

$$\|\hat{\mathbf{x}}_+ - \mathbf{y} + \hat{\eta}\nabla f(\hat{\mathbf{x}}_+)\| \leq (\alpha_1 + \alpha_2)\|\hat{\mathbf{x}}_+ - \mathbf{y}\|. \tag{11}$$

As mentioned in the previous section, the condition in (10) can be satisfied by solving the linear system $(\mathbf{I} + \hat{\eta}\mathbf{B})(\hat{\mathbf{x}}_+ - \mathbf{y}) = -\hat{\eta}\mathbf{g}$ to a desired accuracy. Specifically, we let

$$\mathbf{s}_+ = \mathsf{LinearSolver}(\mathbf{I} + \hat{\eta}\mathbf{B}, -\hat{\eta}\mathbf{g}; \alpha_1) \quad \text{and} \quad \hat{\mathbf{x}}_+ = \mathbf{y} + \mathbf{s}_+, \tag{12}$$

where the oracle LinearSolver is defined as follows.

**Definition 1.** *The oracle* $\mathsf{LinearSolver}(\mathbf{A}, \mathbf{b}; \alpha)$ *takes a matrix* $\mathbf{A} \in \mathbb{S}_+^d$, *a vector* $\mathbf{b} \in \mathbb{R}^d$ *and* $\alpha \in (0, 1)$ *as input, and returns an approximate solution* $\mathbf{s}_+$ *satisfying* $\|\mathbf{A}\mathbf{s}_+ - \mathbf{b}\| \leq \alpha\|\mathbf{s}_+\|$.

The most direct way to implement $\mathsf{LinearSolver}(\mathbf{A}, \mathbf{b}; \alpha)$ is to compute $\mathbf{s}_+ = \mathbf{A}^{-1}\mathbf{b}$, which however costs $\mathcal{O}(d^3)$ arithmetic operations. Alternatively, we can implement the oracle more efficiently by using the conjugate residual method [48], which only requires computing matrix-vector products and thus incurs a cost of $\mathcal{O}(d^2)$. The details are discussed in Appendix E.1. We characterize the total number of required matrix-vector products for this oracle in Theorem 2.

Now we are ready to describe our line search scheme with the LinearSolver oracle (see also Subroutine 1 in Appendix B). Specifically, we start with the step size $\eta$ and then reduce it by a factor $\beta$ until we find a pair $(\hat{\eta}, \hat{\mathbf{x}}_+)$ that satisfies (11). It can be shown that the line search scheme will terminate in a finite number of steps and return a pair $(\hat{\eta}, \hat{\mathbf{x}}_+)$ satisfying both conditions in (10) and (11) (see Appendix B.1). Regarding the output, we distinguish two cases: (i) If we pass the test in (11) on our first attempt, we accept the initial step size $\eta$ and the corresponding iterate $\hat{\mathbf{x}}_+$ (cf. Line 10 in Subroutine 1). (ii) Otherwise, along with the pair $(\hat{\eta}, \hat{\mathbf{x}}_+)$, we also return an auxiliary iterate $\tilde{\mathbf{x}}_+$ that we compute from (12) using the rejected step size $\hat{\eta}/\beta$ (cf. Line 12 in Subroutine 1). As we shall see in Lemma 1, the iterate $\tilde{\mathbf{x}}_+$ is used to derive a lower bound on $\hat{\eta}$, which will be the key to our convergence analysis and guide our update of the Hessian approximation matrix. For ease of notation, let $\mathcal{B}$ be the set of iteration indices where the line search scheme backtracks, i.e., $\mathcal{B} \triangleq \{k : \hat{\eta}_k < \eta_k\}$.

**Lemma 1.** *For* $k \notin \mathcal{B}$ *we have* $\hat{\eta}_k = \eta_k$, *while for* $k \in \mathcal{B}$ *we have*

$$\hat{\eta}_k > \frac{\alpha_2\beta\|\tilde{\mathbf{x}}_{k+1} - \mathbf{y}_k\|}{\|\nabla f(\tilde{\mathbf{x}}_{k+1}) - \nabla f(\mathbf{y}_k) - \mathbf{B}_k(\tilde{\mathbf{x}}_{k+1} - \mathbf{y}_k)\|} \quad and \quad \|\tilde{\mathbf{x}}_{k+1} - \mathbf{y}_k\| \leq \frac{(1 + \alpha_1)}{\beta(1 - \alpha_1)}\|\hat{\mathbf{x}}_{k+1} - \mathbf{y}_k\|. \tag{13}$$

Lemma 1 provides a lower bound on the step size $\hat{\eta}_k$ in terms of the approximation error $\|\nabla f(\tilde{\mathbf{x}}_{k+1}) - \nabla f(\mathbf{y}_k) - \mathbf{B}_k(\tilde{\mathbf{x}}_{k+1} - \mathbf{y}_k)\|$. Hence, a better Hessian approximation matrix $\mathbf{B}_k$ leads to a larger step size, which in turn implies faster convergence. Also note that the lower bound uses the auxiliary iterate $\tilde{\mathbf{x}}_{k+1}$ that is not accepted as the actual iterate. Thus, the second inequality in (13) will be used to relate $\|\tilde{\mathbf{x}}_{k+1} - \mathbf{y}_k\|$ with $\|\hat{\mathbf{x}}_{k+1} - \mathbf{y}_k\|$. Finally, we remark that to fully characterize the computational cost, we need to upper bound the total number of line search steps, each of which requires a call to LinearSolver and a call to the gradient oracle. This will be discussed in Theorem 2.

### 3.2 Hessian Approximation Update via Online Learning with Dynamic Regret

In this section, we discuss how to update the Hessian approximation matrix $\mathbf{B}_k$ in the fourth stage of A-QNPE. As mentioned earlier, instead of following the classical quasi-Newton updates, we directly motivate our update policy for $\mathbf{B}_k$ from the convergence analysis. The first step is to connect the

convergence rate of A-QNPE with the Hessian approximation matrices $\{\mathbf{B}_k\}$. By Proposition 1, if we define the absolute constant $C_1 \triangleq \frac{2(2-\sqrt{\beta})^2}{(1-\sqrt{\beta})^2}$, then we can write

$$f(\mathbf{x}_N) - f(\mathbf{x}^*) \leq \frac{\|\mathbf{z}_0 - \mathbf{x}^*\|^2}{2A_N} \leq \frac{C_1\|\mathbf{z}_0 - \mathbf{x}^*\|^2}{\left(\sum_{k=0}^{N-1} \sqrt{\hat{\eta}_k}\right)^2} \leq \frac{C_1\|\mathbf{z}_0 - \mathbf{x}^*\|^2}{N^{2.5}}\sqrt{\sum_{k=0}^{N-1} \frac{1}{\hat{\eta}_k^2}}, \qquad (14)$$

where the last inequality follows from Hölder's inequality. Furthermore, we can establish an upper bound on $\sum_{k=0}^{N-1} \frac{1}{\hat{\eta}_k^2}$ in terms of the Hessian approximation matrices $\{\mathbf{B}_k\}$, as we show next.

**Lemma 2.** *Let $\{\hat{\eta}_k\}_{k=0}^{N-1}$ be the step sizes in Algorithm 1 using Subroutine 1. Then we have*

$$\sum_{k=0}^{N-1} \frac{1}{\hat{\eta}_k^2} \leq \frac{2-\beta^2}{(1-\beta^2)\sigma_0^2} + \frac{2-\beta^2}{(1-\beta^2)\alpha_2^2\beta^2} \sum_{0 \leq k \leq N-1, k \in \mathcal{B}} \frac{\|\mathbf{w}_k - \mathbf{B}_k\mathbf{s}_k\|^2}{\|\mathbf{s}_k\|^2}, \qquad (15)$$

*where $\mathbf{w}_k \triangleq \nabla f(\tilde{\mathbf{x}}_{k+1}) - \nabla f(\mathbf{y}_k)$ and $\mathbf{s}_k \triangleq \tilde{\mathbf{x}}_{k+1} - \mathbf{y}_k$ for $k \in \mathcal{B}$.*

The proof of Lemma 2 is given in Appendix C.1. On a high level, for those step sizes $\hat{\eta}_k$ with $k \in \mathcal{B}$, we can apply Lemma 1 and directly obtain a lower bound in terms of $\mathbf{B}_k$. On the other hand, for $k \notin \mathcal{B}$, we have $\hat{\eta}_k = \eta_k$ and our update rule in Lines 8 and 13 of Algorithm 1 allows us to connect the sequence $\{\eta_k\}_{k=0}^{N-1}$ with the backtracked step sizes $\{\hat{\eta}_k : k \in \mathcal{B}\}$. As a result, we note that the sum in (15) only involves the Hessian approximation matrices $\{\mathbf{B}_k : k \in \mathcal{B}\}$.

In light of (14) and (15), our update for $\mathbf{B}_k$ aims to make the right-hand side of (15) as small as possible. To achieve this, we adopt the online learning approach in [35] and view the sum in (15) as the cumulative loss incurred by our choice of $\{\mathbf{B}_k\}$. To formalize, define the loss at iteration $k$ by

$$\ell_k(\mathbf{B}) \triangleq \begin{cases} 0, & \text{if } k \notin \mathcal{B}, \\ \frac{\|\mathbf{w}_k - \mathbf{B}\mathbf{s}_k\|^2}{\|\mathbf{s}_k\|^2}, & \text{otherwise,} \end{cases} \qquad (16)$$

and consider the following online learning problem: (i) At the $k$-th iteration, we choose $\mathbf{B}_k \in \mathcal{Z}$ where $\mathcal{Z} \triangleq \{\mathbf{B} \in \mathbb{S}_+^d : 0 \preceq \mathbf{B} \preceq L_1\mathbf{I}\}$; (ii) We receive the loss function $\ell_k(\mathbf{B})$ defined in (16); (iii) We update our Hessian approximation matrix to $\mathbf{B}_{k+1}$. Therefore, we propose to employ an online learning algorithm to update the Hessian approximation matrices $\{\mathbf{B}_k\}$, and the task of proving a convergence rate for our A-QNPE algorithm boils down to analyzing the performance of our online learning algorithm. In particular, an upper bound on the cumulative loss $\sum_{k=0}^{N-1} \ell_k(\mathbf{B}_k)$ will directly translate into a convergence rate for A-QNPE by using (14) and (15).

Naturally, the first idea is to update $\mathbf{B}_k$ by following projected online gradient descent [49]. While this approach would indeed serve our purpose, its implementation could be computationally expensive. Specifically, like other projection-based methods, it requires computing the Euclidean projection onto the set $\mathcal{Z}$ in each iteration, which in our case amounts to performing a full $d \times d$ matrix eigendecomposition and would incur a cost of $\mathcal{O}(d^3)$ (see Appendix C.2). Inspired by the recent work in [50], we circumvent this issue by using a projection-free online learning algorithm, which relies on an approximate separation oracle for $\mathcal{Z}$ instead of a projection oracle. For simplicity, we first translate and rescale the set $\mathcal{Z}$ via the transform $\hat{\mathbf{B}} = \frac{2}{L_1}(\mathbf{B} - \frac{L_1}{2}\mathbf{I})$ to obtain $\hat{\mathcal{Z}} \triangleq \{\hat{\mathbf{B}} \in \mathbb{S}^d : \|\hat{\mathbf{B}}\|_{\text{op}} \leq 1\}$. The approximate separation oracle $\mathsf{SEP}(\mathbf{W}; \delta, q)$ is then defined as follows.

**Definition 2.** *The oracle $\mathsf{SEP}(\mathbf{W}; \delta, q)$ takes a symmetric matrix $\mathbf{W} \in \mathbb{S}^d$, $\delta > 0$, and $q \in (0, 1)$ as input and returns a scalar $\gamma > 0$ and a matrix $\mathbf{S} \in \mathbb{S}^d$ with one of the following possible outcomes:*

- *Case I: $\gamma \leq 1$, which implies that, with probability at least $1 - q$, $\mathbf{W} \in \hat{\mathcal{Z}}$;*

- *Case II: $\gamma > 1$, which implies that, with probability at least $1 - q$, $\mathbf{W}/\gamma \in \hat{\mathcal{Z}}$, $\|\mathbf{S}\|_F \leq 3$ and $\langle \mathbf{S}, \mathbf{W} - \hat{\mathbf{B}} \rangle \geq \gamma - 1 - \delta$ for any $\hat{\mathbf{B}}$ such that $\hat{\mathbf{B}} \in \hat{\mathcal{Z}}$.*

To sum up, $\mathsf{SEP}(\mathbf{W}; \delta, q)$ has two possible outcomes: with probability $1 - q$, either it certifies that $\mathbf{W} \in \hat{\mathcal{Z}}$, or it produces a scaled version of $\mathbf{W}$ that belongs to $\hat{\mathcal{Z}}$ and an approximate separation hyperplane between $\mathbf{W}$ and the set $\hat{\mathcal{Z}}$. As we show in Appendix E.2, implementing this oracle

requires computing the two extreme eigenvectors and eigenvalues of the matrix $\mathbf{W}$ inexactly, which can be implemented efficiently by the randomized Lanczos method [51].

Building on the $\mathrm{SEP}(\mathbf{W}; \delta, q)$ oracle, we design a projection-free online learning algorithm adapted from [35, Subroutine 2]. Since the algorithm is similar to the one proposed in [35], we relegate the details to Appendix C but sketch the main steps in the analysis. To upper bound the cumulative loss $\sum_{k=0}^{N-1} \ell_k(\mathbf{B}_k)$, we compare the performance of our online learning algorithm against a sequence of reference matrices $\{\mathbf{H}_k\}_{k=0}^{N-1}$. Specifically, we aim to control the *dynamic regret* [49, 52, 53] defined by $\mathrm{D\text{-}Reg}_N(\{\mathbf{H}_k\}_{k=0}^{N-1}) \triangleq \sum_{k=0}^{N-1} (\ell_k(\mathbf{B}_k) - \ell_k(\mathbf{H}_k))$, as well as the the cumulative loss $\sum_{k=0}^{N-1} \ell_k(\mathbf{H}_k)$ by the reference sequence. In particular, in our analysis we show that the choice of $\mathbf{H}_k \triangleq \nabla^2 f(\mathbf{y}_k)$ for $k = 0, \ldots, N-1$ allows us to upper bound both quantities.

*Remark* 2. While our online learning algorithm is similar to the one in [35], our analysis is more challenging due to the lack of strong convexity. Specifically, since $f$ is assumed to be strongly convex in [35], the iterates converge to $\mathbf{x}^*$ at least linearly, resulting in less variation in the loss functions $\{\ell_k\}$. Hence, the authors in [35] let $\mathbf{H}_k = \mathbf{H}^* \triangleq \nabla^2 f(\mathbf{x}^*)$ for all $k$ and proved that $\sum_{k=0}^{N-1} \ell_k(\mathbf{H}^*)$ remains bounded. In contrast, without linear convergence, we need to use a time-varying sequence $\{\mathbf{H}_k\}$ to control the cumulative loss. This in turn requires us to bound the variation $\sum_{k=0}^{N-2} \|\mathbf{H}_{k+1} - \mathbf{H}_k\|_F$, which involves a careful analysis of the stability property of the sequence $\{\mathbf{y}_k\}$ in Algorithm 1.

## 4 Complexity Analysis of A-QNPE

In this section, we present our main theoretical results: we establish the convergence rate of A-QNPE (Theorem 1) and characterize its computational cost in terms of gradient queries and matrix-vector product evaluations (Theorem 2). The proofs are provided in Appendices D and E.3.

**Theorem 1.** *Let $\{\mathbf{x}_k\}$ be the iterates generated by Algorithm 1 using the line search scheme in Section 3.1, where $\alpha_1, \alpha_2 \in (0, 1)$ with $\alpha_1 + \alpha_2 < 1$ and $\beta \in (0, 1)$, and using the Hessian approximation update in Section 3.2 (the hyperparameters are given in Appendix D). Then with probability at least $1 - p$, the following statements hold, where $C_i$ ($i = 4, \ldots, 10$) are absolute constants only depending on $\alpha_1$, $\alpha_2$ and $\beta$.*

*(a) For any $k \geq 0$, we have*

$$f(\mathbf{x}_k) - f(\mathbf{x}^*) \leq \frac{C_4 L_1 \|\mathbf{z}_0 - \mathbf{x}^*\|^2}{k^2} + \frac{C_5 \|\mathbf{z}_0 - \mathbf{x}^*\|^2}{\sigma_0 k^{2.5}}. \tag{17}$$

*(b) Furthermore, for any $k \geq 0$,*

$$f(\mathbf{x}_k) - f(\mathbf{x}^*) \leq \frac{\|\mathbf{z}_0 - \mathbf{x}^*\|^2}{k^{2.5}} \left( M + C_{10} L_1 L_2 d \|\mathbf{z}_0 - \mathbf{x}^*\| \log^+ \left( \frac{\max\{\frac{L_1}{\alpha_2 \beta}, \frac{1}{\sigma_0}\} k^{2.5}}{\sqrt{M}} \right) \right)^{\frac{1}{2}}, \tag{18}$$

*where we define $\log^+(x) \triangleq \max\{\log(x), 0\}$ and the quantity $M$ is given by*

$$M \triangleq \frac{C_6}{\sigma_0^2} + C_7 L_1^2 + C_8 \|\mathbf{B}_0 - \nabla^2 f(\mathbf{z}_0)\|_F^2 + C_9 L_2^2 \|\mathbf{z}_0 - \mathbf{x}^*\|^2 + C_{10} L_1 L_2 d \|\mathbf{z}_0 - \mathbf{x}^*\|. \tag{19}$$

Both results in Theorem 1 are global, as they are valid for any initial points $\mathbf{x}_0, \mathbf{z}_0$ and any initial matrix $\mathbf{B}_0$. Specifically, Part (a) of Theorem 1 shows that A-QNPE converges at a rate of $\mathcal{O}(1/k^2)$, matching the rate of NAG [23] that is known to be optimal in the regime where $k = \mathcal{O}(d)$ [28, 29]. Furthermore, Part (b) of Theorem 1 presents a convergence rate of $\mathcal{O}(\sqrt{d \log(k)}/k^{2.5})$. To see this, note that since we have $0 \preceq \mathbf{B}_0 \preceq L_1 \mathbf{I}$ and $0 \preceq \nabla^2 f(\mathbf{z}_0) \preceq L_1 \mathbf{I}$, in the worst case $\|\mathbf{B}_0 - \nabla^2 f(\mathbf{z}_0)\|_F^2$ in the expression of (19) can be upper bounded by $L_1^2 d$. Thus, assuming that $L_1$, $L_2$ and $\|\mathbf{z}_0 - \mathbf{x}^*\|$ are on the order of $\mathcal{O}(1)$, we have $M = \mathcal{O}(d)$ and the convergence rate in (18) can be simplified to $\mathcal{O}(\sqrt{d \log(k)}/k^{2.5})$. Notably, this rate surpasses the $\mathcal{O}(1/k^2)$ rate when $k = \Omega(d \log d)$. To the best of our knowledge, this is the first work to show a convergence rate faster than $\mathcal{O}(1/k^2)$ for a quasi-Newton-type method in the convex setting, thus establishing a provable advantage over NAG.

Table 1: The comparison of NAG and our proposed method in terms of computational cost.

| Methods | Gradient queires | Matrix-vector products |
|---|---|---|
| NAG | $\mathcal{O}(\epsilon^{-0.5})$ | N.A. |
| **A-QNPE (ours)** | $\tilde{\mathcal{O}}(\min\{\epsilon^{-0.5}, d^{0.2}\epsilon^{-0.4}\})$ | $\tilde{\mathcal{O}}(\min\{d^{0.25}\epsilon^{-0.5}, \epsilon^{-0.625}\})$ |

*Remark* 3 (Iteration complexity). Based on Theorem 1, we can find A-QNPE's iteration complexity. Define $N_\epsilon$ as the number of iterations required by A-QNPE to find an $\epsilon$-accurate solution, i.e., $f(\mathbf{x}) - f(\mathbf{x}^*) \leq \epsilon$. When $\sqrt{\epsilon} > \frac{1}{d \log d}$, the rate in (17) is better and we have $N_\epsilon = \mathcal{O}(\frac{1}{\epsilon^{0.5}})$. Conversely, when $\sqrt{\epsilon} < \frac{1}{d \log d}$, the rate in (18) is the better one, resulting in $N_\epsilon = \mathcal{O}((\frac{d}{\epsilon^2} \log \frac{d}{\epsilon^2})^{0.2})$. Hence, to achieve an $\epsilon$-accurate solution A-QNPE requires $\mathcal{O}(\min\{\frac{1}{\epsilon^{0.5}}, \frac{d^{0.2}}{\epsilon^{0.4}} (\log \frac{d}{\epsilon^2})^{0.2}\})$ iterations.

*Remark* 4 (Special case). If the initial point $\mathbf{z}_0$ is close to an optimal solution $\mathbf{x}^*$ and the initial Hessian approximation matrix $\mathbf{B}_0$ is chosen properly, the dependence on $d$ in the convergence rate of (18) can be eliminated. Specifically, if $\|\mathbf{z}_0 - \mathbf{x}^*\| = \mathcal{O}(\frac{1}{d})$ and we set $\mathbf{B}_0 = \nabla^2 f(\mathbf{z}_0)$, then we have $M = \mathcal{O}(1)$ and this leads to a local dimension-independent rate of $\mathcal{O}(\sqrt{\log k}/k^{2.5})$.

Recall that in each iteration of Algorithm 1, we need to execute a line search subroutine (Section 3.1) and a Hessian approximation update subroutine (Section 3.2). Thus, to fully characterize the computational cost of Algorithm 1, we need to upper bound the total number of gradient queries as well as the total number of matrix-vector product evaluations, which is the goal of Theorem 2.

**Theorem 2.** *Recall that $N_\epsilon$ denotes the minimum number of iterations required by Algorithm 1 to find an $\epsilon$-accurate solution according to Theorem 1. Then, with probability at least $1 - p$:*

(a) *The total number of gradient queries is bounded by $3N_\epsilon + \log_{1/\beta}(\frac{\sigma_0 L_1}{\alpha_2})$.*

(b) *The total number of matrix-vector product evaluations in the* LinearSolver *oracle is bounded by $N_\epsilon + C_{11}\sqrt{\sigma_0 L_1} + C_{12}\sqrt{\frac{L_1 \|\mathbf{z}_0 - \mathbf{x}^*\|^2}{2\epsilon}}$, where $C_{11}$ and $C_{12}$ are absolute constants.*

(c) *The total number of matrix-vector product evaluations in the* SEP *oracle is bounded by $\mathcal{O}(N_\epsilon^{1.25}(\log N_\epsilon)^{0.5} \log(\frac{\sqrt{d}N_\epsilon}{p}))$.*

If the initial step size is chosen as $\sigma_0 = \frac{\alpha_2}{L_1}$, Theorem 2(a) implies that A-QNPE requires no more than 3 gradient queries per iteration on average. Thus, the gradient oracle complexity of A-QNPE is the same as the iteration complexity, i.e., $\mathcal{O}(\min\{\frac{1}{\epsilon^{0.5}}, \frac{d^{0.2}}{\epsilon^{0.4}} (\log \frac{d}{\epsilon^2})^{0.2}\})$. On the other hand, the complexity in terms of matrix-vector products is worse. More precisely, by using the expression of $N_\epsilon$ in Remark 3, Parts (b) and (c) imply that the total number of matrix-vector product evaluations in the LinearSolver and SEP oracles can be bounded by $\mathcal{O}(\frac{1}{\epsilon^{0.5}})$ and $\tilde{\mathcal{O}}(\min\{\frac{d^{0.25}}{\epsilon^{0.5}}, \frac{1}{\epsilon^{0.625}}\})$, respectively.

For easier comparison, we summarize the detailed computational costs of NAG and our method A-QNPE to achieve an $\epsilon$-accuracy in Table 1. We observe that A-QNPE outperforms NAG in terms of gradient query complexity: It makes equal or fewer gradient queries especially when $\epsilon \ll \frac{1}{d^2}$. On the other hand, A-QNPE requires additional matrix-vector product computations to implement the LinearSolver and SEP oracles. While this is a limitation of our method, in some cases, gradient evaluations are the main bottleneck and can be more expensive than matrix-vector products. As a concrete example, consider the finite-sum minimization problem $f(\mathbf{x}) = \frac{1}{n}\sum_{i=1}^{n} f_i(\mathbf{x})$. In this case, one gradient query typically costs $\mathcal{O}(nd)$, while one matrix-vector product costs $\mathcal{O}(d^2)$. Thus, the total computational cost of NAG and A-QNPE can be bounded by $\mathcal{O}(\frac{nd}{\epsilon^{0.5}})$ and $\tilde{\mathcal{O}}(\frac{nd^{1.2}}{\epsilon^{0.4}} + \frac{d^{2.25}}{\epsilon^{0.5}})$, respectively. In particular, our method incurs a lower computational cost when $\epsilon \ll \frac{1}{d^2}$ and $n \gg d^{1.25}$.

# 5 Experiments

In this section, we compare the numerical performance of our proposed A-QNPE method with NAG and the classical BFGS quasi-Newton method. For fair comparison, we also use a line search

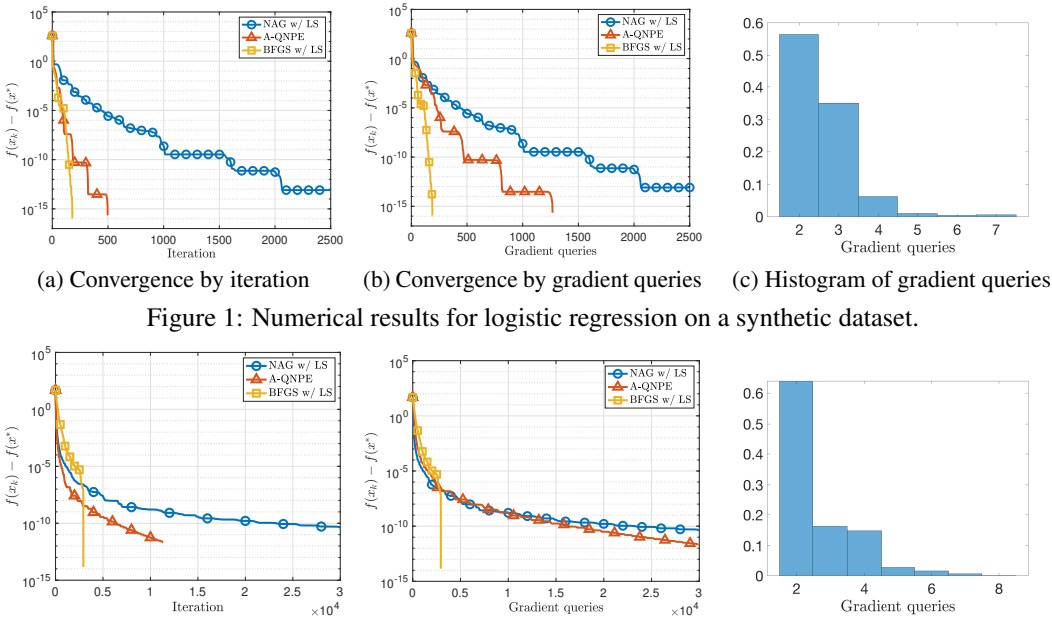

|  |  |  |
| :---: | :---: | :---: |
| (a) Convergence by iteration | (b) Convergence by gradient queries | (c) Histogram of gradient queries |

Figure 1: Numerical results for logistic regression on a synthetic dataset.

|  |  |  |
| :---: | :---: | :---: |
| (a) Convergence by iteration | (b) Convergence by gradient queries | (c) Histogram of gradient queries |

Figure 2: Numerical results for log-sum-exp function on a synthetic dataset.

scheme in NAG and BFGS to obtain their best performance [54, 15]. We would like to highlight that our paper mainly focuses on establishing a provable gain for quasi-Newton methods with respect to NAG, and our experimental results are presented to numerically verify our theoretical findings. In the first experiment, we focus on a logistic regression problem with the loss function $f(\mathbf{x}) = \frac{1}{n}\sum_{i=1}^{n}\log(1+e^{-y_i\langle \mathbf{a}_i, \mathbf{x}\rangle})$, where $\mathbf{a}_1, \ldots, \mathbf{a}_n \in \mathbb{R}^d$ are feature vectors and $\mathbf{y}_1, \ldots, \mathbf{y}_n \in \{-1, 1\}$ are binary labels. We perform our numerical experiments on a synthetic dataset and the data generation process is described in Appendix F. In the second experiment, we consider the log-sum-exp function $f(\mathbf{x}) = \log(\sum_{i=1}^{n} e^{\langle \mathbf{a}_i, \mathbf{x}\rangle - b_i})$, where we generate the dataset $\{(\mathbf{a}_i, b_i)\}_{i=1}^{n}$ following a similar procedure as in [16] (more details in Appendix F). As we observe in Fig. 1(a) and Fig. 2(a), our proposed A-QNPE method converges in much fewer iterations than NAG, while the best performance is achieved by BFGS. Due to the use of line search, we also compare these algorithms in terms of the total number of gradient queries. Moreover, additional plots in terms of the running time are included in Appendix F. As illustrated in Fig. 1(b) and Fig. 2(b), A-QNPE still outperforms NAG but the relative gain becomes less substantial. This is because the line search scheme in NAG only queries the function value at the new point, and thus it only requires one gradient per iteration. On the other hand, we should add that the number of gradient queries per iteration for A-QNPE is still small as guaranteed by our theory. In particular, the histogram of gradient queries in Fig. 1(c) and Fig. 2(c) shows that most of the iterations of A-QNPE require 2-3 gradient queries with an average of less than 3. Finally, although there is no theoretical guarantee showing a convergence gain for BFGS with respect to NAG, we observe that BFGS outperforms all the other considered methods in our experiments. Hence, studying the convergence behavior of BFGS (with line search) in the convex setting is an interesting research direction to explore.

## 6 Conclusions

We proposed a quasi-Newton variant of the accelerated proximal extragradient method for solving smooth convex optimization problems. We established two global convergence rates for our A-QNPE method, showing that it requires $\tilde{\mathcal{O}}(\min\{\frac{1}{\epsilon^{0.5}}, \frac{d^{0.2}}{\epsilon^{0.4}}\})$ gradient queries to find an $\epsilon$-accurate solution. In particular, in the regime where $\epsilon = \Omega(\frac{1}{d^2})$, A-QNPE achieves a gradient oracle complexity of $\mathcal{O}(\frac{1}{\epsilon^{0.5}})$, matching the complexity of NAG. Moreover, in the regime where $\epsilon = \tilde{\mathcal{O}}(\frac{1}{d^2})$, it outperforms NAG and improves the complexity to $\tilde{\mathcal{O}}(\frac{d^{0.2}}{\epsilon^{0.4}})$. To the best of our knowledge, this is the first result showing a provable gain for a quasi-Newton-type method over NAG in the convex setting.

## Acknowledgments and Disclosure of Funding

The research of R. Jiang and A. Mokhtari is supported in part by NSF Grants 2007668, 2019844, and 2112471, ARO Grant W911NF2110226, the Machine Learning Lab (MLL) at UT Austin, the Wireless Networking and Communications Group (WNCG) Industrial Affiliates Program, and the NSF AI Institute for Foundations of Machine Learning (IFML). The authors would also like to thank Yair Carmon and the anonymous reviewers for their comments on the first draft of the paper.

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

# Appendix

## A   Optimal Monteiro-Svaiter Acceleration Framework

In this section, we present some general results that hold for the optimal MS Acceleration framework. In particular, in the first part of this section (Section A.1), we present the proof of Proposition 1. In the second part (Section A.2), we further provide some useful additional lemmas.

### A.1   Proof of Proposition 1

To begin with, we establish a potential function for Algorithm 1, as shown in Proposition 2. The result is similar to Proposition 1 in [32], but for completeness we present its proof loosely following the strategy in [55, Theorem 5.3]. To simplify the notations, we use $f^*$ to denote the optimal $f(\mathbf{x}^*)$.

**Proposition 2.** *Consider the iterates generated by Algorithm 1. If $f$ is convex, then*

$$A_{k+1}(f(\mathbf{x}_{k+1}) - f^*) + \frac{1}{2}\|\mathbf{z}_{k+1} - \mathbf{x}^*\|^2 \leq A_k(f(\mathbf{x}_k) - f^*) + \frac{1}{2}\|\mathbf{z}_k - \mathbf{x}^*\|^2. \tag{20}$$

*Moreover, let $\sigma = \alpha_1 + \alpha_2$ and we have*

$$\sum_{k=0}^{N-1} \frac{a_k^2}{\eta_k^2}\|\hat{\mathbf{x}}_{k+1} - \mathbf{y}_k\|^2 \leq \frac{1}{1-\sigma^2}\|\mathbf{z}_0 - \mathbf{x}^*\|^2. \tag{21}$$

*Proof.* Since $f$ is convex, it holds that

$$f(\mathbf{x}_k) - f(\hat{\mathbf{x}}_{k+1}) - \langle \nabla f(\hat{\mathbf{x}}_{k+1}), \mathbf{x}_k - \hat{\mathbf{x}}_{k+1}\rangle \geq 0,$$
$$f(\mathbf{x}^*) - f(\hat{\mathbf{x}}_{k+1}) - \langle \nabla f(\hat{\mathbf{x}}_{k+1}), \mathbf{x}^* - \hat{\mathbf{x}}_{k+1}\rangle \geq 0.$$

By summing up the two inequalities with weights $a_k$ and $A_k$ respectively, we get

$$A_k(f(\mathbf{x}_k) - f^*) - (A_k + a_k)(f(\hat{\mathbf{x}}_{k+1}) - f^*) - a_k\langle \nabla f(\hat{\mathbf{x}}_{k+1}), \mathbf{x}^* - \hat{\mathbf{x}}_{k+1} - \frac{A_k}{a_k}(\hat{\mathbf{x}}_{k+1} - \mathbf{x}_k)\rangle \geq 0. \tag{22}$$

Let $\tilde{\mathbf{z}}_{k+1} = \hat{\mathbf{x}}_{k+1} + \frac{A_k}{a_k}(\hat{\mathbf{x}}_{k+1} - \mathbf{x}_k)$. By rearranging the terms, (22) can be rewritten as

$$(A_k + a_k)(f(\hat{\mathbf{x}}_{k+1}) - f^*) - A_k(f(\mathbf{x}_k) - f^*) \leq a_k\langle \nabla f(\hat{\mathbf{x}}_{k+1}), \tilde{\mathbf{z}}_{k+1} - \mathbf{x}^*\rangle. \tag{23}$$

Moreover, note that the update rule for $\mathbf{z}_{k+1}$ in both (8) and (9) can be written as

$$\mathbf{z}_{k+1} - \mathbf{z}_k = -\frac{\hat{\eta}_k}{\eta_k}a_k\nabla f(\hat{\mathbf{x}}_{k+1}). \tag{24}$$

Also, since we also have $\mathbf{z}_k = \mathbf{y}_k + \frac{A_k}{a_k}(\mathbf{y}_k - \mathbf{x}_k)$ from (2), we can write

$$\begin{aligned}
\tilde{\mathbf{z}}_{k+1} - \mathbf{z}_k &= \left[\hat{\mathbf{x}}_{k+1} + \frac{A_k}{a_k}(\hat{\mathbf{x}}_{k+1} - \mathbf{x}_k)\right] - \left[\mathbf{y}_k + \frac{A_k}{a_k}(\mathbf{y}_k - \mathbf{x}_k)\right] \\
&= \frac{A_k + a_k}{a_k}(\hat{\mathbf{x}}_{k+1} - \mathbf{y}_k) = \frac{a_k}{\eta_k}(\hat{\mathbf{x}}_{k+1} - \mathbf{y}_k),
\end{aligned} \tag{25}$$

where we used the fact that $(A_k + a_k)\eta_k = a_k^2$ in the last equality (cf. (2)). Hence, combining (24) and (25) leads to

$$\|\tilde{\mathbf{z}}_{k+1} - \mathbf{z}_{k+1}\| = \|\tilde{\mathbf{z}}_{k+1} - \mathbf{z}_k - (\mathbf{z}_{k+1} - \mathbf{z}_k)\| = \frac{a_k}{\eta_k}\|\hat{\mathbf{x}}_{k+1} - \mathbf{y}_k + \hat{\eta}_k\nabla f(\hat{\mathbf{x}}_{k+1})\| \leq \sigma\frac{a_k}{\eta_k}\|\hat{\mathbf{x}}_{k+1} - \mathbf{y}_k\|. \tag{26}$$

where we used (7) in the last inequality. In the following, we distinguish two cases depending on $\hat{\eta}_k = \eta_k$ or $\hat{\eta}_k < \eta_k$. In both cases, we shall prove that

$$A_{k+1}(f(\mathbf{x}_{k+1}) - f^*) + \frac{1}{2}\|\mathbf{z}_{k+1} - \mathbf{x}^*\|^2 \leq A_k(f(\mathbf{x}_k) - f^*) + \frac{1}{2}\|\mathbf{z}_k - \mathbf{x}^*\|^2 - \frac{(1-\sigma^2)a_k^2}{2\eta_k^2}\|\hat{\mathbf{x}}_{k+1} - \mathbf{y}_k\|^2. \tag{27}$$

If this is true, then Proposition 2 immediately follows. Indeed, since $\sigma < 1$, the last term in the right-hand side of (27) is negative, which implies (20). Moreover, (21) follows from summing the inequality in (27) from $k = 0$ to $N - 1$.

**Case I:** $\hat{\eta}_k = \eta_k$. Since by (8) we have $\mathbf{x}_{k+1} = \hat{\mathbf{x}}_{k+1}$ and $A_{k+1} = A_k + a_k$, (23) becomes

$$A_{k+1}(f(\mathbf{x}_{k+1}) - f^*) - A_k(f(\mathbf{x}_k) - f^*) \leq a_k \langle \nabla f(\mathbf{x}_{k+1}), \tilde{\mathbf{z}}_{k+1} - \mathbf{x}^* \rangle.$$

Using $\mathbf{z}_{k+1} = \mathbf{z}_k - a_k \nabla f(\mathbf{x}_{k+1})$ in (8), we have

$$\begin{aligned}
& A_{k+1}(f(\mathbf{x}_{k+1}) - f^*) - A_k(f(\mathbf{x}_k) - f^*) \\
& \leq \langle \mathbf{z}_k - \mathbf{z}_{k+1}, \tilde{\mathbf{z}}_{k+1} - \mathbf{x}^* \rangle \\
& = \langle \mathbf{z}_k - \mathbf{z}_{k+1}, \tilde{\mathbf{z}}_{k+1} - \mathbf{z}_{k+1} \rangle + \langle \mathbf{z}_k - \mathbf{z}_{k+1}, \mathbf{z}_{k+1} - \mathbf{x}^* \rangle \\
& = \frac{1}{2}\|\mathbf{z}_k - \mathbf{z}_{k+1}\|^2 + \frac{1}{2}\|\tilde{\mathbf{z}}_{k+1} - \mathbf{z}_{k+1}\|^2 - \frac{1}{2}\|\tilde{\mathbf{z}}_{k+1} - \mathbf{z}_k\|^2 \\
& \quad + \frac{1}{2}\|\mathbf{z}_k - \mathbf{x}^*\|^2 - \frac{1}{2}\|\mathbf{z}_{k+1} - \mathbf{x}^*\|^2 - \frac{1}{2}\|\mathbf{z}_k - \mathbf{z}_{k+1}\|^2 \\
& \leq \frac{1}{2}\|\mathbf{z}_k - \mathbf{x}^*\|^2 - \frac{1}{2}\|\mathbf{z}_{k+1} - \mathbf{x}^*\|^2 - \frac{(1 - \sigma^2)a_k^2}{2\eta_k^2}\|\mathbf{x}_{k+1} - \mathbf{y}_k\|^2,
\end{aligned} \tag{28}$$

where we used (25) and (26) in the last inequality. This immediately leads to (27) after rearranging the terms.

**Case II:** $\hat{\eta}_k < \eta_k$. Since $0 < \gamma_k < 1$ and $\mathbf{x}_{k+1} = \frac{(1-\gamma_k)A_k}{A_k + \gamma_k a_k}\mathbf{x}_k + \frac{\gamma_k(A_k + a_k)}{A_k + \gamma_k a_k}\hat{\mathbf{x}}_{k+1}$ according to (9), by Jensen's inequality we have $(A_k + \gamma_k a_k)f(\mathbf{x}_{k+1}) \leq \gamma_k(A_k + a_k)f(\hat{\mathbf{x}}_{k+1}) + (1 - \gamma_k)A_k f(\mathbf{x}_k)$, which further implies that

$$(A_k + \gamma_k a_k)(f(\mathbf{x}_{k+1}) - f^*) - A_k(f(\mathbf{x}_k) - f^*) \leq \gamma_k(A_k + a_k)(f(\hat{\mathbf{x}}_{k+1}) - f^*) - \gamma_k A_k(f(\mathbf{x}_k) - f^*).$$

Moreover, since $A_{k+1} = A_k + \gamma_k a_k$ by (9), together with (23) we obtain

$$A_{k+1}(f(\mathbf{x}_{k+1}) - f^*) - A_k(f(\mathbf{x}_k) - f^*) \leq \gamma_k a_k \langle \nabla f(\hat{\mathbf{x}}_{k+1}), \tilde{\mathbf{z}}_{k+1} - \mathbf{x}^* \rangle.$$

Using $\mathbf{z}_{k+1} = \mathbf{z}_k - \gamma_k a_k \nabla f(\hat{\mathbf{x}}_{k+1})$ in (9), we follow the same reasoning as in (28) to get:

$$\begin{aligned}
& A_{k+1}(f(\mathbf{x}_{k+1}) - f^*) - A_k(f(\mathbf{x}_k) - f^*) \\
& \leq \langle \mathbf{z}_k - \mathbf{z}_{k+1}, \tilde{\mathbf{z}}_{k+1} - \mathbf{x}^* \rangle \\
& = \langle \mathbf{z}_k - \mathbf{z}_{k+1}, \tilde{\mathbf{z}}_{k+1} - \mathbf{z}_{k+1} \rangle + \langle \mathbf{z}_k - \mathbf{z}_{k+1}, \mathbf{z}_{k+1} - \mathbf{x}^* \rangle \\
& = \frac{1}{2}\|\mathbf{z}_k - \mathbf{z}_{k+1}\|^2 + \frac{1}{2}\|\tilde{\mathbf{z}}_{k+1} - \mathbf{z}_{k+1}\|^2 - \frac{1}{2}\|\tilde{\mathbf{z}}_{k+1} - \mathbf{z}_k\|^2 \\
& \quad + \frac{1}{2}\|\mathbf{z}_k - \mathbf{x}^*\|^2 - \frac{1}{2}\|\mathbf{z}_{k+1} - \mathbf{x}^*\|^2 - \frac{1}{2}\|\mathbf{z}_k - \mathbf{z}_{k+1}\|^2 \\
& \leq \frac{1}{2}\|\mathbf{z}_k - \mathbf{x}^*\|^2 - \frac{1}{2}\|\mathbf{z}_{k+1} - \mathbf{x}^*\|^2 - \frac{(1 - \sigma^2)a_k^2}{2\eta_k^2}\|\hat{\mathbf{x}}_{k+1} - \mathbf{y}_k\|^2,
\end{aligned}$$

which also leads to (27). $\square$

Next, we prove a lower bound on $A_N$. Recall that $\mathcal{B}$ denotes the set of iteration indices where the line search scheme backtracks, i.e., $\mathcal{B} \triangleq \{k : \hat{\eta}_k < \eta_k\}$.

**Lemma 3.** *For any $N \geq 0$, it holds that*

$$A_N \geq \frac{1}{4}\left(\sqrt{\hat{\eta}_0} + \sum_{1 \leq k \leq N-1, k \notin \mathcal{B}} \sqrt{\hat{\eta}_k}\right)^2. \tag{29}$$

*Proof.* To begin with, according to the update rule of $A_{k+1}$ in (8) and (9) and the expression of $a_k$ in (2), the sequence $\{A_k\}$ follows the dynamic:

$$A_{k+1} = \begin{cases} A_k + a_k, & \text{if } \hat{\eta}_k = \eta_k \ (k \notin \mathcal{B}); \\ A_k + \gamma_k a_k, & \text{if } \hat{\eta}_k < \eta_k \ (k \in \mathcal{B}), \end{cases} \quad \text{where } \gamma_k = \frac{\hat{\eta}_k}{\eta_k} \text{ and } a_k = \frac{\eta_k + \sqrt{\eta_k^2 + 4\eta_k A_k}}{2}.$$

Since we initialize $A_0 = 0$, we have $a_0 = \eta_0$. We further have $A_1 = \hat{\eta}_0$, since we get $A_1 = A_0 + a_0 = \hat{\eta}_0$ if $0 \notin \mathcal{B}$, while we get $A_1 = A_0 + \gamma_0 a_0 = \frac{\hat{\eta}_0}{\eta_0}\eta_0 = \hat{\eta}_0$ if $0 \in \mathcal{B}$. Moreover:

- In **Case I** where $k \notin \mathcal{B}$, we have

$$A_{k+1} = A_k + a_k = A_k + \frac{\eta_k + \sqrt{\eta_k^2 + 4\eta_k A_k}}{2} \geq A_k + \frac{\eta_k}{2} + \sqrt{\eta_k A_k} \geq \left( \sqrt{A_k} + \frac{\sqrt{\eta_k}}{2} \right)^2,$$

which further implies that $\sqrt{A_{k+1}} \geq \sqrt{A_k} + \frac{\sqrt{\eta_k}}{2} = \sqrt{A_k} + \frac{\sqrt{\hat{\eta}_k}}{2}$.

- In **Case II** where $k \in \mathcal{B}$, we have $A_{k+1} = A_k + \gamma_k a_k \geq A_k$, which implies that $\sqrt{A_{k+1}} \geq \sqrt{A_k}$.

Considering the above, we obtain $\sqrt{A_N} \geq \sqrt{A_1} + \sum_{1 \leq k \leq N-1, k \notin \mathcal{B}} \frac{\sqrt{\hat{\eta}_k}}{2}$, which leads to (29). □

Lemma 3 provides a lower bound on $A_N$ in terms of the step sizes $\hat{\eta}_k$ in those iterations where the line search scheme does not backtrack, i.e., $k \notin \mathcal{B}$. The following lemma shows how we can further prove a lower bound in terms of all the step sizes $\{\hat{\eta}_k\}_{k=0}^{N-1}$.

**Lemma 4.** *We have*

$$\sum_{1 \leq k \leq N-1, k \in \mathcal{B}} \sqrt{\hat{\eta}_k} \leq \frac{1}{1 - \sqrt{\beta}} \left( \sqrt{\hat{\eta}_0} + \sum_{1 \leq k \leq N-1, k \notin \mathcal{B}} \sqrt{\hat{\eta}_k} \right). \tag{30}$$

*As a corollary, we have*

$$\sqrt{\hat{\eta}_0} + \sum_{1 \leq k \leq N-1, k \notin \mathcal{B}} \sqrt{\hat{\eta}_k} \geq \frac{1 - \sqrt{\beta}}{2 - \sqrt{\beta}} \sum_{k=0}^{N-1} \sqrt{\hat{\eta}_k}. \tag{31}$$

*Proof.* When the line search scheme backtracks, i.e., $k \in \mathcal{B}$, we have $\hat{\eta}_k \leq \beta \eta_k$. Therefore,

$$\sum_{1 \leq k \leq N-1, k \in \mathcal{B}} \sqrt{\hat{\eta}_k} \leq \sum_{1 \leq k \leq N-1, k \in \mathcal{B}} \sqrt{\beta \eta_k} \leq \sum_{k=1}^{N-1} \sqrt{\beta \eta_k} = \sqrt{\beta \eta_1} + \sum_{k=1}^{N-2} \sqrt{\beta \eta_{k+1}}. \tag{32}$$

Moreover, in the update of Algorithm 1, we have $\eta_{k+1} = \hat{\eta}_k/\beta$ if $k \notin \mathcal{B}$ (cf. Line 8) and $\eta_{k+1} = \hat{\eta}_k$ otherwise (cf. Line 13). This implies that $\eta_1 \leq \hat{\eta}_0/\beta$ and we further have

$$\sqrt{\beta \eta_1} + \sum_{k=1}^{N-2} \sqrt{\beta \eta_{k+1}} = \sqrt{\beta \eta_1} + \sum_{1 \leq k \leq N-2, k \notin \mathcal{B}} \sqrt{\beta \eta_{k+1}} + \sum_{1 \leq k \leq N-2, k \in \mathcal{B}} \sqrt{\beta \eta_{k+1}}$$

$$\leq \sqrt{\hat{\eta}_0} + \sum_{1 \leq k \leq N-2, k \notin \mathcal{B}} \sqrt{\hat{\eta}_k} + \sum_{1 \leq k \leq N-2, k \in \mathcal{B}} \sqrt{\beta \hat{\eta}_k}$$

$$\leq \sqrt{\hat{\eta}_0} + \sum_{1 \leq k \leq N-1, k \notin \mathcal{B}} \sqrt{\hat{\eta}_k} + \sum_{1 \leq k \leq N-1, k \in \mathcal{B}} \sqrt{\beta \hat{\eta}_k}. \tag{33}$$

We combine (32) and (33) to get

$$\sum_{1 \leq k \leq N-1, k \in \mathcal{B}} \sqrt{\hat{\eta}_k} \leq \sqrt{\hat{\eta}_0} + \sum_{1 \leq k \leq N-1, k \notin \mathcal{B}} \sqrt{\hat{\eta}_k} + \sum_{1 \leq k \leq N-1, k \in \mathcal{B}} \sqrt{\beta \hat{\eta}_k}.$$

By rearranging the terms and simple algebraic manipulation, we obtain (30) as desired. Finally, (31) follows by adding $\sqrt{\hat{\eta}_0} + \sum_{1 \leq k \leq N-1, k \notin \mathcal{B}} \sqrt{\hat{\eta}_k}$ to both sides of (30). □

Now we are ready to prove Proposition 1.

*Proof of Proposition 1.* By Proposition 2, the potential function $\phi_k \triangleq A_k(f(\mathbf{x}_k) - f^*) + \frac{1}{2} \|\mathbf{z}_k - \mathbf{x}^*\|^2$ is non-increasing in each iteration. Hence, via a recursive augment we have $A_N(f(\mathbf{x}_N) - f^*) \leq \phi_N \leq \cdots \leq \phi_0 = \frac{1}{2} \|\mathbf{z}_0 - \mathbf{x}^*\|^2$, which yields $f(\mathbf{x}_N) - f^* \leq \frac{\|\mathbf{z}_0 - \mathbf{x}^*\|^2}{2A_N}$. Moreover, combining Lemma 3 and (31) in Lemma 4 leads to the second inequality in Proposition 1. □

## A.2 Additional Supporting Lemmas

A crucial part of our analysis is to bound the path length of the sequence $\{\mathbf{y}_k\}_{k=0}^N$. This is done in Lemma 8. To achieve this goal we first present the results in Lemmas 5-7, which provide the required ingredients for proving the claim in Lemma 8. In our first intermediate result, we establish uniform upper bounds for the error terms $\|\mathbf{z}_k - \mathbf{x}^*\|$ and $\|\mathbf{x}_k - \mathbf{x}^*\|$.

**Lemma 5.** *Recall that $\sigma = \alpha_1 + \alpha_2$. For all $k \geq 0$, we have $\|\mathbf{z}_k - \mathbf{x}^*\| \leq \|\mathbf{z}_0 - \mathbf{x}^*\|$ and $\|\mathbf{x}_k - \mathbf{x}^*\| \leq \sqrt{\frac{2}{1-\sigma^2}} \|\mathbf{z}_0 - \mathbf{x}^*\|$.*

*Proof.* To begin with, it follows from (20) in Proposition 2 that

$$\frac{1}{2}\|\mathbf{z}_k - \mathbf{x}^*\|^2 \leq A_k(f(\mathbf{x}_k) - f^*) + \frac{1}{2}\|\mathbf{z}_k - \mathbf{x}^*\|^2 \leq A_0(f(\mathbf{x}_0) - f^*) + \frac{1}{2}\|\mathbf{z}_0 - \mathbf{x}^*\|^2 = \frac{1}{2}\|\mathbf{z}_0 - \mathbf{x}^*\|^2.$$

Hence, we get $\|\mathbf{z}_k - \mathbf{x}^*\| \leq \|\mathbf{z}_0 - \mathbf{x}^*\|$ for any $k \geq 0$. To show the second inequality, we distinguish two cases and in both cases we will prove that

$$A_{k+1}\|\mathbf{x}_{k+1} - \mathbf{x}^*\|^2 \leq A_k\|\mathbf{x}_k - \mathbf{x}^*\|^2 + (A_{k+1} - A_k)\frac{2\sigma^2 a_k^2}{\eta_k^2}\|\hat{\mathbf{x}}_{k+1} - \mathbf{y}_k\|^2 + 2(A_{k+1} - A_k)\|\mathbf{z}_{k+1} - \mathbf{x}^*\|^2. \tag{34}$$

**Case I**: $\hat{\eta}_k = \eta_k$. Recall that in the proof of Proposition 2 we defined $\tilde{\mathbf{z}}_{k+1} = \hat{\mathbf{x}}_{k+1} + \frac{A_k}{a_k}(\hat{\mathbf{x}}_{k+1} - \mathbf{x}_k)$.

Since $\mathbf{x}_{k+1} = \hat{\mathbf{x}}_{k+1}$, we have $\mathbf{x}_{k+1} = \frac{A_k}{A_k + a_k}\mathbf{x}_k + \frac{a_k}{A_k + a_k}\tilde{\mathbf{z}}_{k+1}$ and by Jensen's inequality

$$\|\mathbf{x}_{k+1} - \mathbf{x}^*\|^2 \leq \frac{A_k}{A_k + a_k}\|\mathbf{x}_k - \mathbf{x}^*\|^2 + \frac{a_k}{A_k + a_k}\|\tilde{\mathbf{z}}_{k+1} - \mathbf{x}^*\|^2.$$

Furthermore, we have

$$\|\tilde{\mathbf{z}}_{k+1} - \mathbf{x}^*\|^2 \leq 2\|\tilde{\mathbf{z}}_{k+1} - \mathbf{z}_{k+1}\|^2 + 2\|\mathbf{z}_{k+1} - \mathbf{x}^*\|^2 \leq \frac{2\sigma^2 a_k^2}{\eta_k^2}\|\hat{\mathbf{x}}_{k+1} - \mathbf{y}_k\|^2 + 2\|\mathbf{z}_{k+1} - \mathbf{x}^*\|^2, \tag{35}$$

where we used (26) in the last inequality. By combining the above two inequalities, we obtain

$$(A_k + a_k)\|\mathbf{x}_{k+1} - \mathbf{x}^*\|^2 \leq A_k\|\mathbf{x}_k - \mathbf{x}^*\|^2 + a_k\frac{2\sigma^2 a_k^2}{\eta_k^2}\|\hat{\mathbf{x}}_{k+1} - \mathbf{y}_k\|^2 + 2a_k\|\mathbf{z}_{k+1} - \mathbf{x}^*\|^2,$$

which leads to (34) (note that $A_{k+1} = A_k + a_k$ in **Case I**).

**Case II**: Since $\mathbf{x}_{k+1} = \frac{(1-\gamma_k)A_k}{A_k + \gamma_k a_k}\mathbf{x}_k + \frac{\gamma_k(A_k + a_k)}{A_k + \gamma_k a_k}\hat{\mathbf{x}}_{k+1}$ and $\hat{\mathbf{x}}_{k+1} = \frac{A_k}{A_k + a_k}\mathbf{x}_k + \frac{a_k}{A_k + a_k}\tilde{\mathbf{z}}_{k+1}$, we have

$$\mathbf{x}_{k+1} = \frac{A_k}{A_k + \gamma_k a_k}\mathbf{x}_k + \frac{\gamma_k a_k}{A_k + \gamma_k a_k}\tilde{\mathbf{z}}_{k+1}.$$

Similarly, by Jensen's inequality we have

$$(A_k + \gamma_k a_k)\|\mathbf{x}_{k+1} - \mathbf{x}^*\|^2 \leq A_k\|\mathbf{x}_k - \mathbf{x}^*\|^2 + \gamma_k a_k\|\tilde{\mathbf{z}}_{k+1} - \mathbf{x}^*\|^2.$$

Combining this inequality with (35), we obtain

$$(A_k + \gamma_k a_k)\|\mathbf{x}_{k+1} - \mathbf{x}^*\|^2 \leq A_k\|\mathbf{x}_k - \mathbf{x}^*\|^2 + \gamma_k a_k\frac{2\sigma^2 a_k^2}{\eta_k^2}\|\hat{\mathbf{x}}_{k+1} - \mathbf{y}_k\|^2 + 2\gamma_k a_k\|\mathbf{z}_{k+1} - \mathbf{x}^*\|^2. \tag{36}$$

which leads to (34) (note that $A_{k+1} = A_k + \gamma_k a_k$ in **Case II**).

Now by summing (34) over $k = 0, \ldots, N-1$, we get

$$A_N\|\mathbf{x}_N - \mathbf{x}^*\|^2 \leq \sum_{k=0}^{N-1}(A_{k+1} - A_k)\frac{2\sigma^2 a_k^2}{\eta_k^2}\|\hat{\mathbf{x}}_{k+1} - \mathbf{y}_k\|^2 + \sum_{k=0}^{N-1}2(A_{k+1} - A_k)\|\mathbf{z}_{k+1} - \mathbf{x}^*\|^2 \tag{37}$$

$$\leq 2\sigma^2\sum_{k=0}^{N-1}(A_{k+1} - A_k)\sum_{k=0}^{N-1}\frac{a_k^2}{\eta_k^2}\|\hat{\mathbf{x}}_{k+1} - \mathbf{y}_k\|^2 + 2\|\mathbf{z}_0 - \mathbf{x}^*\|^2\sum_{k=0}^{N-1}(A_{k+1} - A_k) \tag{38}$$

$$\leq \frac{2\sigma^2}{1-\sigma^2}A_N\|\mathbf{z}_0 - \mathbf{x}^*\|^2 + 2A_N\|\mathbf{z}_0 - \mathbf{x}^*\|^2 \tag{39}$$

$$= \frac{2A_N}{1-\sigma^2}\|\mathbf{z}_0 - \mathbf{x}^*\|^2. \tag{40}$$

Hence, this implies that $\|\mathbf{x}_k - \mathbf{x}^*\|^2 \leq \frac{2}{1-\sigma^2}\|\mathbf{z}_0 - \mathbf{x}^*\|^2$ for any $k \geq 0$. $\qquad\square$

A key term appearing in several of our bounds is $\frac{a_{k+1}}{A_{k+1}+a_{k+1}}$. In the next lemma, we establish an upper bound for this ratio based on a factor of its previous value, for both cases of our algorithm.

**Lemma 6.** *Without loss of generality assume $\beta > 1/5$. In **Case I** we have $\frac{a_{k+1}}{A_{k+1}+a_{k+1}} \leq \frac{1}{\sqrt{\beta}}\frac{a_k}{A_k+a_k}$. Otherwise, in **Case II** we have $\frac{a_{k+1}}{A_{k+1}+a_{k+1}} \leq \frac{2\sqrt{\beta}}{\sqrt{\beta}+1}\frac{a_k}{A_k+a_k}$.*

*Proof.* By the choice of $a_k$ in (2) we have $\eta_k(A_k + a_k) = a_k^2$ for all $k \geq 0$. As a result, we have

$$\frac{a_k}{A_k+a_k} = \frac{\eta_k}{a_k} = \frac{2\eta_k}{\eta_k + \sqrt{\eta_k^2 + 4\eta_k A_k}} = \frac{2}{1 + \sqrt{1 + 4\frac{A_k}{\eta_k}}},$$

and similarly

$$\frac{a_{k+1}}{A_{k+1}+a_{k+1}} = \frac{2}{1 + \sqrt{1 + 4\frac{A_{k+1}}{\eta_{k+1}}}}.$$

In **Case I**, we have $\eta_{k+1} = \eta_k/\beta$ and $A_{k+1} \geq A_k$. Hence, it implies that $A_{k+1}/\eta_{k+1} \geq \beta A_k/\eta_k$, which leads to

$$\frac{a_{k+1}}{A_{k+1}+a_{k+1}} \leq \frac{2}{1 + \sqrt{1 + \frac{4\beta A_k}{\eta_k}}} \leq \frac{2}{\sqrt{\beta} + \sqrt{\beta + \frac{4\beta A_k}{\eta_k}}} = \frac{1}{\sqrt{\beta}}\frac{2}{1 + \sqrt{1 + 4\frac{A_k}{\eta_k}}} = \frac{1}{\sqrt{\beta}}\frac{a_k}{A_k+a_k}.$$

where the second inequality follows from the fact that $\beta \leq 1$.

In **Case II**, we have $\eta_{k+1} = \hat{\eta}_k = \gamma_k\eta_k$ and $A_{k+1} = A_k + \gamma_k a_k$. Since we also have $a_k \geq \eta_k$ and $\gamma_k \leq \beta$, we obtain $A_{k+1}/\eta_{k+1} \geq A_k/(\gamma_k\eta_k) + 1 \geq A_k/(\beta\eta_k) + 1$. Hence,

$$\frac{a_{k+1}}{A_{k+1}+a_{k+1}} \leq \frac{2}{1 + \sqrt{5 + \frac{4A_k}{\beta\eta_k}}} \leq \frac{2}{1 + \frac{1}{\sqrt{\beta}}\sqrt{1 + \frac{4A_k}{\eta_k}}} \leq \frac{2\sqrt{\beta}}{\sqrt{\beta}+1}\frac{2}{1 + \sqrt{1 + \frac{4A_k}{\eta_k}}} = \frac{2\sqrt{\beta}}{\sqrt{\beta}+1}\frac{a_k}{A_k+a_k},$$

where we used $\beta > 1/5$ in the second inequality and the fact that $1 + \frac{1}{\sqrt{\beta}}x \geq \frac{\sqrt{\beta}+1}{2\sqrt{\beta}}(1 + x)$ for $x \geq 1$ in the last inequality. $\qquad\square$

*Remark 5.* If $\beta \leq 1/5$, then in **Case I** we still have $\frac{a_{k+1}}{A_{k+1}+a_{k+1}} \leq \frac{1}{\sqrt{\beta}}\frac{a_k}{A_k+a_k}$, while in **Case II** we have $\frac{a_{k+1}}{A_{k+1}+a_{k+1}} \leq \frac{2}{\sqrt{5}+1}\frac{a_k}{A_k+a_k}$. Thus, in the case where $\beta \leq 1/5$, the derivation below still holds except that the absolute constant $C_2$ will be different.

Next, as a corollary of Lemma 6, we establish an upper bound on the series $\sum_{k=0}^{N-1}\frac{a_k}{A_k+a_k}$. Moreover, we use this result to establish an upper bound for $\sum_{k=0}^{N-1}\|\hat{\mathbf{x}}_{k+1} - \mathbf{y}_k\|$.

**Lemma 7.** *We have*

$$\sum_{k=0}^{N-1}\frac{a_k}{A_k+a_k} \leq \frac{1 + 2\sqrt{\beta} - \beta}{\sqrt{\beta} - \beta}\left(1 + \log\frac{A_N}{A_1}\right). \tag{41}$$

*Moreover,*

$$\sum_{k=0}^{N-1}\|\hat{\mathbf{x}}_{k+1} - \mathbf{y}_k\| \leq \sqrt{\frac{1}{1-\sigma^2}\frac{1 + 2\sqrt{\beta} - \beta}{\sqrt{\beta} - \beta}\left(1 + \log\frac{A_N}{A_1}\right)}\|\mathbf{z}_0 - \mathbf{x}^*\|. \tag{42}$$

*Proof.* Given the initial values of $A_k$ and $a_k$ we have

$$\sum_{k=0}^{N-1}\frac{a_k}{A_k+a_k} = 1 + \sum_{k=1}^{N-1}\frac{a_k}{A_k+a_k} = 1 + \sum_{k\in\mathcal{B}, k\geq 1}\frac{a_k}{A_k+a_k} + \sum_{k\notin\mathcal{B}, k\geq 1}\frac{a_k}{A_k+a_k} \tag{43}$$

Note that using the result in Lemma 6

$$\sum_{k\in\mathcal{B},k\geq 1}\frac{a_k}{A_k+a_k} \leq \sum_{k=0}^{N-2}\frac{a_{k+1}}{A_{k+1}+a_{k+1}} \tag{44}$$

$$= \sum_{k\notin\mathcal{B},k\geq 0}\frac{a_{k+1}}{A_{k+1}+a_{k+1}} + \sum_{k\in\mathcal{B},k\geq 0}\frac{a_{k+1}}{A_{k+1}+a_{k+1}} \tag{45}$$

$$\leq \sum_{k\notin\mathcal{B},k\geq 0}\frac{1}{\sqrt{\beta}}\frac{a_k}{A_k+a_k} + \sum_{k\in\mathcal{B},k\geq 0}\frac{2\sqrt{\beta}}{\sqrt{\beta}+1}\frac{a_k}{A_k+a_k} \tag{46}$$

$$\leq \frac{1}{\sqrt{\beta}} + \sum_{k\notin\mathcal{B},k\geq 1}\frac{1}{\sqrt{\beta}}\frac{a_k}{A_k+a_k} + \sum_{k\in\mathcal{B},k\geq 1}\frac{2\sqrt{\beta}}{\sqrt{\beta}+1}\frac{a_k}{A_k+a_k}. \tag{47}$$

Hence, if we move the last term in the above upper bound to the left hand side and rescale both sides of the resulted inequality we obtain

$$\sum_{k\in\mathcal{B},k\geq 1}\frac{a_k}{A_k+a_k} \leq \frac{1+\sqrt{\beta}}{\sqrt{\beta}-\beta}\left(1+\sum_{k\notin\mathcal{B},k\geq 1}\frac{a_k}{A_k+a_k}\right).$$

Now, if we replace the above upper bound into (43) we obtain

$$\sum_{k=0}^{N-1}\frac{a_k}{A_k+a_k} \leq \frac{1+2\sqrt{\beta}-\beta}{\sqrt{\beta}-\beta}\left(1+\sum_{k\notin\mathcal{B},k\geq 1}\frac{a_k}{A_k+a_k}\right). \tag{48}$$

Moreover, note that for $k\notin\mathcal{B}$, we have $A_{k+1}=A_k+a_k$. Hence,

$$\sum_{k\notin\mathcal{B},k\geq 1}\frac{a_k}{A_k+a_k} = \sum_{k\notin\mathcal{B},k\geq 1}\left(1-\frac{A_k}{A_{k+1}}\right) \leq \sum_{k\notin\mathcal{B},k\geq 1}\left(\log(A_{k+1})-\log(A_k)\right)$$

$$\leq \sum_{k=1}^{N-1}\left(\log(A_{k+1})-\log(A_k)\right) = \log\frac{A_N}{A_1}.$$

Now if we replace the above upper bound, i.e., $\log\frac{A_N}{A_1}$ with $\sum_{k\notin\mathcal{B},k\geq 1}\frac{a_k}{A_k+a_k}$ into the expression in the right-hand side of (48) we obtain the result in (41).

Next, note that by Cauchy-Schwarz inequality, we have

$$\sum_{k=0}^{N-1}\|\hat{\mathbf{x}}_{k+1}-\mathbf{y}_k\| \leq \sqrt{\sum_{k=0}^{N-1}\frac{\eta_k^2}{a_k^2}\sum_{k=0}^{N-1}\frac{a_k^2}{\eta_k^2}\|\hat{\mathbf{x}}_{k+1}-\mathbf{y}_k\|^2} \leq \sqrt{\frac{1}{1-\sigma^2}\sum_{k=0}^{N-1}\frac{\eta_k^2}{a_k^2}}\|\mathbf{z}_0-\mathbf{x}^*\|,$$

where the last inequality follows from (21). Moreover, based on the expression for $a_k$ in (2) and the result in (41) that we just proved, we have

$$\sum_{k=0}^{N-1}\frac{\eta_k^2}{a_k^2} = \sum_{k=0}^{N-1}\frac{a_k^2}{(A_k+a_k)^2} \leq \sum_{k=0}^{N-1}\frac{a_k}{A_k+a_k} \leq \frac{1+2\sqrt{\beta}-\beta}{\sqrt{\beta}-\beta}\left(1+\log\frac{A_N}{A_1}\right).$$

Combining the two inequalities above leads to (42). $\qquad\square$

Now we are ready to present and prove Lemma 8 which characterizes a bound on the path length of the sequence $\{\mathbf{y}_k\}_{k=0}^N$

**Lemma 8.** *Consider the iterates generated by Algorithm 1. Then for any $N$,*

$$\sum_{k=0}^{N-1}\|\mathbf{y}_{k+1}-\mathbf{y}_k\| \leq C_2\left(1+\log\frac{A_N}{A_1}\right)\|\mathbf{z}_0-\mathbf{x}^*\|.$$

*where*

$$C_2 = 2\sqrt{\frac{1}{1-\sigma^2}\frac{1+2\sqrt{\beta}-\beta}{\sqrt{\beta}-\beta}} + \frac{1}{\sqrt{\beta}}\left(1+\sqrt{\frac{2}{1-\sigma^2}}\right)\frac{1+2\sqrt{\beta}-\beta}{\sqrt{\beta}-\beta} \tag{49}$$

*Proof.* By the triangle inequality, we have

$$\|\mathbf{y}_k - \mathbf{y}_{k+1}\| \le \|\hat{\mathbf{x}}_{k+1} - \mathbf{y}_k\| + \|\hat{\mathbf{x}}_{k+1} - \mathbf{y}_{k+1}\|. \tag{50}$$

We again distinguish two cases.

**Case I:** $\hat{\eta}_k = \eta_k$. In this case $\hat{\mathbf{x}}_{k+1} = \mathbf{x}_{k+1}$ and $\mathbf{y}_{k+1} = \frac{A_{k+1}}{A_{k+1}+a_{k+1}}\mathbf{x}_{k+1} + \frac{a_{k+1}}{A_{k+1}+a_{k+1}}\mathbf{z}_{k+1}$, hence

$$\|\hat{\mathbf{x}}_{k+1} - \mathbf{y}_{k+1}\| = \|\mathbf{x}_{k+1} - \mathbf{y}_{k+1}\| = \frac{a_{k+1}\|\mathbf{z}_{k+1} - \mathbf{x}_{k+1}\|}{A_{k+1}+a_{k+1}} \le \frac{1}{\sqrt{\beta}}\left(1 + \sqrt{\frac{2}{1-\sigma^2}}\right)\frac{a_k\|\mathbf{z}_0 - \mathbf{x}^*\|}{A_k+a_k},$$

where we used Lemma 6 and the fact that $\|\mathbf{z}_{k+1} - \mathbf{x}_{k+1}\| \le \|\mathbf{z}_{k+1} - \mathbf{x}^*\| + \|\mathbf{x}_{k+1} - \mathbf{x}^*\| \le (1 + \sqrt{\frac{2}{1-\sigma^2}})\|\mathbf{z}_0 - \mathbf{x}^*\|$ in the last inequality. Therefore, using (50) and the above bound we have

$$\|\mathbf{y}_k - \mathbf{y}_{k+1}\| \le \|\hat{\mathbf{x}}_{k+1} - \mathbf{y}_k\| + \frac{1}{\sqrt{\beta}}\left(1 + \sqrt{\frac{2}{1-\sigma^2}}\right)\frac{a_k}{A_k+a_k}\|\mathbf{z}_0 - \mathbf{x}^*\|. \tag{51}$$

**Case II:** $\hat{\eta}_k < \eta_k$. Since $\mathbf{x}_{k+1} = \frac{A_k}{A_k+\gamma_k a_k}\mathbf{x}_k + \frac{\gamma_k a_k}{A_k+\gamma_k a_k}\tilde{\mathbf{z}}_{k+1}$ and $\hat{\mathbf{x}}_{k+1} = \frac{A_k}{A_k+a_k}\mathbf{x}_k + \frac{a_k}{A_k+a_k}\tilde{\mathbf{z}}_{k+1}$, we get

$$\hat{\mathbf{x}}_{k+1} = \frac{A_k}{A_k+a_k}\left(\mathbf{x}_{k+1} + \frac{\gamma_k a_k}{A_k}(\mathbf{x}_{k+1} - \tilde{\mathbf{z}}_{k+1})\right) + \frac{a_k}{A_k+a_k}\tilde{\mathbf{z}}_{k+1} = \frac{A_k+\gamma_k a_k}{A_k+a_k}\mathbf{x}_{k+1} + \frac{(1-\gamma_k)a_k}{A_k+a_k}\tilde{\mathbf{z}}_{k+1}.$$

Thus, given the above equality and the expression $\mathbf{y}_{k+1} = \frac{A_{k+1}}{A_{k+1}+a_{k+1}}\mathbf{x}_{k+1} + \frac{a_{k+1}}{A_{k+1}+a_{k+1}}\mathbf{z}_{k+1}$, we have

$$\|\hat{\mathbf{x}}_{k+1} - \mathbf{y}_{k+1}\| \le \frac{(1-\gamma_k)a_k}{A_k+a_k}\|\tilde{\mathbf{z}}_{k+1} - \mathbf{z}_{k+1}\| + \left|\frac{(1-\gamma_k)a_k}{A_k+a_k} - \frac{a_{k+1}}{A_{k+1}+a_{k+1}}\right|\|\mathbf{z}_{k+1} - \mathbf{x}_{k+1}\|. \tag{52}$$

Moreover, based on the result in (26), we can upper bound $\|\tilde{\mathbf{z}}_{k+1} - \mathbf{z}_{k+1}\|$ by $\sigma\frac{a_k}{\eta_k}\|\hat{\mathbf{x}}_{k+1} - \mathbf{y}_k\|$ which implies that

$$\frac{(1-\gamma_k)a_k}{A_k+a_k}\|\tilde{\mathbf{z}}_{k+1} - \mathbf{z}_{k+1}\| \le \sigma\frac{(1-\gamma_k)a_k^2}{\eta_k(A_k+a_k)}\|\hat{\mathbf{x}}_{k+1} - \mathbf{y}_k\| = \sigma(1-\gamma_k)\|\hat{\mathbf{x}}_{k+1} - \mathbf{y}_k\| \le \|\hat{\mathbf{x}}_{k+1} - \mathbf{y}_k\|$$

where the equality holds due to the definition of $a_k$, and the last inequality holds as both $\gamma_k$ and $\sigma$ are in $(0, 1)$. On the other hand, note that

$$\frac{(1-\gamma_k)a_k}{A_k+a_k} - \frac{a_{k+1}}{A_{k+1}+a_{k+1}} \le \frac{(1-\gamma_k)a_k}{A_k+a_k} \le \frac{a_k}{A_k+a_k}, \tag{53}$$

$$\frac{a_{k+1}}{A_{k+1}+a_{k+1}} - \frac{(1-\gamma_k)a_k}{A_k+a_k} \le \frac{2\sqrt{\beta}a_k}{\sqrt{\beta+1}(A_k+a_k)} - \frac{(1-\gamma_k)a_k}{A_k+a_k} \le \frac{a_k}{A_k+a_k}. \tag{54}$$

where in the second bound we used the result in Lemma 6 and the fact that $\frac{2sqrt\beta}{\sqrt{\beta+1}} < 1$. Hence, we get

$$\|\hat{\mathbf{x}}_{k+1} - \mathbf{y}_{k+1}\| \le \|\hat{\mathbf{x}}_{k+1} - \mathbf{y}_k\| + \frac{a_k\|\mathbf{z}_{k+1} - \mathbf{x}_{k+1}\|}{A_k+a_k} \le \|\hat{\mathbf{x}}_{k+1} - \mathbf{y}_k\| + \left(1 + \sqrt{\frac{2}{1-\sigma^2}}\right)\frac{a_k\|\mathbf{z}_0 - \mathbf{x}^*\|}{A_k+a_k},$$

where the last inequality follows from the fact $\|\mathbf{z}_{k+1} - \mathbf{x}_{k+1}\| \le \|\mathbf{z}_{k+1} - \mathbf{x}^*\| + \|\mathbf{x}_{k+1} - \mathbf{x}^*\|$ and the bounds in Lemma 5. Now by applying the above upper bound into (50) we obtain that

$$\|\mathbf{y}_k - \mathbf{y}_{k+1}\| \le 2\|\hat{\mathbf{x}}_{k+1} - \mathbf{y}_k\| + \left(1 + \sqrt{\frac{2}{1-\sigma^2}}\right)\frac{a_k}{A_k+a_k}\|\mathbf{z}_0 - \mathbf{x}^*\|. \tag{55}$$

Considering the upper bounds established for $\|\mathbf{y}_k - \mathbf{y}_{k+1}\|$ in case I (equation (51)) and case II (equation (55)), we can conclude that

$$\|\mathbf{y}_k - \mathbf{y}_{k+1}\| \le 2\|\hat{\mathbf{x}}_{k+1} - \mathbf{y}_k\| + \frac{1}{\sqrt{\beta}}\left(1 + \sqrt{\frac{2}{1-\sigma^2}}\right)\frac{a_k}{A_k+a_k}\|\mathbf{z}_0 - \mathbf{x}^*\|. \tag{56}$$

Finally, Lemma 8 follows from summing (56) over $k = 0$ to $N - 1$ and the result of Lemma 7. $\quad\square$

---

**Subroutine 1** Backtracking line search

---

1: **Input:** iterate $\mathbf{y} \in \mathbb{R}^d$, gradient $\mathbf{g} \in \mathbb{R}^d$, Hessian approximation $\mathbf{B} \in \mathbb{S}_+^d$, initial trial step size $\eta > 0$
2: **Parameters:** line search parameters $\beta \in (0,1)$, $\alpha_1 \geq 0$ and $\alpha_2 > 0$ such that $\alpha_1 + \alpha_2 < 1$
3: Set $\hat{\eta} \leftarrow \eta$
4: Compute $\mathbf{s}_+ \leftarrow \mathsf{LinearSolver}(\mathbf{I} + \hat{\eta}\mathbf{B}, -\hat{\eta}\mathbf{g}; \alpha_1)$ and $\hat{\mathbf{x}}_+ \leftarrow \mathbf{y} + \mathbf{s}_+$
5: **while** $\|\hat{\mathbf{x}}_+ - \mathbf{y} + \hat{\eta}\nabla f(\hat{\mathbf{x}}_+)\|_2 > (\alpha_1 + \alpha_2)\|\hat{\mathbf{x}}_+ - \mathbf{y}\|_2$ **do**
6:     Set $\tilde{\mathbf{x}}_+ \leftarrow \hat{\mathbf{x}}_+$ and $\hat{\eta} \leftarrow \beta\hat{\eta}$
7:     Compute $\mathbf{s}_+ \leftarrow \mathsf{LinearSolver}(\mathbf{I} + \hat{\eta}\mathbf{B}, -\hat{\eta}\mathbf{g}; \alpha_1)$ and $\hat{\mathbf{x}}_+ \leftarrow \mathbf{y} + \mathbf{s}_+$
8: **end while**
9: **if** $\hat{\eta} = \eta$ **then**
10:     **Return** $\hat{\eta}$ and $\hat{\mathbf{x}}_+$
11: **else**
12:     **Return** $\hat{\eta}$, $\hat{\mathbf{x}}_+$ and $\tilde{\mathbf{x}}_+$
13: **end if**

---

# B  Line Search Subroutine

In this section, we provide further details on our line search subroutine in Section 3.1. For completeness, the pseudocode of our line search scheme is shown in Subroutine 1. In Section B.1, we prove that Subrountine 1 will always terminate in a finite number of steps. In Section B.2, we provide the proof of Lemma 1.

## B.1  The Line Search Subroutine Terminates Properly

Recall that in our line search scheme, we keep decreasing the step size $\hat{\eta}$ by a factor of $\beta$ until we find a pair $(\hat{\eta}, \hat{\mathbf{x}}_+)$ satisfying (11) (also see Lines 5 and 6 in Subroutine 1). In the following lemma, we show that when the step size $\hat{\eta}$ is smaller than a certain threshold, then the pair $(\hat{\eta}, \hat{\mathbf{x}}_+)$ satisfies both conditions in (10) and (11), which further implies that Subroutine 1 will stop in a finite number of steps.

**Lemma 9.** *Suppose Assumption 1 holds. If $\hat{\eta} < \frac{\alpha_2}{L_1 + \|\mathbf{B}\|_{\mathrm{op}}}$ and $\hat{\mathbf{x}}_+$ is computed according to* (12)*, then the pair $(\hat{\eta}, \hat{\mathbf{x}}_+)$ satisfies the conditions in* (10) *and* (11)*.*

*Proof.* By Definition 1, the pair $(\hat{\eta}, \hat{\mathbf{x}}_+)$ always satisfies the condition in (10) when $\hat{\mathbf{x}}_+$ is computed from (12). Hence, in the following we only need to prove that the condition in (11) also holds. Recall that $\mathbf{g} = \nabla f(\mathbf{y})$. By Assumption 1, the function $f$ is $L_1$-smooth and thus we have

$$\|\nabla f(\hat{\mathbf{x}}_+) - \mathbf{g}\| = \|\nabla f(\hat{\mathbf{x}}_+) - \nabla f(\mathbf{y})\| \leq L_1\|\hat{\mathbf{x}}_+ - \mathbf{y}\|.$$

Moreover, by using the triangle inequality, we get

$$\|\nabla f(\hat{\mathbf{x}}_+) - \mathbf{g} - \mathbf{B}(\hat{\mathbf{x}}_+ - \mathbf{y})\| \leq \|\nabla f(\hat{\mathbf{x}}_+) - \mathbf{g}\| + \|\mathbf{B}(\hat{\mathbf{x}}_+ - \mathbf{y})\| \leq (L_1 + \|\mathbf{B}\|_{\mathrm{op}})\|\hat{\mathbf{x}}_+ - \mathbf{y}\|.$$

Hence, if $\hat{\eta} \leq \frac{\alpha_2}{L_1 + \|\mathbf{B}\|_{\mathrm{op}}}$, we have

$$\hat{\eta}\|\nabla f(\hat{\mathbf{x}}_+) - \mathbf{g} - \mathbf{B}(\hat{\mathbf{x}}_+ - \mathbf{y})\| \leq \alpha_2\|\hat{\mathbf{x}}_+ - \mathbf{y}\|. \tag{57}$$

Finally, by using the triangle inequality, we can combine (10) and (57) to show that

$$\begin{aligned}
\|\hat{\mathbf{x}}_+ - \mathbf{y} + \hat{\eta}\nabla f(\hat{\mathbf{x}}_+)\| &= \|\hat{\mathbf{x}}_+ - \mathbf{y} + \hat{\eta}(\mathbf{g} + \mathbf{B}(\hat{\mathbf{x}}_+ - \mathbf{y})) + \hat{\eta}(\nabla f(\hat{\mathbf{x}}_+) - \mathbf{g} - \mathbf{B}(\hat{\mathbf{x}}_+ - \mathbf{y}))\| \\
&\leq \|\hat{\mathbf{x}}_+ - \mathbf{y} + \hat{\eta}(\mathbf{g} + \mathbf{B}(\hat{\mathbf{x}}_+ - \mathbf{y}))\| + \|\hat{\eta}(\nabla f(\hat{\mathbf{x}}_+) - \mathbf{g} - \mathbf{B}(\hat{\mathbf{x}}_+ - \mathbf{y}))\| \\
&\leq \alpha_1\|\hat{\mathbf{x}}_+ - \mathbf{y}\| + \alpha_2\|\hat{\mathbf{x}}_+ - \mathbf{y}\| \\
&\leq (\alpha_1 + \alpha_2)\|\hat{\mathbf{x}}_+ - \mathbf{y}\|,
\end{aligned}$$

which means the condition in (11) is satisfied. The proof is now complete. $\qquad\square$

## B.2  Proof of Lemma 1

We follow a similar proof strategy as Lemma 3 in [35]. In the first case where $k \notin \mathcal{B}$, by definition, the line search subroutine accepts the initial step size $\eta_k$, i.e., $\hat{\eta}_k = \eta_k$. In the second case where $k \in \mathcal{B}$, the line search subroutine backtracks and returns the auxiliary iterate $\tilde{\mathbf{x}}_{k+1}$, which is computed from

(12) using the step size $\tilde{\eta}_k \triangleq \hat{\eta}_k/\beta$. Since the step size $\tilde{\eta}_k$ is rejected in our line search subroutine, it implies that the pair $(\tilde{\mathbf{x}}_{k+1}, \tilde{\eta}_k)$ does not satisfy (11), i.e.,

$$\|\tilde{\mathbf{x}}_{k+1} - \mathbf{y}_k + \tilde{\eta}_k \nabla f(\tilde{\mathbf{x}}_{k+1})\| > (\alpha_1 + \alpha_2)\|\tilde{\mathbf{x}}_{k+1} - \mathbf{y}_k\|. \tag{58}$$

Moreover, since we compute $\tilde{\mathbf{x}}_{k+1}$ from (12) using step size $\tilde{\eta}_k$, the pair $(\tilde{\eta}_k, \tilde{\mathbf{x}}_{k+1})$ also satisfies the condition in (10), which means

$$\|\tilde{\mathbf{x}}_{k+1} - \mathbf{y}_k + \tilde{\eta}_k(\nabla f(\mathbf{y}_k) + \mathbf{B}_k(\tilde{\mathbf{x}}_{k+1} - \mathbf{y}_k))\| \leq \alpha_1\|\tilde{\mathbf{x}}_{k+1} - \mathbf{y}_k\|. \tag{59}$$

Hence, by using the triangle inequality, we can combine (58) and (59) to get

$$\begin{aligned}
&\tilde{\eta}_k\|\nabla f(\tilde{\mathbf{x}}_{k+1}) - \nabla f(\mathbf{y}_k) - \mathbf{B}_k(\tilde{\mathbf{x}}_{k+1} - \mathbf{y}_k)\| \\
&\geq \|\tilde{\mathbf{x}}_{k+1} - \mathbf{y}_k + \tilde{\eta}_k \nabla f(\tilde{\mathbf{x}}_{k+1})\| - \|\tilde{\mathbf{x}}_{k+1} - \mathbf{y}_k + \tilde{\eta}_k(\nabla f(\mathbf{y}_k) + \mathbf{B}_k(\tilde{\mathbf{x}}_{k+1} - \mathbf{y}_k))\| \\
&> (\alpha_1 + \alpha_2)\|\tilde{\mathbf{x}}_{k+1} - \mathbf{y}_k\| - \alpha_1\|\tilde{\mathbf{x}}_{k+1} - \mathbf{y}_k\| \\
&= \alpha_2\|\tilde{\mathbf{x}}_{k+1} - \mathbf{y}_k\|,
\end{aligned}$$

which implies that

$$\hat{\eta}_k = \beta\tilde{\eta}_k > \frac{\alpha_2\beta\|\tilde{\mathbf{x}}_{k+1} - \mathbf{y}_k\|}{\|\nabla f(\tilde{\mathbf{x}}_{k+1}) - \nabla f(\mathbf{y}_k) - \mathbf{B}_k(\tilde{\mathbf{x}}_{k+1} - \mathbf{y}_k)\|}.$$

This proves the first inequality in (13).

To show the second inequality in (13), first note that $\tilde{\mathbf{x}}_{k+1}$ and $\hat{\mathbf{x}}_{k+1}$ are the inexact solutions of the linear system of equations

$$(\mathbf{I} + \tilde{\eta}_k\mathbf{B}_k)(\mathbf{x} - \mathbf{y}_k) = -\tilde{\eta}_k\mathbf{g}_k \quad \text{and} \quad (\mathbf{I} + \hat{\eta}_k\mathbf{B}_k)(\mathbf{x} - \mathbf{y}_k) = -\hat{\eta}_k\mathbf{g}_k,$$

respectively. Let $\tilde{\mathbf{x}}_{k+1}^*$ and $\hat{\mathbf{x}}_{k+1}^*$ be the exact solutions of the above linear systems, that is, $\tilde{\mathbf{x}}_{k+1}^* = \mathbf{y}_k - \tilde{\eta}_k(\mathbf{I} + \tilde{\eta}_k\mathbf{B}_k)^{-1}\mathbf{g}_k$ and $\hat{\mathbf{x}}_{k+1}^* = \mathbf{y}_k - \hat{\eta}_k(\mathbf{I} + \hat{\eta}_k\mathbf{B}_k)^{-1}\mathbf{g}_k$. We first establish the following inequality between $\|\tilde{\mathbf{x}}_{k+1}^* - \mathbf{y}_k\|$ and $\|\hat{\mathbf{x}}_{k+1}^* - \mathbf{y}_k\|$:

$$\|\tilde{\mathbf{x}}_{k+1}^* - \mathbf{y}_k\| \leq \frac{1}{\beta}\|\hat{\mathbf{x}}_{k+1}^* - \mathbf{y}_k\|. \tag{60}$$

This follows from

$$\|\tilde{\mathbf{x}}_{k+1}^* - \mathbf{y}_k\| = \|\tilde{\eta}_k(\mathbf{I} + \tilde{\eta}_k\mathbf{B}_k)^{-1}\mathbf{g}_k\| \leq \tilde{\eta}_k\|(\mathbf{I} + \hat{\eta}_k\mathbf{B}_k)^{-1}\mathbf{g}_k\| = \frac{\tilde{\eta}_k}{\hat{\eta}_k}\|\hat{\mathbf{x}}_{k+1}^* - \mathbf{y}_k\| = \frac{1}{\beta}\|\hat{\mathbf{x}}_{k+1}^* - \mathbf{y}_k\|,$$

where we used the fact that $(\mathbf{I} + \tilde{\eta}_k\mathbf{B}_k)^{-1} \preceq (\mathbf{I} + \hat{\eta}_k\mathbf{B}_k)^{-1}$ in the first inequality. Furthermore, we can show that

$$(1 - \alpha_1)\|\hat{\mathbf{x}}_{k+1} - \mathbf{y}_k\| \leq \|\hat{\mathbf{x}}_{k+1}^* - \mathbf{y}_k\| \leq (1 + \alpha_1)\|\hat{\mathbf{x}}_{k+1} - \mathbf{y}_k\|, \tag{61}$$
$$(1 - \alpha_1)\|\tilde{\mathbf{x}}_{k+1} - \mathbf{y}_k\| \leq \|\tilde{\mathbf{x}}_{k+1}^* - \mathbf{y}_k\| \leq (1 + \alpha_1)\|\tilde{\mathbf{x}}_{k+1} - \mathbf{y}_k\|. \tag{62}$$

We will only prove (61) in the following, as (62) can be proved similarly. Note that since $(\hat{\eta}_k, \hat{\mathbf{x}}_{k+1})$ satisfies the condition in (10), we can write

$$\|\hat{\mathbf{x}}_{k+1} - \mathbf{y}_k + \hat{\eta}_k(\mathbf{g}_k + \mathbf{B}_k(\hat{\mathbf{x}}_{k+1} - \mathbf{y}_k))\| = \|(\mathbf{I} + \hat{\eta}_k\mathbf{B}_k)(\hat{\mathbf{x}}_{k+1} - \hat{\mathbf{x}}_{k+1}^*)\| \leq \alpha_1\|\hat{\mathbf{x}}_{k+1} - \mathbf{y}_k\|.$$

Moreover, since $\mathbf{B}_k \succeq 0$, we have $\|\hat{\mathbf{x}}_{k+1} - \hat{\mathbf{x}}_{k+1}^*\| \leq \|(\mathbf{I} + \hat{\eta}_k\mathbf{B}_k)(\hat{\mathbf{x}}_{k+1} - \hat{\mathbf{x}}_{k+1}^*)\| \leq \alpha_1\|\hat{\mathbf{x}}_{k+1} - \mathbf{y}_k\|$. Thus, by the triangle inequality, we obtain

$$\|\hat{\mathbf{x}}_{k+1}^* - \mathbf{y}_k\| \leq \|\hat{\mathbf{x}}_{k+1} - \mathbf{y}_k\| + \|\hat{\mathbf{x}}_{k+1}^* - \hat{\mathbf{x}}_{k+1}\| \leq (1 + \alpha_1)\|\hat{\mathbf{x}}_{k+1} - \mathbf{y}_k\|.$$
$$\|\hat{\mathbf{x}}_{k+1}^* - \mathbf{y}_k\| \geq \|\hat{\mathbf{x}}_{k+1} - \mathbf{y}_k\| - \|\hat{\mathbf{x}}_{k+1}^* - \hat{\mathbf{x}}_{k+1}\| \geq (1 - \alpha_1)\|\hat{\mathbf{x}}_{k+1} - \mathbf{y}_k\|.$$

which proves (61). Finally, by combining (60), (61) and (62), we conclude that

$$\|\tilde{\mathbf{x}}_{k+1} - \mathbf{y}_k\| \leq \frac{1}{1 - \alpha_1}\|\tilde{\mathbf{x}}_{k+1}^* - \mathbf{y}_k\| \leq \frac{1}{(1 - \alpha_1)\beta}\|\hat{\mathbf{x}}_{k+1}^* - \mathbf{y}_k\| \leq \frac{1 + \alpha_1}{(1 - \alpha_1)\beta}\|\hat{\mathbf{x}}_{k+1} - \mathbf{y}_k\|.$$

This completes the proof.

## C  Hessian Approximation Update

In this section, we first prove Lemma 2 in Section C.1 and remark on the computational cost of Euclidean projection in Section C.2. Then we present a general online learning algorithm using an approximate separation oracle in Section C.3 and fully describe our Hessian approximation update in Section C.4.

### C.1  Proof of Lemma 2

We decompose the sum $\sum_{k=0}^{N-1} \frac{1}{\hat{\eta}_k^2}$ as

$$\sum_{k=0}^{N-1} \frac{1}{\hat{\eta}_k^2} = \frac{1}{\hat{\eta}_0^2} + \sum_{1 \leq k \leq N-1, k \in \mathcal{B}} \frac{1}{\hat{\eta}_k^2} + \sum_{1 \leq k \leq N-1, k \notin \mathcal{B}} \frac{1}{\hat{\eta}_k^2} \tag{63}$$

Recall that we have $\hat{\eta}_k = \eta_k$ for $k \notin \mathcal{B}$. Hence, we can further bound the last term by

$$\sum_{1 \leq k \leq N-1, k \notin \mathcal{B}} \frac{1}{\hat{\eta}_k^2} = \sum_{1 \leq k \leq N-1, k \notin \mathcal{B}} \frac{1}{\eta_k^2} \leq \sum_{k=1}^{N-1} \frac{1}{\eta_k^2}$$

$$= \frac{1}{\eta_1^2} + \sum_{1 \leq k \leq N-2, k \in \mathcal{B}} \frac{1}{\eta_{k+1}^2} + \sum_{1 \leq k \leq N-2, k \notin \mathcal{B}} \frac{1}{\eta_{k+1}^2}.$$

Recall that we have $\eta_{k+1} = \hat{\eta}_k$ if $k \in \mathcal{B}$ and $\eta_{k+1} = \hat{\eta}_k/\beta$ otherwise. Hence, we further have

$$\sum_{1 \leq k \leq N-1, k \notin \mathcal{B}} \frac{1}{\hat{\eta}_k^2} \leq \frac{1}{\eta_1^2} + \sum_{1 \leq k \leq N-2, k \in \mathcal{B}} \frac{1}{\eta_{k+1}^2} + \sum_{1 \leq k \leq N-2, k \notin \mathcal{B}} \frac{1}{\eta_{k+1}^2}$$

$$= \frac{1}{\eta_1^2} + \sum_{1 \leq k \leq N-2, k \in \mathcal{B}} \frac{1}{\hat{\eta}_k^2} + \sum_{1 \leq k \leq N-2, k \notin \mathcal{B}} \frac{\beta^2}{\hat{\eta}_k^2}$$

$$\leq \frac{1}{\eta_1^2} + \sum_{1 \leq k \leq N-1, k \in \mathcal{B}} \frac{1}{\hat{\eta}_k^2} + \sum_{1 \leq k \leq N-1, k \notin \mathcal{B}} \frac{\beta^2}{\hat{\eta}_k^2}.$$

By moving the last term to the left-hand side and dividing both sides by $1 - \beta^2$, we obtain

$$\sum_{1 \leq k \leq N-1, k \notin \mathcal{B}} \frac{1}{\hat{\eta}_k^2} \leq \frac{1}{1 - \beta^2} \left( \frac{1}{\eta_1^2} + \sum_{1 \leq k \leq N-1, k \in \mathcal{B}} \frac{1}{\hat{\eta}_k^2} \right). \tag{64}$$

Furthermore, since $\eta_1 \geq \hat{\eta}_0$, we have $\frac{1}{\eta_1^2} \leq \frac{1}{\hat{\eta}_0^2}$. Hence, by combining (63) and (64), we get

$$\sum_{k=0}^{N-1} \frac{1}{\hat{\eta}_k^2} \leq \frac{2 - \beta^2}{1 - \beta^2} \left( \frac{1}{\hat{\eta}_0^2} + \sum_{1 \leq k \leq N-1, k \in \mathcal{B}} \frac{1}{\hat{\eta}_k^2} \right) \leq \frac{2 - \beta^2}{(1 - \beta^2)\sigma_0^2} + \frac{2 - \beta^2}{1 - \beta^2} \sum_{0 \leq k \leq N-1, k \in \mathcal{B}} \frac{1}{\hat{\eta}_k^2}, \tag{65}$$

where in the last inequality we used the fact that $\hat{\eta}_k = \sigma_0$ if $0 \notin \mathcal{B}$. Finally, (15) follows from Lemma 1 and (65).

### C.2  The Computational Cost of Euclidean Projection

Recall that $\mathcal{Z} \triangleq \{\mathbf{B} \in \mathbb{S}_+^d : 0 \preceq \mathbf{B} \preceq L_1 \mathbf{I}\}$. As described in [35, Section D.1], the Euclidean projection on $\mathcal{Z}$ has a closed form solution. Specifically, Given the input $\mathbf{A} \in \mathbb{S}^d$, we first need to perform the eigendecomposition $\mathbf{A} = \mathbf{V}\mathbf{\Lambda}\mathbf{V}^\top$, where $\mathbf{V}$ is an orthogonal matrix and $\mathbf{\Lambda} = \text{diag}(\lambda_1, \ldots, \lambda_d)$ is a diagonal matrix. Then the Euclidean projection of $\mathbf{A}$ onto $\mathcal{Z}$ is given by $\mathbf{V}\hat{\mathbf{\Lambda}}\mathbf{V}^\top$, where $\hat{\mathbf{\Lambda}}$ is a diagonal matrix with the diagonals being $\hat{\lambda}_k = \min\{L_1, \max\{0, \lambda_k\}\}$ for $1 \leq k \leq d$. Since the eigendecomposition requires $\mathcal{O}(d^3)$ arithmetic operations in general, the cost of computing the Euclidean projection can be prohibitive.

---

**Algorithm 2** Projection-Free Online Learning

---

1: **Input:** Initial point $\mathbf{w}_0 \in \mathcal{B}_R(0)$, step size $\rho > 0$, $\delta > 0$
2: **for** $t = 0, 1, \ldots T - 1$ **do**
3:      Query the oracle $(\gamma_t, \mathbf{s}_t) \leftarrow \mathsf{SEP}(\mathbf{w}_t; \delta_t)$
4:      **if** $\gamma_t \leq 1$ **then**     *# Case I: we have $\mathbf{w}_t \in \mathcal{C}$*
5:          Set $\mathbf{x}_t \leftarrow \mathbf{w}_t$ and play the action $\mathbf{x}_t$
6:          Receive the loss $\ell_t(\mathbf{x}_t)$ and the gradient $\mathbf{g}_t = \nabla \ell_t(\mathbf{x}_t)$
7:          Set $\tilde{\mathbf{g}}_t \leftarrow \mathbf{g}_t$
8:      **else**     *# Case II: we have $\mathbf{w}_t/\gamma_t \in \mathcal{C}$*
9:          Set $\mathbf{x}_t \leftarrow \mathbf{w}_t/\gamma_t$ and play the action $\mathbf{x}_t$
10:          Receive the loss $\ell_t(\mathbf{x}_t)$ and the gradient $\mathbf{g}_t = \nabla \ell_t(\mathbf{x}_t)$
11:          Set $\tilde{\mathbf{g}}_t \leftarrow \mathbf{g}_t + \max\{0, -\langle \mathbf{g}_t, \mathbf{x}_t \rangle\}\mathbf{s}_t$
12:      **end if**
13:      Update $\mathbf{w}_{t+1} \leftarrow \frac{R}{\max\{\|\mathbf{w}_t - \rho\tilde{\mathbf{g}}_t\|_2, R\}}(\mathbf{w}_t - \rho\tilde{\mathbf{g}}_t)$     *# Euclidean projection onto $\mathcal{B}_R(0)$*
14: **end for**

---

### C.3   Online Learning with an Approximate Separation Oracle

To set the stage for our Hessian approximation matrix update, we first describe a projection-free online learning algorithm in a general setup. Specifically, the online learning protocol is as follows: For rounds $t = 0, 1, \ldots, T - 1$, a learner chooses an action $\mathbf{x}_t \in \mathcal{C}$ from a convex set $\mathcal{C}$ and then observes a loss function $\ell_t : \mathbb{R}^n \to \mathbb{R}$. We measure the performance of an online learning algorithm by the dynamic regret [49, 52] defined by

$$\text{D-Reg}_T(\mathbf{u}_1, \ldots, \mathbf{u}_{T-1}) \triangleq \sum_{t=0}^{T-1} \ell_t(\mathbf{x}_t) - \sum_{t=0}^{T-1} \ell_t(\mathbf{u}_t),$$

where $\{\mathbf{u}_t\}_{t=1}^T$ is a sequence of comparators. Moreover, we assume that the convex set $\mathcal{C}$ is contained in the Euclidean ball $\mathcal{B}_R(0)$ for some $R > 0$, and we assume $0 \in \mathcal{C}$ without loss of generality.

Most existing online learning algorithms are projection-based, that is, they require computing the Euclidean projection on the action set $\mathcal{C}$. However, as we have seen in Section C.2, computing the projection is computationally costly in our setting. Inspired by the work in [50], we will describe an online learning algorithm that relies on an approximate separation oracle defined in Definition 3.

**Definition 3.** *The oracle $\mathsf{SEP}(\mathbf{w}; \delta)$ takes $\mathbf{w} \in \mathcal{B}_R(0)$ and $\delta > 0$ as input and returns a scalar $\gamma > 0$ and a vector $\mathbf{s} \in \mathbb{R}^n$ with one of the following possible outcomes:*

- *Case I: $\gamma \leq 1$ which implies that $\mathbf{w} \in \mathcal{C}$;*
- *Case II: $\gamma > 1$ which implies that $\mathbf{w}/\gamma \in \mathcal{C}$  and  $\langle \mathbf{s}, \mathbf{w} - \mathbf{x} \rangle \geq \gamma - 1 - \delta$   $\forall \mathbf{x} \in \mathcal{C}$.*

To sum up, the oracle $\mathsf{SEP}(\mathbf{w}; \delta)$ has two possible outcomes: it either certifies that $\mathbf{w}$ is feasible, i.e., $\mathbf{w} \in \mathcal{C}$, or it produces a scaled version of $\mathbf{w}$ that is in $\mathcal{C}$ and gives an approximate separating hyperplane between $\mathbf{w}$ and the set $\mathcal{C}$.

The full algorithm is shown in Algorithm 2. The key idea here is to introduce surrogate loss functions $\tilde{\ell}_t(\mathbf{w}) = \langle \tilde{\mathbf{g}}_t, \mathbf{w} \rangle$ on the larger set $\mathcal{B}_R(0)$ for $0 \leq t \leq T - 1$, where $\tilde{\mathbf{g}}_t$ is the surrogate gradient to be defined later. On a high level, we will run online projected gradient descent with $\tilde{\ell}_t(\mathbf{w})$ to update the auxiliary iterates $\{\mathbf{w}_t\}_{t \geq 0}$ (note that the projection on $\mathcal{B}_R(0)$ is easy to compute), and then produce the actions $\{\mathbf{x}_t\}_{t \geq 0}$ for the original problem by calling the $\mathsf{SEP}(\mathbf{w}_t; \delta)$ oracle in Definition 3. The follow lemma shows that the immediate regret $\tilde{\ell}_t(\mathbf{w}_t) - \tilde{\ell}_t(\mathbf{x})$ can serve as an upper bound on $\ell_t(\mathbf{x}_t) - \ell_t(\mathbf{x})$ for any $\mathbf{x} \in \mathcal{C}$.

**Lemma 10.** *Let $\{\mathbf{x}_t\}_{t=0}^{T-1}$ be the iterates generated by Algorithm 2. Then we have $\mathbf{x}_t \in \mathcal{C}$ for $t = 0, 1, \ldots, T - 1$. Also, for any $\mathbf{x} \in \mathcal{C}$, we have*

$$\langle \mathbf{g}_t, \mathbf{x}_t - \mathbf{x} \rangle \leq \langle \tilde{\mathbf{g}}_t, \mathbf{w}_t - \mathbf{x} \rangle + \max\{0, -\langle \mathbf{g}_t, \mathbf{x}_t \rangle\}\delta_t \tag{66}$$

$$\leq \frac{1}{2\rho}\|\mathbf{w}_t - \mathbf{x}\|_2^2 - \frac{1}{2\rho}\|\mathbf{w}_{t+1} - \mathbf{x}\|_2^2 + \frac{\rho}{2}\|\tilde{\mathbf{g}}_t\|_2^2 + \max\{0, -\langle \mathbf{g}_t, \mathbf{x}_t \rangle\}\delta_t, \tag{67}$$

*and*

$$\|\tilde{\mathbf{g}}_t\| \leq \|\mathbf{g}_t\| + |\langle \mathbf{g}_t, \mathbf{x}_t \rangle|\|\mathbf{s}_t\|. \tag{68}$$

---

**Subroutine 2** Online Learning Guided Hessian Approximation Update

---

1: **Input:** Initial matrix $\mathbf{B}_0 \in \mathbb{S}^d$ s.t. $0 \preceq \mathbf{B}_0 \preceq L_1\mathbf{I}$, step size $\rho > 0$, $\delta > 0$, $\{q_t\}_{t=1}^{T-1}$
2: **Initialize:** set $\mathbf{W}_0 \leftarrow \frac{2}{L_1}(\mathbf{B}_0 - \frac{L_1}{2}\mathbf{I})$, $\mathbf{G}_0 \leftarrow \frac{2}{L_1}\nabla\ell_0(\mathbf{B}_0)$ and $\tilde{\mathbf{G}}_0 \leftarrow \mathbf{G}_0$
3: **for** $t = 1, \ldots, T-1$ **do**
4:     Query the oracle $(\gamma_t, \mathbf{S}_t) \leftarrow \mathsf{SEP}(\mathbf{W}_t; \delta_t, q_t)$
5:     **if** $\gamma_t \leq 1$ **then**    *# Case I*
6:         Set $\hat{\mathbf{B}}_t \leftarrow \mathbf{W}_t$ and $\mathbf{B}_t \leftarrow \frac{L_1}{2}\hat{\mathbf{B}}_t + \frac{L_1}{2}\mathbf{I}$
7:         Set $\mathbf{G}_t \leftarrow \frac{2}{L_1}\nabla\ell_t(\mathbf{B}_t)$ and $\tilde{\mathbf{G}}_t \leftarrow \mathbf{G}_t$
8:     **else**    *# Case II*
9:         Set $\hat{\mathbf{B}}_t \leftarrow \mathbf{W}_t/\gamma_t$ and $\mathbf{B}_t \leftarrow \frac{L_1}{2}\hat{\mathbf{B}}_t + \frac{L_1}{2}\mathbf{I}$
10:        Set $\mathbf{G}_t \leftarrow \frac{2}{L_1}\nabla\ell_t(\mathbf{B}_t)$ and $\tilde{\mathbf{G}}_t \leftarrow \mathbf{G}_t + \max\{0, -\langle\mathbf{G}_t, \mathbf{B}_t\rangle\}\mathbf{S}_t$
11:     **end if**
12:     Update $\mathbf{W}_{t+1} \leftarrow \frac{\sqrt{d}}{\max\{\sqrt{d}, \|\mathbf{W}_t - \rho\tilde{\mathbf{G}}_t\|_F\}}(\mathbf{W}_t - \rho\tilde{\mathbf{G}}_t)$    *# Euclidean projection onto $\mathcal{B}_{\sqrt{d}}(0)$*
13: **end for**

---

*Proof.* By the definition of $\mathsf{SEP}$ in Definition 3, we can see that $\mathbf{x}_t \in \mathcal{C}$ for all $t = 1, \ldots, T$. We now show that both (66) and (68) hold. We distinguish two cases depending on the outcomes of $\mathsf{SEP}(\mathbf{w}_t; \delta_t)$.

- If $\gamma_t \leq 1$, then we have $\mathbf{x}_t = \mathbf{w}_t$ and $\tilde{\mathbf{g}}_t = \mathbf{g}_t$. In this case, (66) and (68) trivially hold.

- If $\gamma_t > 1$, then $\mathbf{x}_t = \mathbf{w}_t/\gamma_t$ and $\tilde{\mathbf{g}}_t = \mathbf{g}_t + \max\{0, -\langle\mathbf{g}_t, \mathbf{x}_t\rangle\}\mathbf{s}_t$. We can then write

$$\langle\tilde{\mathbf{g}}_t, \mathbf{w}_t - \mathbf{x}\rangle = \langle\mathbf{g}_t + \max\{0, -\langle\mathbf{g}_t, \mathbf{x}_t\rangle\}\mathbf{s}_t, \mathbf{w}_t - \mathbf{x}\rangle$$
$$= \langle\mathbf{g}_t, \gamma_t\mathbf{x}_t - \mathbf{x}\rangle + \max\{0, -\langle\mathbf{g}_t, \mathbf{x}_t\rangle\}\langle\mathbf{s}_t, \mathbf{w}_t - \mathbf{x}\rangle$$
$$\geq \langle\mathbf{g}_t, \mathbf{x}_t - \mathbf{x}\rangle + (\gamma_t - 1)\langle\mathbf{g}_t, \mathbf{x}_t\rangle + \max\{0, -\langle\mathbf{g}_t, \mathbf{x}_t\rangle\}(\gamma_t - 1 - \delta_t)$$
$$= \langle\mathbf{g}_t, \mathbf{x}_t - \mathbf{x}\rangle - \max\{0, -\langle\mathbf{g}_t, \mathbf{x}_t\rangle\}\delta_t + (\gamma_t - 1)\max\{0, \langle\mathbf{g}_t, \mathbf{x}_t\rangle\}$$
$$\geq \langle\mathbf{g}_t, \mathbf{x}_t - \mathbf{x}\rangle - \max\{0, -\langle\mathbf{g}_t, \mathbf{x}_t\rangle\}\delta_t,$$

which leads to (66) after rearranging. Also, by the triangle inequality we obtain

$$\|\tilde{\mathbf{g}}_t\| \leq \|\mathbf{g}_t\| + \max\{0, -\langle\mathbf{g}_t, \mathbf{x}_t\rangle\}\|\mathbf{s}_t\| \leq \|\mathbf{g}_t\| + |\langle\mathbf{g}_t, \mathbf{x}_t\rangle|\|\mathbf{s}_t\|,$$

which proves (68).

Finally, from the update rule of $\mathbf{w}_{t+1}$, for any $\mathbf{x} \in \mathcal{C} \subset \mathcal{B}_R(0)$ we have $\langle\mathbf{w}_t - \rho\tilde{\mathbf{g}}_t - \mathbf{w}_{t+1}, \mathbf{w}_{t+1} - \mathbf{x}\rangle \geq 0$, which further implies that

$$\langle\tilde{\mathbf{g}}_t, \mathbf{w}_t - \mathbf{x}\rangle \leq \langle\tilde{\mathbf{g}}_t, \mathbf{w}_t - \mathbf{w}_{t+1}\rangle + \frac{1}{\rho}\langle\mathbf{w}_t - \mathbf{w}_{t+1}, \mathbf{w}_{t+1} - \mathbf{x}\rangle \tag{69}$$

$$= \langle\tilde{\mathbf{g}}_t, \mathbf{w}_t - \mathbf{w}_{t+1}\rangle + \frac{1}{2\rho}\|\mathbf{w}_t - \mathbf{x}\|_2^2 - \frac{1}{2\rho}\|\mathbf{w}_{t+1} - \mathbf{x}\|_2^2 - \frac{1}{2\rho}\|\mathbf{w}_t - \mathbf{w}_{t+1}\|_2^2 \tag{70}$$

$$\leq \frac{1}{2\rho}\|\mathbf{w}_t - \mathbf{x}\|_2^2 - \frac{1}{2\rho}\|\mathbf{w}_{t+1} - \mathbf{x}\|_2^2 + \frac{\rho}{2}\|\tilde{\mathbf{g}}_t\|_2^2. \tag{71}$$

Combining (66) and (71) leads to (67). $\qquad\square$

### C.4 Projection-free Hessian Approximation Update

Now we are ready to describe our Hessian approximation matrix update, which is an specific instantiation of the general projection-free online learning algorithm shown in Algorithm 2. In particular, we only need to specify the convex set $\mathcal{C}$ as well as the $\mathsf{SEP}$ oracle.

Note that we have $\mathcal{Z} = \{\mathbf{B} \in \mathbb{S}_+^d : 0 \preceq \mathbf{B} \preceq L_1\mathbf{I}\}$ in our online learning problem in Section 3.2. Since the projection-free scheme in Subroutine 2 requires the set $\mathcal{C}$ to contain the origin, we consider the transform $\hat{\mathbf{B}} \triangleq \frac{2}{L_1}(\mathbf{B} - \frac{L_1}{2}\mathbf{I})$ and let $\mathcal{C} = \hat{\mathcal{Z}} \triangleq \{\hat{\mathbf{B}} \in \mathbb{S}^d : -\mathbf{I} \preceq \hat{\mathbf{B}} \preceq \mathbf{I}\} = \{\hat{\mathbf{B}} \in \mathbb{S}^d : \|\hat{\mathbf{B}}\|_{\mathrm{op}} \leq 1\}$. We note that $0 \in \hat{\mathcal{Z}}$ and $\hat{\mathcal{Z}} \subset \mathcal{B}_{\sqrt{d}}(0) = \{\mathbf{W} \in \mathbb{S}^d : \|\mathbf{W}\|_F \leq \sqrt{d}\}$. Moreover, we can see that the approximate separation oracle $\mathsf{SEP}(\mathbf{W}; \delta, q)$ defined in Definition 2 corresponds to a stochastic version of the oracle in Definition 3. The full algorithm is described in Subroutine 2 and we defer the specific implementation details of $\mathsf{SEP}(\mathbf{W}; \delta, q)$ to Section E.2.

# D Proof of Theorem 1

Regarding the choices of the hyper-parameters, we consider Algorithm 1 with the line search scheme in Subroutine 1, where $\alpha_1, \alpha_2 \in (0,1)$ with $\alpha_1 + \alpha_2 < 1$ and $\beta \in (0,1)$, and with the Hessian approximation update in Subroutine 2, where $\rho = \frac{1}{128}$, $q_t = p/2.5(t+1)\log^2(t+1)$ for $t \geq 1$, and $\delta_t = 1/(\sqrt{t+2}\ln(t+2))$ for $t \geq 0$. In the following, we first provide a proof sketch of Theorem 1. The complete proofs of the lemmas shown below will be provided in the subsequent sections.

*Proof Sketch.* To begin with, throughout the proof, we assume that every call of the SEP oracle in Definition 2 is successful during the execution of Algorithm 1. Indeed, by using the union bound, we can bound the failure probability by $\sum_{t=1}^{T-1} q_t \leq \frac{p}{2.5} \sum_{t=2}^{\infty} \frac{1}{t\log^2 t} \leq p$. In particular, we note that Subroutine 2 ensures that $0 \preceq \mathbf{B}_k \preceq L_1 \mathbf{I}$ for any $k \geq 0$.

We first prove Part (a) of Theorem 1, which relies on the following lemma.

**Lemma 11.** *For $k \in \mathcal{B}$, we have $\ell_k(\mathbf{B}_k) \triangleq \frac{\|\mathbf{w}_k - \mathbf{B}_k \mathbf{s}_k\|^2}{\|\mathbf{s}_k\|^2} \leq L_1^2$.*

We combine Lemma 2 and Lemma 11 to derive

$$\sum_{k=0}^{N-1} \frac{1}{\hat{\eta}_k^2} \leq \frac{2-\beta^2}{(1-\beta^2)\sigma_0^2} + \frac{2-\beta^2}{(1-\beta^2)\alpha_2^2\beta^2} \sum_{k\in\mathcal{B}} \frac{\|\mathbf{w}_k - \mathbf{B}_k \mathbf{s}_k\|^2}{\|\mathbf{s}_k\|^2} \leq \frac{2-\beta^2}{(1-\beta^2)\sigma_0^2} + \frac{(2-\beta^2)L_1^2}{(1-\beta^2)\alpha_2^2\beta^2} N.$$

By further using (14) and the elementary inequality that $\sqrt{a+b} \leq \sqrt{a} + \sqrt{b}$, we obtain

$$f(\mathbf{x}_N) - f(\mathbf{x}^*) \leq \frac{C_4 L_1 \|\mathbf{z}_0 - \mathbf{x}^*\|^2}{N^2} + \frac{C_5 \|\mathbf{z}_0 - \mathbf{x}^*\|^2}{\sigma_0 N^{2.5}},$$

where $C_4 = C_1 \sqrt{\frac{2-\beta^2}{(1-\beta^2)\sigma_0^2} + \frac{(2-\beta^2)}{(1-\beta^2)\alpha_2^2\beta^2}}$ and $C_5 = C_1 \sqrt{\frac{2-\beta^2}{(1-\beta^2)\sigma_0^2}}$.

Next, we divide the proof of Part (b) of Theorem 1 into the following steps.

**Step 1:** We first use regret analysis to control the cumulative loss $\sum_{t=0}^{T-1} \ell_t(\mathbf{B}_t)$ incurred by our online learning algorithm in Subroutine 2. In particular, we prove a dynamic regret bound, where we compare the cumulative loss of our algorithm against the one achieved by the sequence $\{\mathbf{H}_t\}_{t=0}^{T-1}$.

**Lemma 12.** *We have*

$$\sum_{t=0}^{T-1} \ell_t(\mathbf{B}_t) \leq 256\|\mathbf{B}_0 - \mathbf{H}_0\|_F^2 + 4\sum_{t=0}^{T-1} \ell_t(\mathbf{H}_t) + 2L_1^2 \sum_{t=0}^{T-1} \delta_t^2 + 512 L_1 \sqrt{d} \sum_{t=0}^{T-1} \|\mathbf{H}_{t+1} - \mathbf{H}_t\|_F,$$

*where $\mathbf{H}_t \triangleq \nabla^2 f(\mathbf{y}_t)$.*

**Step 2:** In light of Lemma 12, it suffices to upper bound the cumulative loss $\sum_{t=0}^{T-1} \ell_t(\mathbf{H}_t)$ and the path-length $\sum_{t=0}^{T-1} \|\mathbf{H}_{t+1} - \mathbf{H}_t\|_F$ in the following lemma. To achieve this, we use the stability properties of our algorithm in (21) and Lemma 8, which is most technical part of the proof.

**Lemma 13.** *We have*

$$\sum_{t=0}^{T-1} \ell_t(\mathbf{H}_t) \leq \frac{C_3}{4} L_2^2 \|\mathbf{z}_0 - \mathbf{x}^*\|^2 \quad \text{and} \quad \sum_{t=0}^{T-1} \|\mathbf{H}_{t+1} - \mathbf{H}_t\|_F \leq C_2 \sqrt{d} L_2 \left(1 + \log \frac{A_N}{A_1}\right) \|\mathbf{z}_0 - \mathbf{x}^*\|,$$

(72)

*where $C_2$ is defined in (49) and $C_3 = \frac{(1+\alpha_1)^2}{\beta^2(1-\alpha_1)^2(1-\sigma^2)}$.*

**Step 3:** Thus, we obtain an upper bound on $\sum_{t=0}^{T-1} \ell_t(\mathbf{B}_t)$ by combining Lemma 12 and Lemma 13. Finally, in the following lemma, we prove an upper bound on $\frac{1}{A_N}$ by further using Lemma 2 and Proposition 1.

**Lemma 14.** *We have*

$$\frac{1}{A_N} \leq \frac{1}{N^{2.5}} \left( M + C_{10} L_1 L_2 d \|\mathbf{z}_0 - \mathbf{x}^*\| \log^+ \left( \frac{\max\{\frac{L_1}{\alpha_2 \beta}, \frac{1}{\sigma_0}\} N^{2.5}}{\sqrt{M}} \right) \right)^{\frac{1}{2}},$$

*where we define* $\log^+(x) \triangleq \max\{\log(x), 0\}$,

$$M = \frac{C_6}{\sigma_0^2} + C_7 L_1^2 + C_8 \|\mathbf{B}_0 - \mathbf{H}_0\|_F^2 + C_9 L_2^2 \|\mathbf{z}_0 - \mathbf{x}^*\|^2 + C_{10} L_1 L_2 d \|\mathbf{z}_0 - \mathbf{x}^*\|,$$

*and* $C_i$ *(*$i = 6, \ldots, 10$*) are absolute constants given by*

$$C_6 = \frac{4 C_1^2 (2 - \beta^2)}{1 - \beta^2}, \; C_7 = \frac{5 C_6}{\alpha_2^2 \beta^2}, \; C_8 = \frac{256 C_6}{\alpha_2^2 \beta^2}, \; C_9 = \frac{C_3 C_6}{\alpha_2^2 \beta^2}, \; C_{10} = \frac{512 C_2 C_6}{\alpha_2^2 \beta^2}.$$

*Therefore, Part (b) of Theorem 1 immediately follows from Proposition 1.* □

In the remaining of this section, we present the proofs for the above lemmas that we used to prove the results in Theorem 1.

## D.1 Proof of Lemma 11

Recall that $\mathbf{w}_k \triangleq \nabla f(\tilde{\mathbf{x}}_{k+1}) - \nabla f(\mathbf{y}_k)$ and $\mathbf{s}_k \triangleq \tilde{\mathbf{x}}_{k+1} - \mathbf{y}_k$ for $k \in \mathcal{B}$. We can write $\nabla f(\tilde{\mathbf{x}}_{k+1}) - \nabla f(\mathbf{y}_k) = \bar{\mathbf{H}}_k(\tilde{\mathbf{x}}_{k+1} - \mathbf{y}_k)$ by using the fundamental theorem of calculus, where $\bar{\mathbf{H}}_k = \int_0^1 \nabla^2 f(t\tilde{\mathbf{x}}_{k+1} + (1 - t)\mathbf{y}_k) \, dt$. Since we have $0 \preceq \nabla^2 f(\mathbf{x}) \preceq L_1 \mathbf{I}$ for all $\mathbf{x} \in \mathbb{R}^d$ by Assumption 1, it implies that $0 \preceq \bar{\mathbf{H}}_k \preceq L_1 \mathbf{I}$. Moreover, since $0 \preceq \mathbf{B}_k \preceq L_1 \mathbf{I}$, we further have $-L_1 \mathbf{I} \preceq \bar{\mathbf{H}}_k - \mathbf{B}_k \preceq L_1 \mathbf{I}$, which yields $\|\bar{\mathbf{H}}_k - \mathbf{B}_k\|_{\mathrm{op}} \leq L_1$. Thus, we have

$$\|\mathbf{w}_k - \mathbf{B}_k \mathbf{s}_k\| = \|(\bar{\mathbf{H}}_k - \mathbf{B}_k)(\tilde{\mathbf{x}}_{k+1} - \mathbf{y}_k)\| \leq L_1 \|\tilde{\mathbf{x}}_{k+1} - \mathbf{y}_k\|,$$

which proves that $\ell_k(\mathbf{B}_k) \leq L_1^2$.

## D.2 Proof of Lemma 12

To prove Lemma 12, we first present the following lemma showing a smooth property of the loss function $\ell_k$. The proof is similar to [35, Lemma 15].

**Lemma 15.** *For* $k \in \mathcal{B}$, *we have*

$$\nabla \ell_k(\mathbf{B}) = \frac{1}{\|\mathbf{s}_k\|^2} \left( -\mathbf{s}_k (\mathbf{w}_k - \mathbf{B}\mathbf{s}_k)^\mathsf{T} - (\mathbf{w}_k - \mathbf{B}\mathbf{s}_k)\mathbf{s}_k^\mathsf{T} \right). \tag{73}$$

*Moreover, for any* $\mathbf{B} \in \mathbb{S}^d$, *it holds that*

$$\|\nabla \ell_k(\mathbf{B})\|_F \leq \|\nabla \ell_k(\mathbf{B})\|_* \leq 2\sqrt{\ell_k(\mathbf{B})}, \tag{74}$$

*where* $\|\cdot\|_F$ *and* $\|\cdot\|_*$ *denote the Frobenius norm and the nuclear norm, respectively.*

*Proof.* It is straightforward to verify the expression in (73). The first inequality in (74) follows from the fact that $\|\mathbf{A}\|_F \leq \|\mathbf{A}\|_*$ for any matrix $\mathbf{A} \in \mathbb{S}^d$. For the second inequality, note that

$$\|\nabla \ell_k(\mathbf{B})\|_* \leq \frac{1}{\|\mathbf{s}_k\|^2} \left( \|\mathbf{s}_k (\mathbf{w}_k - \mathbf{B}\mathbf{s}_k)^\mathsf{T}\|_* + \|(\mathbf{w}_k - \mathbf{B}\mathbf{s}_k)\mathbf{s}_k^\mathsf{T}\|_* \right)$$

$$\leq \frac{2}{\|\mathbf{s}_k\|^2} \|\mathbf{w}_k - \mathbf{B}\mathbf{s}_k\| \|\mathbf{s}_k\| = \frac{2\|\mathbf{w}_k - \mathbf{B}\mathbf{s}_k\|}{\|\mathbf{s}_k\|} = 2\sqrt{\ell_k(\mathbf{B})},$$

where in the first inequality we used the triangle inequality, and in the second inequality we used the fact that the rank-one matrix $\mathbf{u}\mathbf{v}^\top$ has only one nonzero singular value $\|\mathbf{u}\| \|\mathbf{v}\|$. □

We will also need the following helper lemma.

**Lemma 16.** *If the real number* $x$ *satisfies* $x \leq A + B\sqrt{x}$, *then we have* $x \leq 2A + B^2$.

*Proof.* From the assumption, we have

$$\left(\sqrt{x} - \frac{B}{2}\right)^2 \le A + \frac{B^2}{4}.$$

Hence, we obtain

$$x \le \left(\sqrt{A + \frac{B^2}{4}} + \frac{B}{2}\right)^2 \le 2A + B^2.$$

$\square$

Before proving Lemma 12, we also present the following lemma that bounds the loss in each round.

**Lemma 17.** *For any* $\mathbf{H} \in \mathcal{Z}$, *we have*

$$\ell_t(\mathbf{B}_t) \le 4\ell_t(\mathbf{H}) + 64L_1^2\|\mathbf{W}_t - \hat{\mathbf{H}}\|_F^2 - 64L_1^2\|\mathbf{W}_{t+1} - \hat{\mathbf{H}}\|_F^2 + 2L_1^2\delta_t^2.$$

*Proof.* By letting $\mathbf{x}_t = \hat{\mathbf{B}}_t$, $\mathbf{x} = \hat{\mathbf{H}} \triangleq \frac{2}{L_1}(\mathbf{H} - \frac{L_1}{2}\mathbf{I})$, $\mathbf{g}_t = \mathbf{G}_t \triangleq \frac{2}{L_1}\nabla\ell_t(\mathbf{B}_t)$, $\tilde{\mathbf{g}}_t = \tilde{\mathbf{G}}_t$, $\mathbf{w}_t = \mathbf{W}_t$ in Lemma 10, we obtain:

(i) $\hat{\mathbf{B}}_t \in \hat{\mathcal{Z}}$, which means that $\|\hat{\mathbf{B}}_t\|_{\mathrm{op}} \le 1$.

(ii) It holds that

$$\langle \mathbf{G}_t, \hat{\mathbf{B}}_t - \hat{\mathbf{H}}\rangle \le \frac{1}{2\rho}\|\mathbf{W}_t - \hat{\mathbf{H}}\|_F^2 - \frac{1}{2\rho}\|\mathbf{W}_{t+1} - \hat{\mathbf{H}}\|_F^2 + \frac{\rho}{2}\|\tilde{\mathbf{G}}_t\|_F^2 + \max\{0, -\langle\mathbf{G}_t, \hat{\mathbf{B}}_t\rangle\}\delta_t,$$
(75)

$$\|\tilde{\mathbf{G}}_t\|_F \le \|\mathbf{G}_t\|_F + |\langle\mathbf{G}_t, \hat{\mathbf{B}}_t\rangle|\|\mathbf{S}_t\|_F.$$
(76)

First, note that $\|\mathbf{S}_t\|_F \le 3$ by Definition 2 and $|\langle\mathbf{G}_t, \hat{\mathbf{B}}_t\rangle| \le \|\mathbf{G}_t\|_*\|\hat{\mathbf{B}}_t\|_{\mathrm{op}} \le \|\mathbf{G}_t\|_*$. Together with (76), we get

$$\|\tilde{\mathbf{G}}_t\|_F \le \|\mathbf{G}_t\|_F + 3\|\mathbf{G}_t\|_* \le 4\|\mathbf{G}_t\|_* \le \frac{16}{L_1}\sqrt{\ell_t(\mathbf{B}_t)},$$
(77)

where we used the fact that $\mathbf{G}_t = \frac{2}{L_1}\nabla\ell_t(\mathbf{B}_t)$ and Lemma 15 in the last inequality. Furthermore, since $\ell_t$ is convex, we have

$$\ell_t(\mathbf{B}_t) - \ell_t(\mathbf{H}) \le \langle\nabla\ell_t(\mathbf{B}_t), \mathbf{B}_t - \mathbf{H}\rangle = \left(\frac{L_1}{2}\right)^2 \langle\mathbf{G}_t, \hat{\mathbf{B}}_t - \hat{\mathbf{H}}\rangle,$$

where we used $\mathbf{G}_t = \frac{2}{L_1}\nabla\ell_t(\mathbf{B}_t)$, $\hat{\mathbf{B}}_t \triangleq \frac{2}{L_1}(\mathbf{B}_t - \frac{L_1}{2}\mathbf{I})$, and $\hat{\mathbf{H}} \triangleq \frac{2}{L_1}(\mathbf{H} - \frac{L_1}{2}\mathbf{I})$. Therefore, by combining (75) and (77) we get

$$\ell_t(\mathbf{B}_t) - \ell_t(\mathbf{H}) \le \frac{L_1^2}{8\rho}\|\mathbf{W}_t - \hat{\mathbf{H}}\|_F^2 - \frac{L_1^2}{8\rho}\|\mathbf{W}_{t+1} - \hat{\mathbf{H}}\|_F^2 + \frac{\rho}{8}L_1^2\|\tilde{\mathbf{G}}_t\|_F^2 + \frac{L_1^2}{4}\|\mathbf{G}_t\|_*\delta_t \quad (78)$$

$$\le \frac{L_1^2}{8\rho}\|\mathbf{W}_t - \hat{\mathbf{H}}\|_F^2 - \frac{L_1^2}{8\rho}\|\mathbf{W}_{t+1} - \hat{\mathbf{H}}\|_F^2 + 32\rho\ell_t(\mathbf{B}_t) + L_1\sqrt{\ell_t(\mathbf{B}_t)}\delta_t. \quad (79)$$

Note that $\ell_t(\mathbf{B}_t)$ appears on both sides of (79). By further applying Lemma 16, we obtain

$$\ell_t(\mathbf{B}_t) \le 2\ell_t(\mathbf{H}) + \frac{L_1^2}{4\rho}\|\mathbf{W}_t - \hat{\mathbf{H}}\|_F^2 - \frac{L_1^2}{4\rho}\|\mathbf{W}_{t+1} - \hat{\mathbf{H}}\|_F^2 + 64\rho\ell_t(\mathbf{B}_t) + L_1^2\delta_t^2.$$

Since $\rho = 1/128$, by rearranging and simplifying terms in the above inequality, we obtain

$$\ell_t(\mathbf{B}_t) \le 4\ell_t(\mathbf{H}) + 64L_1^2\|\mathbf{W}_t - \hat{\mathbf{H}}\|_F^2 - 64L_1^2\|\mathbf{W}_{t+1} - \hat{\mathbf{H}}\|_F^2 + 2L_1^2\delta_t^2.$$

$\square$

*Proof of Lemma 12.* We let $\mathbf{H}_t = \nabla^2 f(\mathbf{y}_t)$ for $t = 0, 1, \ldots, T-1$. Thus, we get

$$\ell_t(\mathbf{B}_t) \leq 4\ell_t(\mathbf{H}_t) + 64L_1^2\|\mathbf{W}_t - \hat{\mathbf{H}}_t\|_F^2 - 64L_1^2\|\mathbf{W}_{t+1} - \hat{\mathbf{H}}_t\|_F^2 + 2L_1^2\delta_t^2$$
$$= 4\ell_t(\mathbf{H}_t) + 64L_1^2\|\mathbf{W}_t - \hat{\mathbf{H}}_t\|_F^2 - 64L_1^2\|\mathbf{W}_{t+1} - \hat{\mathbf{H}}_{t+1}\|_F^2 + 2L_1^2\delta_t^2$$
$$+ 64L_1^2\big(\|\mathbf{W}_{t+1} - \hat{\mathbf{H}}_{t+1}\|_F^2 - \|\mathbf{W}_{t+1} - \hat{\mathbf{H}}_t\|_F^2\big).$$

Furthermore, note that

$$\|\mathbf{W}_{t+1} - \hat{\mathbf{H}}_{t+1}\|_F^2 - \|\mathbf{W}_{t+1} - \hat{\mathbf{H}}_t\|_F^2$$
$$= (\|\mathbf{W}_{t+1} - \hat{\mathbf{H}}_{t+1}\|_F + \|\mathbf{W}_{t+1} - \hat{\mathbf{H}}_t\|_F)(\|\mathbf{W}_{t+1} - \hat{\mathbf{H}}_{t+1}\|_F - \|\mathbf{W}_{t+1} - \hat{\mathbf{H}}_t\|_F)$$
$$\leq 4\sqrt{d}\|\hat{\mathbf{H}}_{t+1} - \hat{\mathbf{H}}_t\|_F = \frac{8\sqrt{d}}{L_1}\|\mathbf{H}_{t+1} - \mathbf{H}_t\|_F,$$

where in the last inequality we used the fact that $\hat{\mathbf{H}}_t, \hat{\mathbf{H}}_{t+1}, \mathbf{W}_{t+1} \in \mathcal{B}_{\sqrt{d}}(0)$ and the triangle inequality. Therefore, we get

$$\ell_t(\mathbf{B}_t) \leq 4\ell_t(\mathbf{H}_t) + 64L_1^2\|\mathbf{W}_t - \hat{\mathbf{H}}_t\|_F^2 - 64L_1^2\|\mathbf{W}_{t+1} - \hat{\mathbf{H}}_{t+1}\|_F^2 + 2L_1^2\delta_t^2 + 512L_1\sqrt{d}\|\mathbf{H}_{t+1} - \mathbf{H}_t\|_F.$$

By summing the above inequality from $t = 0$ to $T-1$, we get

$$\sum_{t=0}^{T-1} \ell_t(\mathbf{B}_t) \leq 64L_1^2\|\mathbf{W}_0 - \hat{\mathbf{H}}_0\|_F^2 + 4\sum_{t=0}^{T-1}\ell_t(\mathbf{H}_t) + 2L_1^2\sum_{t=0}^{T-1}\delta_t^2 + 512L_1\sqrt{d}\sum_{t=0}^{T-1}\|\mathbf{H}_{t+1} - \mathbf{H}_t\|_F.$$

Finally, we use the fact that $\mathbf{W}_0 \triangleq \frac{2}{L_1}(\mathbf{B}_0 - \frac{L_1}{2}\mathbf{I})$, and $\hat{\mathbf{H}}_0 \triangleq \frac{2}{L_1}(\mathbf{H}_0 - \frac{L_1}{2}\mathbf{I})$ to obtain Lemma 12. $\quad\square$

### D.3 Proof of Lemma 13

By Assumption 2, we have $\|\mathbf{w}_t - \mathbf{H}_t\mathbf{s}_t\| = \|\nabla f(\tilde{\mathbf{x}}_{t+1}) - \nabla f(\mathbf{y}_t) - \nabla f(\mathbf{y}_t)(\tilde{\mathbf{x}}_{t+1} - \mathbf{y}_t)\| \leq \frac{L_2}{2}\|\tilde{\mathbf{x}}_{t+1} - \mathbf{y}_t\|^2$. Thus,

$$\ell_t(\mathbf{H}_t) = \frac{\|\mathbf{w}_t - \mathbf{H}_t\mathbf{s}_t\|^2}{\|\mathbf{s}_t\|^2} \leq \frac{L_2^2}{4}\|\tilde{\mathbf{x}}_{t+1} - \mathbf{y}_t\|^2 \leq \frac{(1+\alpha_1)^2 L_2^2}{4\beta^2(1-\alpha_1)^2}\|\hat{\mathbf{x}}_{t+1} - \mathbf{y}_t\|^2,$$

where we used Lemma 1 in the last inequality. Also, Since $a_k \geq \eta_k$ for all $k \geq 0$, by (21) we get

$$\sum_{k=0}^{N-1} \|\hat{\mathbf{x}}_{k+1} - \mathbf{y}_k\|^2 \leq \sum_{k=0}^{N-1} \frac{a_k^2}{\eta_k^2}\|\hat{\mathbf{x}}_{k+1} - \mathbf{y}_k\|^2 \leq \frac{1}{1-\sigma^2}\|\mathbf{z}_0 - \mathbf{x}^*\|^2.$$

Hence, we have

$$\sum_{t=0}^{T-1} \ell_t(\mathbf{H}_t) \leq \frac{(1+\alpha_1)^2 L_2^2}{4\beta^2(1-\alpha_1)^2}\sum_{k\in\mathcal{B}}\|\hat{\mathbf{x}}_{k+1} - \mathbf{y}_k\|^2 \leq \frac{(1+\alpha_1)^2 L_2^2}{4\beta^2(1-\alpha_1)^2}\sum_{k=0}^{N-1}\|\hat{\mathbf{x}}_{k+1} - \mathbf{y}_k\|^2$$
$$\leq \frac{(1+\alpha_1)^2 L_2^2\|\mathbf{z}_0 - \mathbf{x}^*\|^2}{4\beta^2(1-\alpha_1)^2(1-\sigma^2)},$$

which proves the first inequality in (72).

Furthermore, by Assumption 2, we have

$$\|\mathbf{H}_{t+1} - \mathbf{H}_t\|_F = \|\nabla^2 f(\mathbf{y}_{t+1}) - \nabla^2 f(\mathbf{y}_t)\|_F \leq \sqrt{d}\|\nabla^2 f(\mathbf{y}_{t+1}) - \nabla^2 f(\mathbf{y}_t)\|_{\text{op}} \leq \sqrt{d}L_2\|\mathbf{y}_{t+1} - \mathbf{y}_t\|.$$

Hence, by using the triangle inequality, we can bound

$$\sum_{t=0}^{T-1} \|\mathbf{H}_{t+1} - \mathbf{H}_t\|_F \leq \sqrt{d}L_2\sum_{k=0}^{N-1}\|\mathbf{y}_{k+1} - \mathbf{y}_k\| \leq \sqrt{d}L_2 C_2\Big(1 + \log\frac{A_N}{A_1}\Big)\|\mathbf{z}_0 - \mathbf{x}^*\|,$$

where we used Lemma 8 in the last inequality.

## D.4 Proof of Lemma 14

Before presenting the proof of Lemma 14, we start with a helper lemma that shows a lower bound on $A_1$.

**Lemma 18.** *We have $A_1 = \hat{\eta}_0 \geq \min\{\sigma_0, \frac{\alpha_2 \beta}{L_1}\}$.*

*Proof.* The equality $A_1 = \hat{\eta}_0$ is shown in the proof of Lemma 3. To show the lower bound on $\hat{\eta}_0$, we use Lemma 1 and separate two cases. If $0 \notin \mathcal{B}$, then we have $\hat{\eta}_0 = \eta_0 = \sigma_0$. Otherwise, if $0 \in \mathcal{B}$, then we have

$$\hat{\eta}_0 \geq \frac{\alpha_2 \beta \|\tilde{\mathbf{x}}_1 - \mathbf{y}_0\|}{\|\nabla f(\tilde{\mathbf{x}}_1) - \nabla f(\mathbf{y}_0) - \mathbf{B}_k(\tilde{\mathbf{x}}_1 - \mathbf{y}_0)\|}.$$

Moreover, as shown in the proof of Lemma 11, we have $\|\nabla f(\tilde{\mathbf{x}}_1) - \nabla f(\mathbf{y}_0) - \mathbf{B}_k(\tilde{\mathbf{x}}_1 - \mathbf{y}_0)\| \leq L_1 \|\tilde{\mathbf{x}}_1 - \mathbf{y}_0\|$, which further implies that $\hat{\eta}_0 \geq \frac{\alpha_2 \beta}{L_1}$. This completes the proof. $\qquad \square$

We combine Lemma 12 and Lemma 13 to get

$$\sum_{k \in \mathcal{B}} \frac{\|\mathbf{w}_k - \mathbf{B}_k \mathbf{s}_k\|^2}{\|\mathbf{s}_k\|^2} = \sum_{t=0}^{T-1} \ell_t(\mathbf{B}_t) \leq 256 \|\mathbf{B}_0 - \mathbf{H}_0\|_F^2 + C_3 L_2^2 \|\mathbf{z}_0 - \mathbf{x}^*\|^2 + 2L_1^2 \sum_{t=0}^{T-1} \delta_t^2$$

$$+ 512 C_2 L_1 L_2 d \left(1 + \log \frac{A_N}{A_1}\right) \|\mathbf{z}_0 - \mathbf{x}^*\|.$$

Since $\delta_t = 1/(\sqrt{t+2}\ln(t+2))$, we have

$$\sum_{t=0}^{T-1} \delta_t^2 = \sum_{t=2}^{T+1} \frac{1}{t \ln^2 t} \leq \frac{1}{2 \ln^2 2} + \int_2^{T+1} \frac{1}{t \ln^2 t} dt = \frac{1}{2 \ln^2 2} + \frac{1}{\ln 2} - \frac{1}{\ln(T+1)} \leq 2.5.$$

Hence, it further follows from (14) and Lemma 2 that

$$\frac{N^5}{A_N^2} \leq 4C_1^2 \sum_{k=0}^{N-1} \frac{1}{\hat{\eta}_k^2}$$

$$\leq \frac{4C_1^2(2 - \beta^2)}{(1 - \beta^2)\sigma_0^2} + \frac{4C_1^2(2 - \beta^2)}{(1 - \beta^2)\alpha_2^2\beta^2} \sum_{k \in \mathcal{B}} \frac{\|\mathbf{w}_k - \mathbf{B}_k \mathbf{s}_k\|^2}{\|\mathbf{s}_k\|^2}$$

$$\leq \frac{C_6}{\sigma_0^2} + C_7 L_1^2 + C_8 \|\mathbf{B}_0 - \mathbf{H}_0\|_F^2 + C_9 L_2^2 \|\mathbf{z}_0 - \mathbf{x}^*\|^2$$

$$+ C_{10} L_1 L_2 d \left(1 + \log \frac{A_N}{A_1}\right) \|\mathbf{z}_0 - \mathbf{x}^*\|. \tag{80}$$

To simplify the notation, define

$$M = \frac{C_6}{\sigma_0^2} + C_7 L_1^2 + C_8 \|\mathbf{B}_0 - \mathbf{H}_0\|_F^2 + C_9 L_2^2 \|\mathbf{z}_0 - \mathbf{x}^*\|^2 + C_{10} L_1 L_2 d \|\mathbf{z}_0 - \mathbf{x}^*\|,$$

and the inequality in (80) becomes $\frac{N^5}{A_N^2} \leq M + C_{10} L_1 L_2 d \|\mathbf{z}_0 - \mathbf{x}^*\| \log \frac{A_N}{A_1}$. Let $A_N^*$ be the number that achieves the equality

$$\frac{N^5}{(A_N^*)^2} = M + C_{10} L_1 L_2 d \|\mathbf{z}_0 - \mathbf{x}^*\| \log \frac{A_N^*}{A_1},$$

and we can see that $A_N \geq A_N^*$. Thus, we instead try to construct a lower bound on $A_N^*$. If $A_N^* \leq A_1$, then $\log(A_N^*/A_1) \leq 0$ and furthermore

$$\frac{N^5}{(A_N^*)^2} \leq M \quad \Rightarrow \quad \frac{1}{A_N} \leq \frac{\sqrt{M}}{N^{2.5}}. \tag{81}$$

Otherwise, assume that $A_N^* > A_1$. Then $\log(A_N^*/A_1) > 0$ and we first show an upper bound on $A_N^*$:

$$\frac{N^5}{(A_N^*)^2} = M + C_{10} L_1 L_2 d \|\mathbf{z}_0 - \mathbf{x}^*\| \log \frac{A_N^*}{A_1} \geq M \quad \Rightarrow \quad A_N^* \leq \frac{1}{\sqrt{M}} N^{2.5}.$$

This in turn leads to a lower bound on $A_N^*$:

$$\frac{N^5}{(A_N^*)^2} = M + C_{10}L_1L_2d\|\mathbf{z}_0 - \mathbf{x}^*\| \log \frac{A_N^*}{A_1} \leq M + C_{10}L_1L_2d\|\mathbf{z}_0 - \mathbf{x}^*\| \log \left( \frac{\max\{\frac{L_1}{\alpha_2\beta}, \frac{1}{\sigma_0}\}N^{2.5}}{\sqrt{M}} \right),$$

where we also used the fact that $A_1 \geq \min\{\sigma_0, \frac{\alpha_2\beta}{L_1}\}$ (cf. Lemma 18). Thus, we get

$$\frac{1}{A_N} \leq \frac{1}{A_N^*} \leq \frac{1}{N^{2.5}} \left( M + C_{10}L_1L_2d\|\mathbf{z}_0 - \mathbf{x}^*\| \log \left( \frac{\max\{\frac{L_1}{\alpha_2\beta}, \frac{1}{\sigma_0}\}N^{2.5}}{\sqrt{M}} \right) \right)^{\frac{1}{2}}. \tag{82}$$

Combining both cases in (81) and (82), we conclude the proof of Lemma 14.

**Subroutine 3** LinearSolver($\mathbf{A}, \mathbf{b}; \alpha$)

1: **Input:** $\mathbf{A} \in \mathbb{S}_+^d, \mathbf{b} \in \mathbb{R}^d, 0 < \alpha < 1$
2: **Initialize:** $\mathbf{s}_0 \leftarrow 0, \mathbf{r}_0 \leftarrow \mathbf{b} - \mathbf{A}\mathbf{s}_0, \mathbf{p}_0 \leftarrow \mathbf{r}_0$
3: **for** $k = 0, 1, \ldots$ **do**
4:     **if** $\|\mathbf{r}_k\|_2 \leq \alpha\|\mathbf{s}_k\|_2$ **then**
5:         **Return** $\mathbf{s}_k$
6:     **end if**
7:     $\alpha_k \leftarrow \langle \mathbf{r}_k, \mathbf{A}\mathbf{r}_k \rangle / \langle \mathbf{A}\mathbf{p}_k, \mathbf{A}\mathbf{p}_k \rangle$
8:     $\mathbf{s}_{k+1} \leftarrow \mathbf{s}_k + \alpha_k \mathbf{p}_k$
9:     $\mathbf{r}_{k+1} \leftarrow \mathbf{r}_k - \alpha_k \mathbf{A}\mathbf{p}_k$
10:    Compute and store $\mathbf{A}\mathbf{r}_{k+1}$
11:    $\beta_k \leftarrow \langle \mathbf{r}_{k+1}, \mathbf{A}\mathbf{r}_{k+1} \rangle / \langle \mathbf{r}_k, \mathbf{A}\mathbf{r}_k \rangle$
12:    $\mathbf{p}_{k+1} \leftarrow \mathbf{r}_{k+1} + \beta_k \mathbf{p}_k$
13:    Compute and store $\mathbf{A}\mathbf{p}_{k+1} \leftarrow \mathbf{A}\mathbf{r}_{k+1} + \beta_k \mathbf{A}\mathbf{p}_k$
14: **end for**

# E    Characterizing the Computational Cost

In this section, we first specify the implementation details of the LinearSolver oracle in Definition 1 and the SEP oracle in Definition 2. Then in Section E.3, we present the proof of Theorem 2.

## E.1    Implementation of the LinearSolver **Oracle**

We implement the LinearSolver oracle by running the conjugate residual (CR) method [48] to solve the linear system $\mathbf{A}\mathbf{s} = \mathbf{b}$. In particular, we initialize the CR method with $\mathbf{s}_0 = 0$ and returns the iterate $\mathbf{s}_k$ once we achieve $\|\mathbf{A}\mathbf{s}_k - \mathbf{b}\| \leq \alpha\|\mathbf{s}_k\|$. The following lemma provides the convergence guarantee of the CR method, which will be later used in the proof of Theorem 2.

**Lemma 19** ([56, Chapter 12.4]). *Let $\mathbf{s}^*$ be any optimal solution of $\mathbf{A}\mathbf{s}^* = \mathbf{b}$ and let $\{\mathbf{s}_k\}$ be the iterates generated by Subroutine 3. Then we have*

$$\|\mathbf{r}_k\|_2 = \|\mathbf{A}\mathbf{s}_k - \mathbf{b}\|_2 \leq \frac{\lambda_{\max}(\mathbf{A})\|\mathbf{s}^*\|_2}{(k+1)^2}.$$

## E.2    Implementation of SEP **Oracle**

We implement the SEP oracle in Definition 2 based on the classical Lanczos method with a random start, where the initial vector is chosen randomly and uniformly from the unit sphere (see, e.g., [57, 58]). For completeness, the full algorithm is described in Subroutine 4.

To prove the correctness of our algorithm, we first recall a classical result in [51] on the convergence behavior of the Lanczos method.

**Proposition 3** ([51, Theorem 4.2]). *Consider a symmetric matrix $\mathbf{W}$ and let $\lambda_1(\mathbf{W})$ and $\lambda_d(\mathbf{W})$ denote its largest and smallest eigenvalues, respectively. Then after $k$ iterations of the Lanczos method with a random start, we find unit vectors $\mathbf{u}^{(1)}$ and $\mathbf{u}^{(d)}$ such that*

$$\mathbb{P}(\langle \mathbf{W}\mathbf{u}^{(1)}, \mathbf{u}^{(1)} \rangle \leq \lambda_1(\mathbf{W}) - \epsilon(\lambda_1(\mathbf{W}) - \lambda_d(\mathbf{W}))) \leq 1.648\sqrt{d}e^{-\sqrt{\epsilon}(2k-1)},$$
$$\mathbb{P}(\langle \mathbf{W}\mathbf{u}^{(d)}, \mathbf{u}^{(d)} \rangle \geq \lambda_d(\mathbf{W}) + \epsilon(\lambda_1(\mathbf{W}) - \lambda_d(\mathbf{W}))) \leq 1.648\sqrt{d}e^{-\sqrt{\epsilon}(2k-1)},$$

*As a corollary, to ensure that, with probability at least $1 - q$,*

$\langle \mathbf{W}\mathbf{u}^{(1)}, \mathbf{u}^{(1)} \rangle > \lambda_1(\mathbf{W}) - \epsilon(\lambda_1(\mathbf{W}) - \lambda_d(\mathbf{W}))$ *and* $\langle \mathbf{W}\mathbf{u}^{(d)}, \mathbf{u}^{(d)} \rangle < \lambda_n(\mathbf{W}) + \epsilon(\lambda_1(\mathbf{W}) - \lambda_d(\mathbf{W}))$,

*the number of iterations can be bounded by* $\lceil \frac{1}{4}\epsilon^{-1/2}\log(11d/q^2) + \frac{1}{2} \rceil$.

**Lemma 20.** *Let $\gamma$ and $\mathbf{S}$ be the output of SEP$(\mathbf{W}; \delta, q)$ in Subroutine 4. Then with probability at least $1 - q$, they satisfy one of the following properties:*

- *Case I: $\gamma \leq 1$, then we have $\|\mathbf{W}\|_{\mathrm{op}} \leq 1$;*
- *Case II: $\gamma > 1$, then we have $\|\mathbf{W}/\gamma\|_{\mathrm{op}} \leq 1$, $\|\mathbf{S}\|_F = 3$ and $\langle \mathbf{S}, \mathbf{W} - \hat{\mathbf{B}} \rangle \geq \gamma - 1$ for any $\hat{\mathbf{B}}$ such that $\|\hat{\mathbf{B}}\|_{\mathrm{op}} \leq 1$.*

---

**Subroutine 4** SEP$(\mathbf{W}; \delta, q)$

---

1: **Input:** $\mathbf{W} \in \mathbb{S}^d$, $\delta > 0$, $q \in (0, 1)$
2: Set the number of iterations $N_1 \leftarrow \min\left\{\left\lceil \log \frac{11d}{q^2} + \frac{1}{2} \right\rceil, d\right\}$
3: Run Lanczos method with a random start for $N_1$ iterations to get $\mathbf{u}^{(1)}$ and $\mathbf{u}^{(d)}$ (cf. Proposition 3)
4: Set $\hat{\lambda}_1 \leftarrow \langle \mathbf{W}\mathbf{u}^{(1)}, \mathbf{u}^{(1)} \rangle$ and $\hat{\lambda}_d \leftarrow \langle \mathbf{W}\mathbf{u}^{(d)}, \mathbf{u}^{(d)} \rangle$
5: Set $\hat{\lambda}_{\max} \leftarrow \max\{\hat{\lambda}_1, -\hat{\lambda}_d\}$
6: **if** $\hat{\lambda}_{\max} \leq 1/2$ **then**     # *Case I: $\gamma \leq 1$, which implies $\|\mathbf{W}\|_{\mathrm{op}} \leq 1$*
7:     Return $\gamma = 2\hat{\lambda}_{\max}$ and $\mathbf{S} = 0$
8: **else if** $\hat{\lambda}_{\max} \geq 2$ **then**     # *Case II: $\gamma > 1$ and $\mathbf{S}$ defines a separating hyperplane*
9:     **if** $\hat{\lambda}_1 > -\hat{\lambda}_d$ **then**
10:         Return $\gamma = 2\hat{\lambda}_{\max}$ and $\mathbf{S} = 3\mathbf{u}^{(1)}(\mathbf{u}^{(1)})^\top$
11:     **else**
12:         Return $\gamma = 2\hat{\lambda}_{\max}$ and $\mathbf{S} = -3\mathbf{u}^{(d)}(\mathbf{u}^{(d)})^\top$
13:     **end if**
14: **else**    # $\frac{1}{2} < \hat{\lambda}_{\max} < 2$
15:     Set the number of iterations $N_2 \leftarrow \min\left\{\left\lceil \frac{1}{4\sqrt{2\delta}} \log \frac{11d}{q^2} + \frac{1}{2} \right\rceil, d\right\}$
16:     Run Lanczos method with a random start for $N_2$ iterations to get $\tilde{\mathbf{u}}^{(1)}$ and $\tilde{\mathbf{u}}^{(d)}$ (cf. Proposition 3)
17:     Set $\tilde{\lambda}_1 \leftarrow \langle \mathbf{W}\tilde{\mathbf{u}}^{(1)}, \tilde{\mathbf{u}}^{(1)} \rangle$ and $\tilde{\lambda}_d \leftarrow \langle \mathbf{W}\tilde{\mathbf{u}}^{(d)}, \tilde{\mathbf{u}}^{(d)} \rangle$
18:     Set $\tilde{\lambda}_{\max} = \max\{\tilde{\lambda}_1, -\tilde{\lambda}_d\}$
19:     **if** $\tilde{\lambda}_{\max} \leq 1 - \delta$ **then**
20:         Return $\gamma = \tilde{\lambda}_{\max} + \delta$ and $\mathbf{S} = 0$
21:     **else if** $\tilde{\lambda}_1 \geq -\tilde{\lambda}_d$ **then**
22:         Return $\gamma = \tilde{\lambda}_{\max} + \delta$ and $\mathbf{S} = \tilde{\mathbf{u}}^{(1)}(\tilde{\mathbf{u}}^{(1)})^\top$
23:     **else**
24:         Return $\gamma = \tilde{\lambda}_{\max} + \delta$ and $\mathbf{S} = -\tilde{\mathbf{u}}^{(d)}(\tilde{\mathbf{u}}^{(d)})^\top$
25:     **end if**
26: **end if**

---

*Proof.* Note that in Subroutine 4, we first run the Lanczos method for $\left\lceil \epsilon^{-1/2} \log \frac{11d}{q^2} + \frac{1}{2} \right\rceil$ iterations, where $\epsilon = \frac{1}{4}$. Thus, by Proposition 3, with probability at least $1 - q/2$ we have

$$\hat{\lambda}_1 \triangleq \langle \mathbf{W}\mathbf{u}^{(1)}, \mathbf{u}^{(1)} \rangle \geq \lambda_1(\mathbf{W}) - \frac{1}{4}(\lambda_1(\mathbf{W}) - \lambda_d(\mathbf{W})), \tag{83}$$

$$\hat{\lambda}_d \triangleq \langle \mathbf{W}\mathbf{u}^{(d)}, \mathbf{u}^{(d)} \rangle \leq \lambda_d(\mathbf{W}) + \frac{1}{4}(\lambda_1(\mathbf{W}) - \lambda_d(\mathbf{W})). \tag{84}$$

Combining (83) and (84), we get

$$\frac{1}{2}(\lambda_1(\mathbf{W}) - \lambda_d(\mathbf{W})) \leq \hat{\lambda}_1 - \hat{\lambda}_d \quad \Rightarrow \quad \lambda_1(\mathbf{W}) - \lambda_d(\mathbf{W}) \leq 2(\hat{\lambda}_1 - \hat{\lambda}_d).$$

By plugging the above inequality back into (83) and (84), we further have

$$\lambda_1(\mathbf{W}) \leq \hat{\lambda}_1 + \frac{1}{4}(\lambda_1(\mathbf{W}) - \lambda_d(\mathbf{W})) \leq \hat{\lambda}_1 + \frac{1}{2}(\hat{\lambda}_1 - \hat{\lambda}_d), \tag{85}$$

$$\lambda_d(\mathbf{W}) \geq \hat{\lambda}_d - \frac{1}{4}(\lambda_1(\mathbf{W}) - \lambda_d(\mathbf{W})) \geq \hat{\lambda}_d - \frac{1}{2}(\hat{\lambda}_1 - \hat{\lambda}_d). \tag{86}$$

Let $\hat{\lambda}_{\max} = \max\{\hat{\lambda}_1, -\hat{\lambda}_d\}$. By (85) and (86), we can further bound the eigenvalues of $\mathbf{W}$ by

$$\lambda_1(\mathbf{W}) \leq \hat{\lambda}_{\max} + \frac{1}{2} \cdot 2\hat{\lambda}_{\max} = 2\hat{\lambda}_{\max} \quad \text{and} \quad \lambda_d(\mathbf{W}) \geq -\hat{\lambda}_{\max} - \frac{1}{2} \cdot 2\hat{\lambda}_{\max} = -2\hat{\lambda}_{\max}.$$

Hence, we can see that $\|\mathbf{W}\|_{\mathrm{op}} = \max\{\lambda_1(\mathbf{W}), -\lambda_d(\mathbf{W})\} \leq 2\hat{\lambda}_{\max}$. Now we distinguish three cases.

(a) If $\hat{\lambda}_{\max} \leq \frac{1}{2}$, then we are in **Case I** and the ExtEvec oracle outputs $\gamma = 2\hat{\lambda}_{\max} \leq 1$ and $\mathbf{S} = 0$. In this case, we indeed have $\|\mathbf{W}\|_{\mathrm{op}} \leq \gamma \leq 1$.

(b) If $\hat{\lambda}_{\max} \geq 2$, then we are in **Case II**. In addition, if $\hat{\lambda}_1 \geq -\hat{\lambda}_d$, then the ExtEvec oracle returns $\gamma = 2\hat{\lambda}_{\max}$ and $\mathbf{S} = 3\mathbf{u}^{(1)}(\mathbf{u}^{(1)})^\top$. Similarly, if $-\hat{\lambda}_d > \hat{\lambda}_1$, then the ExtEvec oracle returns $\gamma = 2\hat{\lambda}_{\max}$ and $\mathbf{S} = -3\mathbf{u}^{(d)}(\mathbf{u}^{(d)})^\top$. Without loss of generality, consider the case where $\hat{\lambda}_1 \geq -\hat{\lambda}_d$. Since $\|\mathbf{W}\|_{\mathrm{op}} \leq 2\hat{\lambda}_{\max} = \gamma$, we have $\|\mathbf{W}/\gamma\|_{\mathrm{op}} \leq 1$. Also, since $\mathbf{u}_1$ is a unit vector, we have $\|\mathbf{S}\|_F = 3\|\mathbf{u}^{(1)}\|^2 = 3$. Finally, for any $\hat{\mathbf{B}}$ such that $\|\hat{\mathbf{B}}\|_{\mathrm{op}} \leq 1$, we have

$$\langle \mathbf{S}, \mathbf{W} - \hat{\mathbf{B}} \rangle = 3(\mathbf{u}^{(1)})^\top \mathbf{W} \mathbf{u}^{(1)} - 3(\mathbf{u}^{(1)})^\top \hat{\mathbf{B}} \mathbf{u}^{(1)} \geq 3\hat{\lambda}_{\max} - 3 \geq 2\hat{\lambda}_{\max} - 1 = \gamma - 1,$$

where we used the fact that $\hat{\lambda}_{\max} \geq 2$ in the last inequality.

(c) If $\frac{1}{2} < \hat{\lambda}_{\max} < 2$, we continue to run the Lanczos method for a total number of $\left\lceil \frac{1}{4}\epsilon^{-1/2} \log \frac{11d}{q^2} + \frac{1}{2} \right\rceil$ iterations, where $\epsilon = \frac{1}{8}\delta$. Thus, by Proposition 3, with probability at least $1 - q/2$ we have

$$\tilde{\lambda}_1 \triangleq \langle \mathbf{W}\tilde{\mathbf{u}}^{(1)}, \tilde{\mathbf{u}}^{(1)} \rangle \geq \lambda_1(\mathbf{W}) - \frac{1}{8}\delta(\lambda_1(\mathbf{W}) - \lambda_d(\mathbf{W})), \tag{87}$$

$$\tilde{\lambda}_d \triangleq \langle \mathbf{W}\tilde{\mathbf{u}}^{(d)}, \tilde{\mathbf{u}}^{(d)} \rangle \leq \lambda_d(\mathbf{W}) + \frac{1}{8}\delta(\lambda_1(\mathbf{W}) - \lambda_d(\mathbf{W})). \tag{88}$$

Let $\tilde{\lambda}_{\max} = \max\{\tilde{\lambda}_1, -\tilde{\lambda}_d\}$. Since we have $\lambda_1(\mathbf{W}) \leq 2\hat{\lambda}_{\max} \leq 4$ and $\lambda_d(\mathbf{W}) \geq -2\hat{\lambda}_{\max} \geq -4$, the above implies that $\tilde{\lambda}_1 \geq \lambda_1(\mathbf{W}) - \delta$ and $\tilde{\lambda}_d \leq \lambda_d(\mathbf{W}) + \delta$. Hence, we can see that $\|\mathbf{W}\|_{\mathrm{op}} = \max\{\lambda_1(\mathbf{W}), -\lambda_d(\mathbf{W})\} \leq \hat{\lambda}_{\max} + \delta$. We further consider two subcases.

(c1) If $\tilde{\lambda}_{\max} \leq 1 - \delta$, then we are in **Case I** and the ExtEvec oracle outputs $\gamma = \tilde{\lambda}_{\max} + \delta$ and $\mathbf{S} = 0$. In this case, we indeed have $\|\mathbf{W}\|_{\mathrm{op}} \leq \gamma \leq 1$.

(c2) If $\tilde{\lambda}_{\max} > 1 - \delta$, then we are in **Case II**. In addition, if $\tilde{\lambda}_1 \geq -\tilde{\lambda}_d$, then the ExtEvec oracle returns $\gamma = \tilde{\lambda}_{\max} + \delta$ and $\mathbf{S} = \tilde{\mathbf{u}}^{(1)}(\tilde{\mathbf{u}}^{(1)})^\top$. Similarly, if $-\tilde{\lambda}_d > \tilde{\lambda}_1$, then the ExtEvec oracle returns $\gamma = \tilde{\lambda}_{\max} + \delta$ and $\mathbf{S} = -\tilde{\mathbf{u}}^{(d)}(\tilde{\mathbf{u}}^{(d)})^\top$. Without loss of generality, consider the case where $\tilde{\lambda}_1 \geq -\tilde{\lambda}_d$. Since $\|\mathbf{W}\|_{\mathrm{op}} \leq \tilde{\lambda}_{\max} + \delta = \gamma$, we have $\|\mathbf{W}/\gamma\|_{\mathrm{op}} \leq 1$. Also, since $\tilde{\mathbf{u}}^{(1)}$ is a unit vector, we have $\|\mathbf{S}\|_F = \|\tilde{\mathbf{u}}^{(1)}\|^2 = 1$. Finally, for any $\hat{\mathbf{B}}$ such that $\|\hat{\mathbf{B}}\|_{\mathrm{op}} \leq 1$, we have

$$\langle \mathbf{S}, \mathbf{W} - \hat{\mathbf{B}} \rangle = (\tilde{\mathbf{u}}^{(1)})^\top \mathbf{W} \tilde{\mathbf{u}}^{(1)} - (\tilde{\mathbf{u}}^{(1)})^\top \hat{\mathbf{B}} \tilde{\mathbf{u}}^{(1)} \geq \tilde{\lambda}_{\max} - 1 = \gamma - 1 - \delta.$$

This completes the proof. $\qquad\square$

### E.3 Proof of Theorem 2

We divide the proof of Theorem 2 into the following three lemmas.

**Lemma 21.** *If we run Algorithm 1 as specified in Theorem 1 for $N$ iterations, then the total number of line search steps can be bounded by $2N + \log_{1/\beta}(\sigma_0 L_1/\alpha_2)$. As a corollary, the total number of gradient queries is bounded by $3N_\epsilon + \log_{1/\beta}(\frac{\sigma_0 L_1}{\alpha_2})$.*

*Proof.* In our backtracking scheme, the number of steps in each iteration is given by $\log_{1/\beta}(\eta_k/\hat{\eta}_k) + 1$. Also note that $\eta_{k+1} \leq \hat{\eta}_k/\beta$ for all $k \geq 0$. Thus, we have

$$\sum_{k=0}^{N-1} \left( \log_{1/\beta} \frac{\eta_k}{\hat{\eta}_k} + 1 \right) = N + \log_{1/\beta} \frac{\sigma_0}{\hat{\eta}_0} + \sum_{k=0}^{N-2} \log_{1/\beta} \frac{\eta_{k+1}}{\hat{\eta}_{k+1}}$$

$$\leq N + \log_{1/\beta} \frac{\sigma_0}{\hat{\eta}_0} + \sum_{k=0}^{N-2} \left( \log_{1/\beta} \frac{\hat{\eta}_k}{\hat{\eta}_{k+1}} + 1 \right)$$

$$\leq 2N - 1 + \log_{1/\beta} \frac{\sigma_0}{\hat{\eta}_{N-1}}$$

Furthermore, since $\hat{\eta}_k \geq \alpha_2 \beta / L_1$ for all $k \geq 0$, we arrive at the conclusion. $\qquad\square$

**Lemma 22.** *The total number of matrix-vector product evaluations in the* LinearSolver *oracle is bounded by* $N_\epsilon + C_{11}\sqrt{\sigma_0 L_1} + C_{12}\sqrt{\frac{L_1\|\mathbf{z}_0 - \mathbf{x}^*\|^2}{2\epsilon}}$, *where $C_{11}$ and $C_{12}$ are absolute constants.*

*Proof.* Our proof loosely follows the strategy in [32]. We first bound the number of steps required by Subroutine 3 before it terminates.

**Lemma 23.** *Suppose* $\mathbf{A} \succeq \mathbf{I}$*. Then Subroutine 3 terminates after at most* $\left\lceil \sqrt{\frac{\alpha+1}{\alpha}\lambda_{\max}(\mathbf{A})} - 1 \right\rceil$ *iterations.*

*Proof.* Note that $\|\mathbf{s}_k\|_2 \geq \|\mathbf{s}^*\|_2 - \|\mathbf{s}_k - \mathbf{s}^*\|_2$. Also, since $\mathbf{A} \succeq \mathbf{I}$, we have $\|\mathbf{s}_k - \mathbf{s}^*\|_2 \leq \|\mathbf{A}(\mathbf{s}_k - \mathbf{s}^*)\|_2 = \|\mathbf{r}_k\|_2$. Therefore, we have

$$\|\mathbf{r}_k\|_2 \leq \alpha\|\mathbf{s}_k\|_2 \quad \Leftarrow \quad \|\mathbf{r}_k\|_2 \leq \alpha\|\mathbf{s}^*\|_2 - \alpha\|\mathbf{r}_k\|_2 \quad \Leftarrow \quad \|\mathbf{r}_k\|_2 \leq \frac{\alpha}{\alpha+1}\|\mathbf{s}^*\|_2.$$

By using Lemma 19, we only need $k \geq \sqrt{\frac{\alpha+1}{\alpha}\lambda_{\max}(\mathbf{A})} - 1$ to achieve $\|\mathbf{A}\mathbf{s}_k - \mathbf{b}\| \leq \alpha\|\mathbf{s}_k\|$. $\quad\square$

Moreover, when the step size is smaller enough, we can show that Subroutine 3 will terminate in one iteration.

**Lemma 24.** *Let* $\mathbf{A} = \mathbf{I} + \eta\mathbf{B}$*. When* $\eta \leq \frac{\alpha}{2L_1}$*, Algorithm 3 terminates in one iteration.*

*Proof.* From the update rule of Subroutine 3, we can compute that $\mathbf{s}_1 = \frac{\mathbf{b}^\top \mathbf{A}\mathbf{b}}{\|\mathbf{A}\mathbf{b}\|_2^2}\mathbf{b}$, which implies

$$\|\mathbf{s}_1\| = \|\mathbf{b}\| \cdot \frac{\|\mathbf{A}^{1/2}\mathbf{b}\|^2}{(\mathbf{A}^{1/2}\mathbf{b})^\top \mathbf{A}(\mathbf{A}^{1/2}\mathbf{b})} \geq \frac{\|\mathbf{b}\|}{\lambda_{\max}(\mathbf{A})} \geq \frac{\|\mathbf{b}\|}{1 + \eta L_1}.$$

On the other hand, we also have

$$\|\mathbf{r}_1\| \leq \|\mathbf{A}\mathbf{b} - \mathbf{b}\| = \eta\|\mathbf{B}\mathbf{b}\| \leq \eta L_1\|\mathbf{b}\|.$$

Moreover, when $\eta \leq \frac{\alpha}{2L_1}$, we have $\eta L_1 \leq \frac{\alpha}{1+\eta L_1}$, which implies that $\|\mathbf{r}_1\| \leq \alpha\|\mathbf{s}_1\|$. $\quad\square$

Now we upper bound the total number of matrix-vector products in Algorithm 1. Note that at the $k$-th iteration, we use the LinearSolver oracle with $\mathbf{A} = \mathbf{I} + \eta_+\mathbf{B}_k$ where $\eta_+ = \eta_k\beta^i$. We can store the vector $\mathbf{B}_k\mathbf{b}$ at the beginning and reuse it to compute $\mathbf{s}_1$ when the step size $\eta_+ < \frac{\alpha_1}{2L_1}$. And when $\beta^i\eta_k L_1 \geq \frac{\alpha_1}{2}$, it holds that

$$1 + \beta^i\eta_k L_1 \leq \frac{\alpha_1 + 2}{\alpha_1}\beta^i\eta_k L_1.$$

Thus, at the $k$-th iteration, the number of matrix-vector products can be bounded by

$$\mathsf{MV}_k \leq 1 + \sum_{i \geq 0, \eta_k\beta^i \geq \frac{\alpha_1}{2L_1}} \sqrt{\frac{\alpha_1 + 1}{\alpha_1}(1 + \eta_k\beta^i L_1)}$$

$$\leq 1 + \sum_{i \geq 0, \eta_k\beta^i \geq \frac{\alpha_1}{2L_1}} \frac{\alpha_1 + 2}{\alpha_1}\sqrt{\beta^i\eta_k L_1}$$

$$\leq 1 + \frac{\alpha_1 + 2}{\alpha_1}\frac{1}{1 - \sqrt{\beta}}\sqrt{\eta_k L_1}.$$

Furthermore, we can bound that

$$\sum_{k=0}^{N-1}\sqrt{\eta_k} \leq \sqrt{\sigma_0} + \sum_{k=1}^{N-1}\sqrt{\eta_k} \leq \sqrt{\sigma_0} + \frac{1}{\sqrt{\beta}}\sum_{k=0}^{N-2}\sqrt{\widehat{\eta}_k} \leq \sqrt{\sigma_0} + \frac{2(2 - \sqrt{\beta})}{\sqrt{\beta}(1 - \sqrt{\beta})}\sqrt{A_{N-1}}$$

Note that $\epsilon < f(x_{N-1}) - f(\mathbf{x}^*) \leq \frac{\|\mathbf{z}_0 - \mathbf{x}^*\|^2}{2A_{N-1}}$. Hence, we have $A_{N-1} \leq \frac{\|\mathbf{z}_0 - \mathbf{x}^*\|^2}{2\epsilon}$. Thus, we can bound the total number of matrix-vector product evaluations by

$$\mathsf{MV} = \sum_{k=0}^{N_\epsilon - 1} \mathsf{MV}_k \leq N_\epsilon + \frac{\alpha_1 + 2}{\alpha_1} \frac{1}{1 - \sqrt{\beta}} \left( \sqrt{\sigma_0 L_1} + \frac{2(2 - \sqrt{\beta})}{\sqrt{\beta}(1 - \sqrt{\beta})} \sqrt{\frac{L_1 \|\mathbf{z}_0 - \mathbf{x}^*\|^2}{2\epsilon}} \right),$$

$$= N_\epsilon + C_{11} \sqrt{\sigma_0 L_1} + C_{12} \sqrt{\frac{L_1 \|\mathbf{z}_0 - \mathbf{x}^*\|^2}{2\epsilon}},$$

where we define $C_{11} = \frac{\alpha_1 + 2}{\alpha_1} \frac{1}{1 - \sqrt{\beta}}$ and $C_{12} = \frac{\alpha_1 + 2}{\alpha_1} \frac{1}{1 - \sqrt{\beta}} \frac{2(2 - \sqrt{\beta})}{\sqrt{\beta}(1 - \sqrt{\beta})}$. $\qquad\square$

**Lemma 25.** *The total number of matrix-vector product evaluations in the* SEP *oracle is bounded by* $\mathcal{O}\big(N_\epsilon^{1.25}(\log N_\epsilon)^{0.5} \log\big(\frac{\sqrt{d}N_\epsilon}{p}\big)\big)$.

*Proof.* Note that we have $N_t \leq \left\lceil \frac{1}{4\sqrt{2\delta_t}} \log \frac{44d}{q_t^2} + \frac{1}{2} \right\rceil$ in Subroutine 4, where $\delta_t = 1/(\sqrt{t + 2} \log(t + 2))$ and $q_t = p/(2.5(t + 1) \log^2(t + 1))$. Thus, we have

$$N = \sum_{t=0}^{T-1} N_t \leq \sum_{t=0}^{T-1} \frac{(t + 2)^{0.25} \log^{0.5}(t + 2)}{2\sqrt{2}} \log \frac{2.5\sqrt{44d}(t + 1) \log^2(t + 1)}{p} \qquad (89)$$

$$= \mathcal{O}\left( N_\epsilon^{1.25} \sqrt{\log N_\epsilon} \log \frac{\sqrt{d}N_\epsilon}{p} \right). \qquad (90)$$

$\qquad\square$

# F   Experiments

In our experiments, we consider the logistic regression problem and minimizing the log-sum-exp function. Below we provide more details about the data generation scheme as well as the implementation of Nesterov's accelerated gradient method, BFGS, and our proposed A-QPNE algorithm.

**Dataset generation.** In the first experiment of logistic regression, the dataset consists of $n$ data points $\{(\mathbf{a}_i, y_i)\}_{i=1}^n$, where $\mathbf{a}_i \in \mathbb{R}^d$ is the $i$-th feature vector and $\mathbf{y}_i \in \{-1, 1\}$ is its corresponding label. The labels $\{y_i\}_{i=1}^n$ are generated by

$$y_i = \mathrm{sign}(\langle \mathbf{a}_i^*, \mathbf{x}^* \rangle), \quad i = 1, 2, \ldots, n,$$

where $\mathbf{a}_i^* \in \mathbb{R}^{d-1}$ and $\mathbf{x}^* \in \mathbb{R}^{d-1}$ are the underlying true feature vector and the underlying true parameter, respectively. Moreover, each entry of $\mathbf{a}_i^*$ and $\mathbf{x}^*$ is drawn independently according to the standard normal distribution $\mathcal{N}(0, 1)$. Note that the true feature vectors $\{\mathbf{a}_i^*\}_{i=1}^n$ are not given in our dataset; instead, we generate $\{\mathbf{a}_i\}_{i=1}^n$ by adding noises and appending an extra dimension to $\{\mathbf{a}_i^*\}_{i=1}^n$. Specifically, we let $\mathbf{a}_i = [\mathbf{a}_i^* + \mathbf{n}_i + \mathbf{1}; 1]^\top \in \mathbb{R}^d$, where $\mathbf{n}_i \sim \mathcal{N}(0, \sigma^2 \mathbf{I})$ is the i.i.d. Gaussian noise vector and $\mathbf{1} \in \mathbb{R}^{d-1}$ denotes the all-one vector. In our experiment, we set $n = 2,000$, $d = 150$ and $\sigma = 0.8$.

In the second experiment of log-sum-exp function, we follow a similar procedure as [16] to generate the dataset $\{(\mathbf{a}_i, b_i)\}_{i=1}^n$, where $\mathbf{a}_i \in \mathbb{R}^d$ and $b_i \in \mathbb{R}$. First, we generate the auxiliary random vectors $\{\hat{\mathbf{a}}_i\}_{i=1}^n$ by sampling each entry of $\hat{\mathbf{a}}_i$ uniformly and independently from the interval $[-1, 1]$. Moreover, we generate $\{b_i\}_{i=1}^n$ independently from the standard normal distribution $\mathcal{N}(0, 1)$. Given $\{(\hat{\mathbf{a}}_i, b_i)\}_{i=1}^n$, we define an auxiliary function $\hat{f}(\mathbf{x}) = \log(\sum_{i=1}^n e^{\langle \hat{\mathbf{a}}_i, \mathbf{x} \rangle - b_i})$ and finally let $\mathbf{a}_i = \hat{\mathbf{a}}_i - \nabla \hat{f}(0)$ for $i = 1, \ldots, n$. As discussed in [16], the purpose of this procedure is to ensure that $\nabla f(0) = 0$ and thus $0$ is the unique minimizer of $f$. In our experiment, we set $n = d = 250$.

**NAG.** We implemented a monotone variant of the Nesterov accelerated gradient method as described in [54, Section 10.7.4]. Moreover, we determine the step size using a backtracking line search scheme.

**BFGS.** We implemented the classical BFGS algorithm, where the step size is determined by the Moré–Thuente line search scheme using an implementation by Diane O'Leary[1].

---

[1] http://www.cs.umd.edu/users/oleary/software/

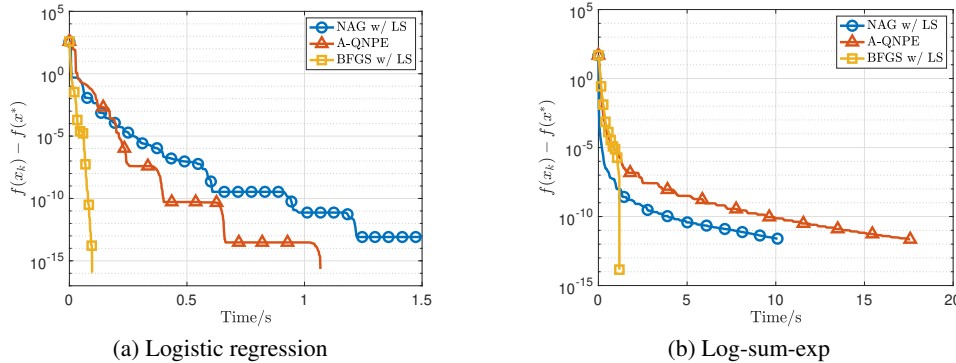

(a) Logistic regression  (b) Log-sum-exp

Figure 3: Plots of Suboptimality gap in terms of the running time.

**A-QPNE (our method).** We implemented our proposed A-QPNE method following the pseudocode in Algorithm 1, where the line search scheme is given in Subroutine 1 and the Hessian approximation update is given in Subroutine 2. Moreover, the implementations of the LinearSolver oracle and the SEP oracle are given by Subroutines 3 and 4, respectively.

### F.1 Additional Plots

In Fig. 3, we compare the performance of our proposed A-QNPE method with NAG and BFGS in terms of the running time. All experiments are conducted using MATLAB R2021b on a MacBook Pro with an Apple M1 chip and 16GB RAM. We observe from Fig. 3(a) that our method requires less running time than NAG due to its faster convergence, especially when we are seeking a solution of high accuracy. On the other hand, there are cases where our method is slower than NAG in terms of the running time, as shown in Fig. 3(b). This is because, in this case, the cost of gradient computation is comparable to the cost of matrix-vector product evaluation, and therefore our method incurs a higher computational cost per iteration than NAG.

