# OpenReview forum: "Accelerated Quasi-Newton Proximal Extragradient: Faster Rate for Smooth Convex Optimization"
_NeurIPS.cc/2023/Conference — NeurIPS 2023 spotlight_

### Official Review · Reviewer_N8Fk · 2023-07-05

**Soundness:** 4 excellent
**Presentation:** 4 excellent
**Contribution:** 4 excellent
**Rating:** 8
**Confidence:** 4

**Summary:**

This paper proposes a first-order optimization method for convex optimization that requires gradient computations and matrix-vector products. Based on recent advances on quasi-newton methods, this method converges at a rate that matches the optimal rate ($1/k^2$) when $k = \Omega(d)$ and improves upon the optimal rate when $k >>d$. In order to achieve this rate the authors consider a quasi-Newton method with a backtracking line search and a projection-free online algorithm to approximate the hessian of the objective.

**Strengths:**

This paper has several strengths:
- the rate proposed improved upon the best-known rates.
In particular, it is faster than the convergence rate of NAG when the number of iterations is larger than the dimension.
- The authors really try to give the main steps of the results in the main paper.
- While the paper is quite technical, I found that the contributions are well-presented,


**Weaknesses:**

I found the paper could be slightly improved:
- The experiments could support the theory more:
  - The logistic loss is strictly convex. Thus it seems that the rate in the experiments should be super-linear. It is not clear in the experiments if we are observing these super-linear rates or if we see sublinear convergence. I would instead try the log-sum exp loss (logistic regression for multi-class), which is not strictly convex
  - I would be curious to see if, actually, one can find a practical loss for which one can observe the sublinear rate $1/k^{2.5}$.
  - A plot with time as a x-axis could illustrate the fact that in many situations, matrix-vector product computations are not the bottleneck.
- Some details on the Online method are missing in the main paper:
   - The subroutine could be described (at least intuitively)
   - The main result about the regret bound could be stated (basically logarithmic regret)
   - In theorem 1 the probability p does not appear in the bound. I understand it only appears in the matrix-vector product complexity but it makes the statement quite odd. I would suggest the authors to at least mention it in a remark.

**Questions:**

Can you develop why you cannot use standard projection-free no-regret methods? such as


New Projection-free Algorithms for Online Convex Optimization with Adaptive Regret Guarantees
Dan Garber and Ben Kretzu


I assume it would not be straightforward since your notion of regret is different (dynamic regret) but I would be curious to know how your method compares with respect to this related work.


**Limitations:**

They authors mostly do address the limitation of their work.

They mention that their algorithm requires many matrix-vector products and, thus, their rate is faster than NAG if the gradient computation is considered the bottleneck.

They mention that their algorithm is slower than BFGS in practice.

---

> ### Author Rebuttal · Authors · 2023-08-09
>
> We thank the reviewer for the detailed comments. We address your concerns below.
>
> ---
>
> **Q1 It is not clear in the experiments if we are observing super-linear rates or if we see sublinear convergence. Try the log-sum exp loss, which is not strictly convex.**
>
> **A1**
> This is a very good observation. We conducted new experiments on the log-sum-exp function $f(x) = \log( \sum_{j=1}^n e^{\langle a_j, x\rangle - b_j}) $, where the dimension is $d=150$, the number of samples is $n=150$, and we follow the procedure in [1] to generate $\\{a_j\\}$ and $\\{b_j\\}$. As shown in Fig. 3 in the attached pdf, we observe that both NAG and A-QNPE converge at a sublinear rate.
>
> ---
>
> **Q2 I would be curious to see if one can find a practical loss for which one can observe the sublinear rate $O(1/k^{2.5})$.**
>
> **A2**
> Thanks for raising this point. So far we have not observed the sublinear rate $O(1/k^{2.5})$ in all of our experiments. Indeed, our theoretical result is pessimistic by nature, and the empirical performance of A-QNPE can be better than what the theory predicts. For instance, in our new experiment, Fig. 3(c) indicates that on this specific problem NAG converges at the rate of $O(1/k^{3})$, while A-QNPE converges at the rate of $O(1/k^5)$.
>
> ---
>
> **Q3 A plot with time as a x-axis.**
>
> **A3**
> Thanks for your suggestion. We have included additional plots in terms of the running time in the attached pdf.  All experiments are conducted using MATLAB R2021b on a MacBook Pro with an Apple M1 chip and 16GB RAM.
>
> Specifically, in Fig. 1 of the attached pdf file, we consider the logistic regression problem $f(x)= \frac{1}{n}\sum_{j=1}^n \log(1+e^{-y_j \langle a_j, x\rangle})$ as described in the paper, where the dimension is $d = 150$ and the number of samples is $n=2000$. As shown in Fig. 1(c), if we are seeking a solution of high accuracy,  our method can require less running time than NAG due to its faster convergence.
>
> In addition, in Fig. 2 of the attached pdf file, we consider the log-sum-exp function $f(x) = \log( \sum_{j=1}^n e^{\langle a_j, x\rangle - b_j}) $, where $d=150$, $n=150$, and we follow the procedure in [1] to generate $\\{a_j\\}$ and $\\{b_j\\}$. In this case, Fig. 2(c) shows that the run-time performance of A-QNPE is comparable to that of NAG.
>
> Finally, we would like to emphasize that the primary objective of our numerical experiments is to validate our theoretical discovery that A-QNPE can attain a faster convergence rate than NAG. With a more meticulous implementation, there is potential to enhance the practical efficacy of our method, and we have deferred this for future investigation.
>
> ---
>
> **Q4 Some details on the Online method are missing in the main paper.**
>
> **A4**
> Thanks for raising this point. Due to the page limit, we have to relegate the details of our online learning algorithm to the appendix (Sections C.3 and C.4). In the revision, we will describe the subroutine in more detail and present the key regret bound (Lemma 12 in Section D) in the main paper.
>
> ---
>
> **Q5 In theorem 1 the probability $p$ does not appear in the bound.**
>
> **A5** Thanks for raising this point. Note that our algorithm relies on the SEP oracle in Definition 2, which succeeds with a certain probability. To prove Theorem 1, we first show that, with probability at least $1-p$, every call of the SEP oracle is successful during the execution of our algorithm. Conditioned on this event, we proceed to prove our convergence rates, and thus they do not depend on $p$.  Following your suggestion, we will add a remark on this in our revision.
>
> ---
>
> **Q6 Can you develop why you cannot use standard projection-free no-regret methods?**
>
> **A6**
> This is a good point. We note that standard projection-free no-regret methods, such as online Frank-Wolfe [2], are based on a linear minimization oracle (LMO). Unfortunately, implementing the LMO in our setting is also computationally expensive. Specifically, the constraint set in our online learning problem is given by $\mathcal{Z} = \\{\mathbf{B} \in \mathbb{S}^d_+: 0 \preceq \mathbf{B} \preceq L_1 \mathbf{I}\\}$. Consider the linear minimization problem $\min_{\mathbf{X} \in \mathcal{Z}} \langle \mathbf{A}, \mathbf{X}  \rangle.$
> We first need to compute the eigendecomposition $\mathbf{A} = \mathbf{V}\mathbf{\Lambda} \mathbf{V}^\top$, where $\mathbf{V}$ is an orthogonal matrix and $\mathbf{\Lambda} = \mathrm{diag}(\lambda_1,\dots,\lambda_d)$ is a diagonal matrix. Then the solution is given by $\mathbf{V}\mathbf{\Lambda}' \mathbf{V}^\top$, where  $$\lambda_k' = \begin{cases}
>     0, & \text{if } \lambda_k \geq 0; \\\\
>     L_1, & \text{otherwise.}
> \end{cases}$$
> Hence, implementing the LMO  would require a full matrix eigendecomposition, which requires $O(d^3)$ arithmetic operations in general.
>
> On the other hand, as we discuss in this paper, the (approximate) separation oracle of $\mathcal{Z}$ is more efficient to implement, since it only requires computing the two extreme eigenvectors and eigenvalues of the given matrix. To the best of our knowledge, only the two recent papers [3,4] consider no-regret algorithms based on the separation oracle. In this paper, we develop our online learning algorithm based on [4] for two main reasons: (1) the algorithm in [4] appears to be simpler as it only requires one call to the separation oracle per iteration; (2) it is relatively straightforward to allow inexactness of the separation oracle. That said, we think it might also be possible to adapt the algorithm in [3] and achieve similar convergence guarantees.
>
>
> ---
>
> [1] A. Rodomanov and Y. Nesterov. Greedy quasi-Newton methods with explicit superlinear convergence, 2021.
>
> [2] E. Hazan and S. Kale. Projection-free online learning, 2012.
>
> [3] D. Garber and B. Kretzu. New Projection-free Algorithms for Online Convex Optimization with Adaptive Regret Guarantees, 2022.
>
> [4] Z. Mhammedi. Efficient projection-free online convex optimization with membership oracle, 2022.

---

> > ### Comment · Reviewer_N8Fk · 2023-08-14
> > **Thank you**
> >
> > I have read the rebuttal, and I maintain my score.

---

### Official Review · Reviewer_H1ti · 2023-07-06

**Soundness:** 3 good
**Presentation:** 3 good
**Contribution:** 3 good
**Rating:** 6
**Confidence:** 4

**Summary:**

This paper proposed a novel quasi-Newton method with faster global convergence rate. The algorithm uses the framework of MS acceleration and updates the Hessian estimator via online learning. The obtained convergence rate is impressive, it firstly show a faster global rate of quasi-Newton method which cannot be achieved by the first order methods.

**Strengths:**

The proposed algorithm is novel. The global convergence rate of $\tilde{\mathcal{O}}(\sqrt{d}/k^{-2.5})$ is a significant theoretical result.

**Weaknesses:**

1. Although the iteration complexity is impressive and significant, the total computation complexity of the proposed method is not very satisfactory. Compared with the first-order method, it requires additional Hessian-vector products with the complexity of $\min\lbrace d^{0.25}/\epsilon^{0.5},1/\epsilon^{0.625}\rbrace$ which may lead to an even higher computation cost than NAG.

2. The proposed method is complicated and may not practical to use. For the experimental part, the authors do not present the comparison between their method and the baselines in term of running time.


**Questions:**

1. Can you compare the detailed computation cost in a table to make the results more clear?
2. Can you conduct some experimental results in terms of the running time?

---

> ### Author Rebuttal · Authors · 2023-08-09
>
> We thank the reviewer for their comment. We address your concerns below.
>
> ---
>
> **Q1 Compare the detailed computation cost in a table.**
>
> **A1**
> Thanks for your suggestion. In the following table, we summarize the detailed computation cost of NAG and our method A-QNPE, and we will also include it in our revision. In addition, we would like to clarify that our method requires extra matrix-vector products, rather than Hessian-vector products. Indeed, as a quasi-Newton method, we only need access to the gradient oracle.
>
> |        | Gradient queries  | Matrix-vector products
> ------- |    -------------  | -------------
> NAG     | $O(\frac{1}{\epsilon^{0.5}})$ | N.A.
> A-QNPE  | $\tilde{O}(\min\\{\frac{1}{{\epsilon}^{0.5}},\frac{d^{0.2}}{\epsilon^{0.4}}\\})$  | $\tilde{O}(\min\\{\frac{d^{0.25}}{\epsilon^{0.5}}, \frac{1}{\epsilon^{0.625}}\\})$
>
>
>
>
> We observe that A-QNPE outperforms NAG in terms of gradient query complexity: it makes fewer or equal gradient queries, especially when $\epsilon < \frac{1}{d^2}$. On the other hand, A-QNPE requires additional matrix-vector product computations to implement the LinearSolver and SEP oracles.
>
>
> To assess the overall computation cost, we have to consider the cost of gradient computation, which varies depending on the specific problems. As a concrete example,
> consider the finite-sum minimization problem $f(x) = \frac{1}{n} \sum_{i=1}^nf_i(x)$. In this case, one gradient query typically costs $O(nd)$, while one matrix-vector product costs $O(d^2)$. Thus, the total computation cost of NAG and A-QNPE can be bounded by $O(\frac{nd}{\epsilon^{0.5}})$ and $ O(\frac{n d^{1.2}}{\epsilon^{0.4}}+ \frac{d^{2.25}}{\epsilon^{0.5}})$, respectively. In particular, our method will incur a lower computation cost when $\epsilon \ll \frac{1}{d^2}$ and $n \gg d^{1.25}$.
>
> As a final remark, we acknowledge that our method may be faster or slower than NAG, depending on the specific problem. Nevertheless, we would like to highlight that this is the first work to **theoretically demonstrate that quasi-Newton-type methods can outperform NAG in certain regimes**. Indeed, previous works [1,2] on quasi-Newton methods provide a convergence rate matching NAG, and as a result their overall computational cost will always be larger than NAG in theory. Thus, we believe our paper is an important conceptual advance for quasi-Newton methods and we leave the task of further reducing the computation cost as future work.
>
> ---
>
> **Q2 Conduct some experimental results in terms of the running time?**
>
> **A2**
> Thanks for your suggestion. We have included additional plots in terms of the running time; please check Figs. 1 and 2 in the attached pdf. All experiments are conducted using MATLAB R2021b on a MacBook Pro with an Apple M1 chip and 16GB RAM.
>
> Specifically, in Fig. 1 of the attached pdf file, we consider the logistic regression problem $f(x)= \frac{1}{n}\sum_{j=1}^n \log(1+e^{-y_j \langle a_j, x\rangle})$ as described in the paper, where the dimension is $d = 150$ and the number of samples is $n=2000$. As shown in Fig. 1(c), if we are seeking a solution of high accuracy,  our method can require less running time than NAG due to its faster convergence.
>
> In addition, in Fig. 2 of the attached pdf file, we consider the log-sum-exp function following the suggestion of Reviewer N8Fk. The loss function is given by $f(x) = \log( \sum_{j=1}^n e^{\langle a_j, x\rangle - b_j}) $, where the dimension is $d=150$, the number of samples is $n=150$, and we follow the procedure in [1] to generate $\\{a_j\\}$ and $\\{b_j\\}$. In this case, Fig. 2(c) shows that the run-time performance of A-QNPE is comparable to that of NAG.
>
> Finally, we would like to emphasize that the primary objective of our numerical experiments is to validate our theoretical discovery that A-QNPE can attain a faster convergence rate than NAG. With a more meticulous implementation, there is potential to enhance the practical efficacy of our method, and we have deferred this as a prospect for future investigation.
>
>
> [1] A. Rodomanov and Y. Nesterov. Greedy quasi-Newton methods with explicit superlinear convergence, 2021

---

### Official Review · Reviewer_CWLw · 2023-07-09

**Soundness:** 3 good
**Presentation:** 3 good
**Contribution:** 3 good
**Rating:** 5
**Confidence:** 3

**Summary:**

This paper proposes an accelerated quasi-Newton proximal extragradient method for solving unconstrained smooth convex optimization problems. The algorithm  can achieve a convergence rate of $\mathcal{O}\bigl(\min\{\frac{1}{k^2}, \frac{\sqrt{d\log k}}{k^{2.5}}\}\bigr)$, where $d$ is the problem dimension and $k$ is the number of iterations. In particular, in the regime where $k = \mathcal{O}(d)$, our method matches the optimal rate of $\mathcal{O}(\frac{1}{k^2})$ by Nesterov's accelerated gradient (NAG). Moreover, in the the regime where $k = \Omega(d \log d)$, it outperforms NAG and converges at a faster rate of $\mathcal{O}\bigl(\frac{\sqrt{d\log k}}{k^{2.5}}\bigr)$.

**Strengths:**

This paper proposes an accelerated quasi-Newton proximal extragradient method for solving unconstrained smooth convex optimization problems. The algorithm  can achieve a convergence rate of $\mathcal{O}\bigl(\min\{\frac{1}{k^2}, \frac{\sqrt{d\log k}}{k^{2.5}}\}\bigr)$, where $d$ is the problem dimension and $k$ is the number of iterations. In particular, in the regime where $k = \mathcal{O}(d)$, our method matches the optimal rate of $\mathcal{O}(\frac{1}{k^2})$ by Nesterov's accelerated gradient (NAG). Moreover, in the the regime where $k = \Omega(d \log d)$, it outperforms NAG and converges at a faster rate of $\mathcal{O}\bigl(\frac{\sqrt{d\log k}}{k^{2.5}}\bigr)$.

**Weaknesses:**

I have a concern about the total computation cost.
Let us consider the finite-sum form, that is, $f(x) = \frac{1}{n} \sum_{i=1}^nf_i(x)$ with $n = \mathcal{O}(d)$.
Then, the algorithm takes $N_\eps$ gradient queries which implies $\mathcal{O}(N_\eps d^2)$ computation cost.
By Theorem 2 (c), the cost of computing SEP oracles, the computation cost is $ \mathcal{O}(N_\eps^{1.25} d^2) $.
In this case, A-QNPE does not have advantages over  Nesterov's accelerated gradient (NAG).

**Questions:**

No

---

> ### Author Rebuttal · Authors · 2023-08-09
>
> We thank the reviewer for their comment. We address your concern below.
>
> ---
>
> **Q1 Comparison with NAG in terms of the total computational cost.**
>
> **A1** Thanks for raising this point. For easier comparison, we follow the suggestion by Reviewer H1ti and summarize the computation cost of NAG and our proposed method A-QNPE to achieve an $\epsilon$ accuracy in the following table.
>
> |        | Gradient queries  | Matrix-vector products
> ------- |    -------------  | -------------
> NAG     | $O(\frac{1}{\epsilon^{0.5}})$ | N.A.
> A-QNPE  | $\tilde{O}(\min\\{\frac{1}{{\epsilon}^{0.5}},\frac{d^{0.2}}{\epsilon^{0.4}}\\})$  | $\tilde{O}(\min\\{\frac{d^{0.25}}{\epsilon^{0.5}}, \frac{1}{\epsilon^{0.625}}\\})$
>
>
> We observe that A-QNPE outperforms NAG in terms of gradient query complexity: it makes fewer or equal gradient queries, especially when $\epsilon < \frac{1}{d^2}$. On the other hand, A-QNPE requires additional matrix-vector product computations to implement the LinearSolver and SEP oracles.
>
>
> To assess the overall computation cost, we have to consider the cost of gradient computation, which varies depending on the specific problems. As a concrete example,
> consider the finite-sum minimization problem $f(x) = \frac{1}{n} \sum_{i=1}^nf_i(x)$. In this case, one gradient query typically costs $O(nd)$, while one matrix-vector product costs $O(d^2)$. Thus, the total computation cost of NAG and A-QNPE can be bounded by $O(\frac{nd}{\epsilon^{0.5}})$ and $ O(\frac{n d^{1.2}}{\epsilon^{0.4}}+ \frac{d^{2.25}}{\epsilon^{0.5}})$, respectively. In particular, our method will incur a lower computation cost when $\epsilon \ll \frac{1}{d^2}$ and $n \gg d^{1.25}$.
>
> As a final remark, we acknowledge that our method may be faster or slower than NAG, depending on the specific problem. Nevertheless, we would like to highlight that this is the first work to **theoretically demonstrate that quasi-Newton-type methods can outperform NAG in certain regimes**. Indeed, previous works [1,2] on quasi-Newton methods provide a convergence rate matching NAG, and as a result their overall computational cost will always be larger than NAG in theory. Thus, we believe our paper is an important conceptual advance for quasi-Newton methods and we leave the task of further reducing the computation cost as future work.
>
> ---
>
> References:
>
> [1] K. Scheinberg and X. Tang. Practical inexact proximal quasi-Newton method with global complexity analysis, 2016
>
> [2] H. Ghanbari and K. Scheinberg. Proximal quasi-Newton methods for regularized convex optimization with linear and accelerated sublinear convergence rates, 2018.

---

> > ### Comment · Reviewer_CWLw · 2023-08-12
> >
> > I have read the rebuttal.

---

### Official Review · Reviewer_MQHh · 2023-07-26

**Soundness:** 4 excellent
**Presentation:** 3 good
**Contribution:** 4 excellent
**Rating:** 7
**Confidence:** 4

**Summary:**

This paper uses the optimal and adaptive Monteiro Svaiter acceleration framework to create a quasi-Newton method that solves unconstrainted convex problems with Lipschitz gradients and Lipschitz hessians at Nesterov's accelerated rate $O(1/k^2)$ but when the number of iterations is greater enough than the dimension, namely $\Omega(d\log d)$ it provides better convergence guarantees in terms of gradient oracle complexity. The number of operations is superlinear $\widetilde{O}(N_\epsilon^{1.25})$ in the number of gradient queries $N_\epsilon$ and memory that is quadratic in the dimension is used.


**Strengths:**

Authors use a great range of technical and powerful tools like optimal and adaptive MS acceleration, they use projection free online learning with a separation oracle, the conjugate gradients method and Lanczos algorithms.  The results are novel and interesting.


**Weaknesses:**

The abstract (and some parts of the paper) says that the method matches the optimal rate O(1/k^2) by NAG, and this rate is known optimal for functions that are convex smooth. Here your setting is convex smooth + Lipchitz Hessians. It would be good to explicitly comment on how that construction has a hessian Lipschitzness L_2 = 0 (since it is a quadratic) and therefore it also applies to your setting.

I guess you can modify your algorithm to get results under the additional assumption of \mu strong convexity. This could take a lot of work, but on the other hand you can follow the spirit of the usual reductions to show rates for your algorithm under strong convexity by using a sequence of restarts. So I would suggest to add a remark with this. The argument would be like this, you run the algorithm in stages and after each stage you guarantee you halve the distance to x^ast and so you only need to run a logarithmic number of stages. In order to do that, you want to run the algorithm for k iterations such that you guarantee the last equality here:
$$
\mu/2 \|x_t-x^\ast\|^2 \leq f(x_t) - f(x^\ast) \leq O(\text{your bound}) = \mu/8 \|x_0-x^\ast\|^2
$$
The first inequality is strong convexity and the second one is your guarantee. In the regime in which \mu is small enough, the number of iterations necessary in each stage should be \Omega(d\log(d)) and so your improved bound kicks in.
After doing this, it would be desirable to have a comparison of these results with the quasi-Newton results mentioned in the introduction that apply to strongly convex functions.


Similarly, regarding "However, all of the results above only apply under the restrictive assumption that the objective function f is strictly or strongly convex. In the more general setting where f is merely convex...". Given the reduction that solves convex smooth  problems by regularizing with + \epsilon/ R^2 \norm{x_0-x^\ast}^2, where R is an upper bound on the initial distance to the minimizer, you should discuss what one can get with previous methods under the reduction. That is, the strongly convex setting is not necessarily restrictive and a comparison / discussion should be done.

L839 "subourtine" -> "subroutine"


**Questions:**

.

**Limitations:**

see above

---

> ### Author Rebuttal · Authors · 2023-08-09
>
> We thank the reviewer for the insightful comments. We address your concerns below.
>
> ---
>
> **Q1 In your setting the function is convex smooth + Lipchitz Hessians. It would be good to comment on how the worst-case construction also applies to your setting.**
>
> **A1** This is an excellent point. The lower bound of $\Omega(1/k^2)$ for the class of convex smooth functions is established by a worst-case quadratic function, whose Hessian is constant (Lipschitz continuous with $L_2 = 0$). Therefore, the additional assumption of Lipschitz Hessian does not eliminate this worst-case construction from the considered problem class, and thus the $\Omega(1/k^2)$ lower bound also applies to our setting. Thanks for raising this excellent point and we will add it to the revision.
>
> ---
>
> **Q2 Use restart to extend the results to the strongly convex setting.**
>
> **A2** This is a very good suggestion. Indeed, it is possible to use the restarting technique to further extend our result to the strongly convex setting, as suggested by the reviewer. However, it appears that such a reduction would only lead to a linear rate with a better dependence on the condition number, instead of a superlinear rate that we would expect from a quasi-Newton method in the strongly convex setting.
>
> To be more precise, if we follow the arguments of the reviewer and run our algorithm in multiple stages, where we guarantee that the distance to $x^\*$ is halved after each stage, we obtain from Theorem 1 that
> $$f(x_t) - f(x^\*) = O\left( \min\left\\{\frac{L_1\\|x_0-x^\*\\|^2}{t^2}, \frac{L_1\sqrt{d}\\|x_0-x^\*\\|^2}{t^{2.5}}\right\\} \right).$$ To get this bound, we upper bound $\\|B_0 - \nabla^2 f(z_0)\\|_F^2$ by $L_1^2 d$, ignore the log factor in (19) and only focus on the dominant term for simplicity.
> As the reviewer points out, by using strong convexity we have $\\|x_t-x^\*\\| \leq \frac{1}{2}\\|x_0-x\^\*\\|$ if $f(x_t)-f(x^\*) \leq \frac{\mu}{8}\\|x_0-x^\*\\|^2$. Hence, the number of iterations required in each stage can be bounded by $O(\min\\{({\frac{L_1}{\mu}})^{0.5}, d^{0.2}(\frac{L_1}{\mu})^{0.4}\\})$, which implies a total complexity of $$ O\left(\min\left\\{\left(\frac{L_1}{\mu}\right)^{0.5}, d^{0.2}\left(\frac{L_1}{\mu}\right)^{0.4}\right\\} \log \frac{1}{\epsilon}\right).$$
>
> In the regime where $d \leq \sqrt{\frac{L_1}{\mu}}$, the obtained complexity bound outperforms NAG in terms of the dependence on the condition number.
> On the other hand, we note that several papers [1,2] have established a local non-asymptotic superlinear rate of the form $O((1/\sqrt{k})^k)$ for classical quasi-Newton methods and their variants. More recently, a global non-asymptotic superlinear rate is also shown for a quasi-Newton proximal extragradient method [3]. In comparison, the restarting scheme described above can only achieve global linear convergence.
>
> While the main focus of this paper is on demonstrating a provable gain for a quasi-Newton-type
> method over NAG in the convex setting, the above argument for extending our algorithm to the strongly convex setting is an interesting observation and we will add it as a remark to our revised paper.
>
> ---
>
> **Q3 Discuss what one can get with previous methods under the regularization reduction.**
>
> **A3**
> Thanks for your insightful comment. As the reviewer rightly pointed out, one can regularize $f$ with $\frac{\epsilon}{R^2}\\|x-x_0\\|^2$ to reduce a convex smooth problem into a strongly-convex one with $\mu = \frac{\epsilon}{R^2}$. However, to the best of our knowledge, applying this reduction directly to the existing analysis of quasi-Newton methods would not lead to a global complexity bound better than the one for NAG, as we elaborate next.
>
> - The results in [1,2] are crucially based on local analysis and require the initial point $x_0$ to be close enough to the optimal solution $x^*$. However, it is unclear how to explicitly bound the number of iterations before the iterate enters the local neighborhood, and even if this can be done, it seems unlikely that the total complexity would be better than NAG, as these results only provide a local convergence analysis.
>
> - The result from [3] seems to be the only strongly-convex result that can be compared with NAG in the convex setting using the mentioned regularization idea, as it provides a global convergence analysis with an explicit overall complexity bound. Based on the discussions in Appendix D.2 of [3], the authors showed a global complexity bound in the form of
> $O\left(\min\left\\{\frac{L_1}{\mu} \log \frac{1}{\epsilon},d^{\frac{1}{3}}\left(\frac{L_1}{\mu}\log\frac{1}{\epsilon}\right)^{\frac{2}{3}} \right\\}\right)$ for strongly-convex objectives. Since we have $\mu = \frac{\epsilon}{2D^2}$ under the reduction, this translates into a complexity bound of $\tilde{O}\left(\min\left\\{ \frac{1}{\epsilon}, d^{\frac{1}{3}}\left(\frac{1}{\epsilon}\right)^{\frac{2}{3}}\right\\}\right)$ for convex problems, which is worse than the bound $O(\left(\frac{1}{\epsilon}\right)^{\frac{1}{2}})$ by NAG. This is conceivable since there is no form of acceleration in the proposed method of [3].
>
> Thus, we conclude that simply applying the standard reduction to the existing analysis would not result in a complexity bound better than NAG. This highlights the need for a distinct algorithm and analysis specifically tailored to the convex setting, as presented in this paper. We will add a remark to the revised paper regarding this point.
>
> ---
>
> **Typo.**
> Thanks for catching the typo. We will fix this in the revision.
>
> ---
>
> References:
>
> [1] Q. Jin and A. Mokhtari. Non-asymptotic superlinear convergence of standard quasi-Newton methods, 2022.
>
> [2] A. Rodomanov and Y. Nesterov. New results on superlinear convergence of classical quasi-Newton methods, 2021.
>
> [3] R. Jiang, Q. Jin, and A. Mokhtari. Online learning guided curvature approximation: A quasi-Newton method with global non-asymptotic superlinear convergence, 2023.

---

> > ### Comment · Reviewer_MQHh · 2023-08-12
> > **reply**
> >
> > I have read the rebuttal. Please add those three points to the paper. Good work.

---

### Author Rebuttal · Authors · 2023-08-09

We thank all reviewers for their time and effort in evaluating our paper. Following the suggestions by **Reviewer H1ti** and **Reviewer N8Fk**, we have included additional plots in the attached pdf file.

- In Fig. 1, we consider the logistic regression problem $f(x)= \frac{1}{n}\sum_{j=1}^n \log(1+e^{-y_j \langle a_j, x\rangle})$ as described in the paper, where the dimension is $d = 150$ and the number of samples is $n=2000$. As shown in Fig. 1(c), if we are seeking a solution of high accuracy,  our method can require less running time than NAG due to its faster convergence.

- In Fig. 2, we consider the log-sum-exp function following the suggestion of **Reviewer N8Fk**. The loss function is given by $f(x) = \log( \sum_{j=1}^n e^{\langle a_j, x\rangle - b_j}) $, where the dimension is $d=150$, the number of samples is $n=150$, and we follow the procedure in [1] to generate $\\{a_j\\}$ and $\\{b_j\\}$. In this case, Fig. 2(c) shows that the run-time performance of A-QNPE is comparable to that of NAG.

- In Fig. 3, we plot both the suboptimality gap $f(x_k) - f(x^*)$ and the number of iterations on a log scale for the log-sum-exp experiment. For this specific problem, we can observe empirically that NAG converges at a sublinear rate of $O(1/k^{3})$, while A-QNPE converges at a faster rate of $O(1/k^5)$.

---

[1] A. Rodomanov and Y. Nesterov. Greedy quasi-Newton methods with explicit superlinear convergence, 2021

---

### Decision · Program_Chairs · 2023-09-21

**Decision:**

Accept (spotlight)

**Comment:**

The reviewers reached a consensus about accepting this paper, which two of the reviewers expressing excitement about the progress this paper makes in theoretically establishing quasi-Newton methods. Consequently, I recommend acceptance of the paper.

When making the camera-ready revision, please take care to implement all the changes proposed during the review and rebuttal process, and particularly the ones concerning the discussion and empirical evaluation of the computational cost of the proposed method.